# Structural basis for recognition of diverse localizing mRNAs by Egl–BicD

Kashish Singh [1,3], Sabila Chilaeva[2,3], Mark A. McClintock [2,3] ✉,
Andrew P. Carter [1] ✉ & Simon L. Bullock [2] ✉

Localization of mRNAs is a widespread mechanism for dictating where proteins operate in cells and underpins many fundamental processes, from embryonic patterning to synaptic plasticity. This spatial control is mediated by the interaction of 'localization signals' in target mRNAs with RNA-binding proteins (RBPs). These signals frequently lack overt sequence or structural patterns, raising the question of how specificity is achieved. Here we investigate this issue using the *Drosophila* RBP Egalitarian (Egl), which couples mRNAs to microtubule-based transport through Bicaudal D (BicD) and the dynein motor. We present cryo-electron microscopy structures of Egl–BicD bound to six different RNAs. Egl uses multiple noncanonical double-stranded RNA-binding domains to cooperatively form a recognition pocket around localization signals. Despite substantial variation in length and sequence, each signal adopts a bent stem-loop conformation that, together with base-pair identities at two defined sites, drives Egl engagement. We further demonstrate that Egl dimers couple RNA binding to transport initiation through coincident detection of two RNA elements within the same transcript. Thus, localizing mRNAs are recognized through a combination of shape, positional sequence features and number of structured RNA elements. This work reveals a molecular strategy by which diverse mRNAs can be selectively engaged by a single RBP.

Localization of mRNAs to specific regions of cells is an evolutionarily conserved strategy for determining where proteins are synthesized and function[1,2]. This process is essential for numerous biological processes, including embryonic patterning, germline development, establishment of cell polarity, axonal morphogenesis and synaptic plasticity[3–6]. The main pathways driving mRNA localization are local protection from degradation, hitchhiking on motile organelles and direct coupling to cytoskeletal motors by adaptor proteins[2,3,7–9]. These mechanisms all rely on the recognition of 'localization signals' in mRNAs by RNA-binding proteins (RBPs)[8,10].

Our current understanding of how RBPs identify mRNAs comes largely from cases involving conserved RNA sequences that form linear or structural motifs[9–12]. However, such clear patterns have proven elusive for most localizing transcripts[9]. Even in cases where RNA localization signals have been identified and are known to recruit the same RBP, they have considerable divergence in sequence and size[13–23]. This variability raises the question of how specificity for diverse mRNAs is achieved. The difficulty in identifying localization signals in transcripts has led to suggestions that interactions with RBPs rely on poorly defined structural motifs[13–21,24–26] or combinatorial low-affinity interactions within ribonucleoprotein granules or biomolecular condensates[27–29]. However, the molecular basis of how such mechanisms can be used for recognition is not well understood.

An attractive model for investigating how RBPs associate with multiple targets is the Egalitarian (Egl) protein. In the fruit fly *Drosophila melanogaster*, Egl is responsible for patterning and segmentation of

[1]Division of Structural Studies, MRC Laboratory of Molecular Biology, Cambridge, UK. [2]Division of Cell Biology, MRC Laboratory of Molecular Biology, Cambridge, UK. [3]These authors contributed equally: Kashish Singh, Sabila Chilaeva, Mark A. McClintock. ✉e-mail: markmc@mrc-lmb.cam.ac.uk; cartera@mrc-lmb.cam.ac.uk; sbullock@mrc-lmb.cam.ac.uk

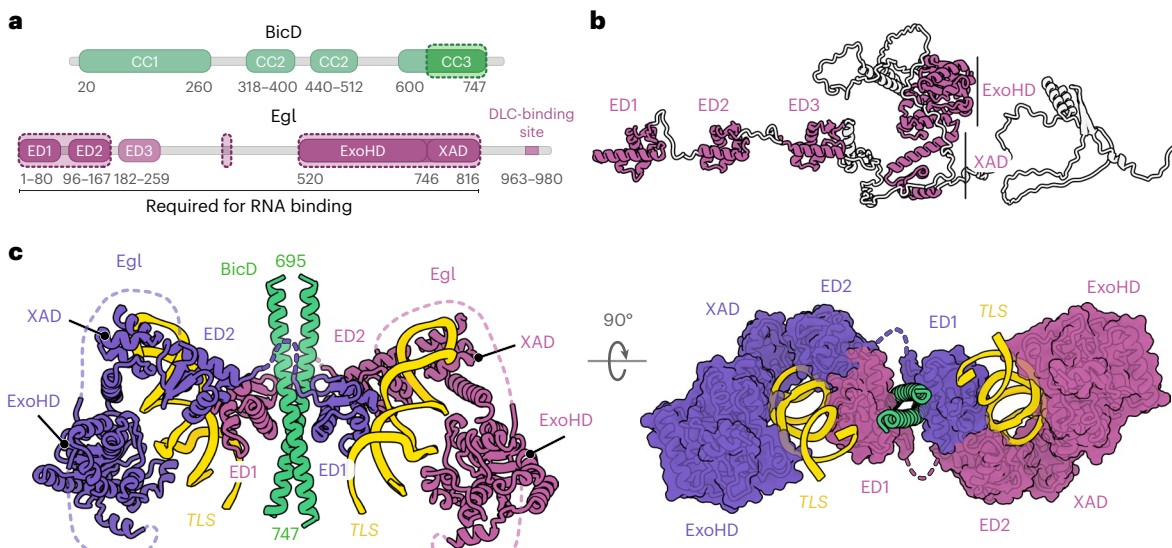

**Fig. 1 | Architecture of the Egl–BicD–*TLS* complex. a**, Domain architecture of Egl and BicD. Regions enclosed in dashed lines indicate those visualized in the cryo-EM structure. The Egl C terminus contains a dynein light chain (DLC)-binding site that promotes Egl function in vivo but is dispensable for dynein activation and RNA binding[34,35,47]. **b**, AlphaFold2 model of Egl with linker regions extended to facilitate visualization of predicted folded domains (colored in magenta; nonextended prediction in Extended Data Fig. 1a). **c**, Cartoon and surface models of the Egl–BicD–*TLS* complex, as determined by cryo-EM. Unresolved linkers between domains are represented by dotted lines. The structure was resolved to a nominal resolution of 3.4 Å, with local resolution varying from 2.9 to 4.5 Å (Supplementary Fig. 3), allowing assignment of RNA and protein sequences to the cryo-EM density (Extended Data Fig. 2a,b).

the body axes during oogenesis and embryogenesis[5,30,31], as well as dendritic morphogenesis in larval stages[32]. It recruits many developmentally important mRNAs to the microtubule motor dynein by binding Bicaudal D (BicD), a long coiled-coil protein with an evolutionarily conserved role in regulating dynein activity[24,33]. Association with a target mRNA stabilizes the interaction of two copies of Egl with BicD[34,35]. This relieves BicD autoinhibition, leading to activation of dynein movement by the dynactin complex. Many of the RNA localization signals that bind Egl have been defined experimentally[16–18,24,34,35]. However, beyond forming a double-stranded RNA (dsRNA) stem-loop structure, these signals have no obvious similarity in primary sequence or secondary structure[16–18,36,37]. Thus, it is unclear how mRNA recognition is achieved.

## Results

### Architecture of Egl–BicD bound to an RNA localization signal

Egl is a 1,004-amino acid protein in which the first 814 residues are required for specific binding to localization signals[24]. AlphaFold2 (ref. 38) predicts that the RNA-binding region contains five folded domains (Fig. 1a,b and Extended Data Fig. 1a). At the N terminus are three 'Egl domains' (ED1–ED3), which adopt a winged helix–turn–helix-like architecture[39] that closely resembles the LOTUS domain (named for its presence in Limkain, Oskar and TDRD5/7 proteins[40–44]; Extended Data Fig. 1b–e). Sequence comparison of Egl orthologs shows that ED1 and ED2 are well conserved across species, whereas ED3 is comparatively divergent (Supplementary Figs. 1 and 2). The EDs are followed by a conserved module comprising a 3′–5′ exonuclease homology domain (ExoHD)[33] and a C-terminal tetrahelical region, which we term the exonuclease-adjacent domain (XAD; Supplementary Figs. 1 and 2). The individual EDs and ExoHD–XAD module are connected by flexible linkers that have low sequence conservation (Supplementary Figs. 1 and 2). Whereas LOTUS and 3′–5′ exonuclease domains can interact with single-stranded RNA[42,45], there is no prior evidence they bind dsRNA such as that found in stem-loop localization signals.

To elucidate how Egl recognizes its targets, we determined a cryo-electron microscopy (cryo-EM) structure of the purified Egl–BicD complex bound to the 44-nt 'transport and localization signal' (*TLS*) of the *Drosophila fs(1)K10* mRNA[16,46] (hereafter *K10*; Fig. 1c, Extended Data Figs. 1f and 2a–c, Supplementary Fig. 3 and Table 1). This reveals that two *TLS* stem loops are recruited by two Egl molecules to the C-terminal coiled coil (CC3) of BicD, which is the only region of the latter protein that is resolved in the structure (Fig. 1a,c). Each copy of Egl binds opposite faces of the BicD coiled coil through ED1 (Fig. 1c and Extended Data Fig. 2d,e). ED1 also contacts the *TLS* as part of an RNA-binding pocket that includes ED2, ExoHD and XAD (Fig. 1c). All residues in contact with the RNA are well conserved, particularly on ED1, ED2 and XAD (Extended Data Fig. 3a and Supplementary Figs. 1 and 2). ED3 is not visible in the structure, indicating that it does not stably contact the localization signal. Strikingly, within each pocket, the *TLS* is bound by ED1 and ED2 from different Egl polypeptides. This is because of the flexible linker between ED1 and ED2 that allows the latter domain to contact ED1 of the neighboring Egl molecule but is too short to permit intramolecular ED1–ED2 contacts (Fig. 1c). Thus, RNA binding requires two Egl molecules, explaining how localization signals stabilize the dimeric form of Egl that activates transport[34,35].

### The structural basis of dsRNA binding by Egl

Each *TLS* comprises a 4-nt loop, two A-form helices ('upper' and 'lower'; Supplementary Tables 1 and 2), and a short intervening helix flanked by 3′-strand bulges (Fig. 2a). ED1 and ED2 bind on one side of the stem loop, contacting the lower and upper helices, respectively (Fig. 2b). A conserved interface between the EDs positions them to bind adjacent RNA minor grooves (Fig. 2c and Extended Data Fig. 3b). At both sites, the second α-helix (H2) of the ED inserts into the minor groove and, together with neighboring residues, makes multiple contacts with the RNA sugar-phosphate backbone (Extended Data Fig. 3c,d). In addition, side chains from H2 interact with individual bases (Fig. 2c). In ED1, His31 is close to both the U6–A39 and U7–A38 base pairs, with the geometry suggesting that it is more likely to make a hydrogen bond with A39. Cys35, which is substituted for a tyrosine in the classical *egl⁴ᵉ* loss-of-function mutation[47], sits close to the uracil of U6–A39. In ED2, Ser127 and Gln128 make hydrogen bonds with the adenines of U15–A28 and A16–U27, respectively.

The XAD sits in the major groove formed between the *TLS* upper helix and loop, where it makes extensive electrostatic contacts with the

**Table 1 | Cryo-EM data collection and refinement statistics for structures of Egl–BicD bound to *TLS* and *hSL1***

| Sample | *TLS*-A | *TLS*-B | *TLS*-C | *TLS*-D | *TLS*-E | *hSL1*-A | *hSL1*-B | *hSL1*-C |
|---|---|---|---|---|---|---|---|---|
| **EM Data Bank** | EMD-54292 | EMD-54293 | EMD-54294 | EMD-54295 | EMD-54296 | EMD-54297 | EMD-54298 | EMD-54299 |
| **Protein Data Bank** | PDB 9RVY | PDB 9RVZ | PDB 9RW0 | PDB 9RW1 | PDB 9RW2 | PDB 9RW3 | PDB 9RW4 | PDB 9RW5 |
| **Data collection and processing** | | | | | | | | |
| Micrographs collected | 8,568 | | | | | 12,420 | | 16,453 |
| Magnification | ×81,000 | | | | | ×81,000 | | ×81,000 |
| Voltage | 300 | | | | | 300 | | 300 |
| Pixel size (Å) | 1.059 | | | | | 1.06 | | 0.91 |
| Electron exposure (e⁻ per Å²) | 48 | | | | | 50 | | 52 |
| Frames collected | 40 | | | | | 50 | | 56 |
| Defocus range (µm) | 1–3.5 | | | | | 0.5–3.5 | | |
| Final particle images | 540,532 | 256,835 | 238,571 | 76,319 | 138,411 | 274,911 | 42,550 | 33,983 |
| Symmetry imposed | $C_1$ | $C_1$ | $C_1$ | $C_1$ | $C_1$ | $C_1$ | $C_1$ | $C_1$ |
| Map resolution (Å) | 3 | 3.2 | 3.4 | 3.7 | 3.6 | 3.2 | 3.9 | 4.1 |
| FSC[1] threshold | 0.143 | 0.143 | 0.143 | 0.143 | 0.143 | 0.143 | 0.143 | 0.143 |
| Sharpening *B* factor (Å²) | −60 | −60 | −60 | −60 | −60 | −60 | −30 | −30 |
| **Refinement** | | | | | | | | |
| Model resolution (Å) | 3 | 3.4 | 3.6 | 3.7 | 4.1 | 3.3 | 4 | 4.1 |
| FSC threshold | 0.5 | 0.5 | 0.5 | 0.5 | 0.5 | 0.5 | 0.5 | 0.5 |
| Model composition | | | | | | | | |
| Nonhydrogen atoms | 7,634 | 7,682 | 8,225 | 8,373 | 8,134 | 7,502 | 7,788 | 6,832 |
| Protein residues | 720 | 722 | 1,028 | 1,028 | 1,028 | 717 | 1,002 | 1,014 |
| Nucleotides | 88 | 88 | 88 | 88 | 88 | 86 | 86 | 86 |
| Mean *B* factors (Å²) | | | | | | | | |
| Protein | 76.88 | 90.92 | 99.61 | 87.49 | 128.89 | 62.4 | 136.75 | 184.74 |
| Nucleotides | 69.48 | 91.8 | 83.31 | 52.23 | 100.42 | 78.9 | 78.83 | 123.32 |
| R.m.s.d. | | | | | | | | |
| Bond lengths (Å) | 0.007 | 0.004 | 0.005 | 0.003 | 0.003 | 0.004 | 0.002 | 0.004 |
| Bond angles (°) | 0.755 | 0.696 | 0.742 | 0.68 | 0.65 | 0.667 | 0.551 | 1.005 |
| **Validation** | | | | | | | | |
| Clashscore | 5.93 | 4.7 | 6.37 | 5.7 | 6.66 | 5.97 | 9.22 | 2.9 |
| MolProbity score | 1.65 | 1.47 | 1.74 | 1.54 | 1.64 | 1.69 | 1.73 | 1.43 |
| Rotamer outliers (%) | 0.16 | 0.16 | 0.29 | 0.26 | 0 | 0.17 | 0 | 0 |
| Ramachandran plot | | | | | | | | |
| Favored (%) | 95.31 | 96.46 | 94.25 | 96.53 | 96.03 | 94.74 | 96.35 | 94.89 |
| Allowed (%) | 4.69 | 3.54 | 5.75 | 3.27 | 3.97 | 5.26 | 3.35 | 5.11 |
| Outliers (%) | 0 | 0 | 0 | 0.2 | 0 | 0 | 0.3 | 0 |

For each complex, the best-resolved conformation (structure A) was used for analysis (Supplementary Figs. 3 and 4). [1]FSC, Fourier shell correlation.

RNA backbone (Fig. 2d and Extended Data Fig. 3e–h). These interactions would not be possible with a regular A-form RNA helix and require the widened major groove at the tip of the *TLS* stem loop. Furthermore, an interaction between XAD and ED2 sets the distance between the ED2 interaction site and the start of the loop (Extended Data Fig. 3e). Completing the RNA-binding pocket is the ExoHD, which contacts the opposite side of the *TLS* to ED1–ED2 (Fig. 2b). The putative catalytic residues of the ExoHD[47] are -15 Å away from the RNA (Extended Data Fig. 3i), indicating that they do not have a role in *TLS* recognition. This is consistent with the dispensability of these residues for RNA binding in vitro[24] and Egl function in vivo[47]. The interaction of the ExoHD with the RNA involves two clusters of positively charged residues (Fig. 2e and Extended Data Fig. 3h). Whereas the ExoHD residues interacting

with the *TLS* upper helix are well resolved, those contacting the lower helix are not, suggesting a more dynamic interaction with this region of the RNA.

Microscale thermophoresis (MST) (Extended Data Fig. 4a–c) revealed that deleting ED2 or XAD severely reduced binding of Egl–BicD to the *TLS* (Fig. 2f). By contrast, and in line with its absence in our structure, removal of ED3 did not affect *TLS* binding (Extended Data Fig. 4d). As deleting ED1 or ExoHD was not technically possible (Methods), we introduced point mutations targeting their RNA-binding interfaces and found that these significantly impaired association with the *TLS*, albeit less dramatically than the domain deletions of ED2 or XAD (Fig. 2f).

Collectively, our data show how a combination of four noncanonical dsRNA-binding domains engage distinct features of the *TLS*.

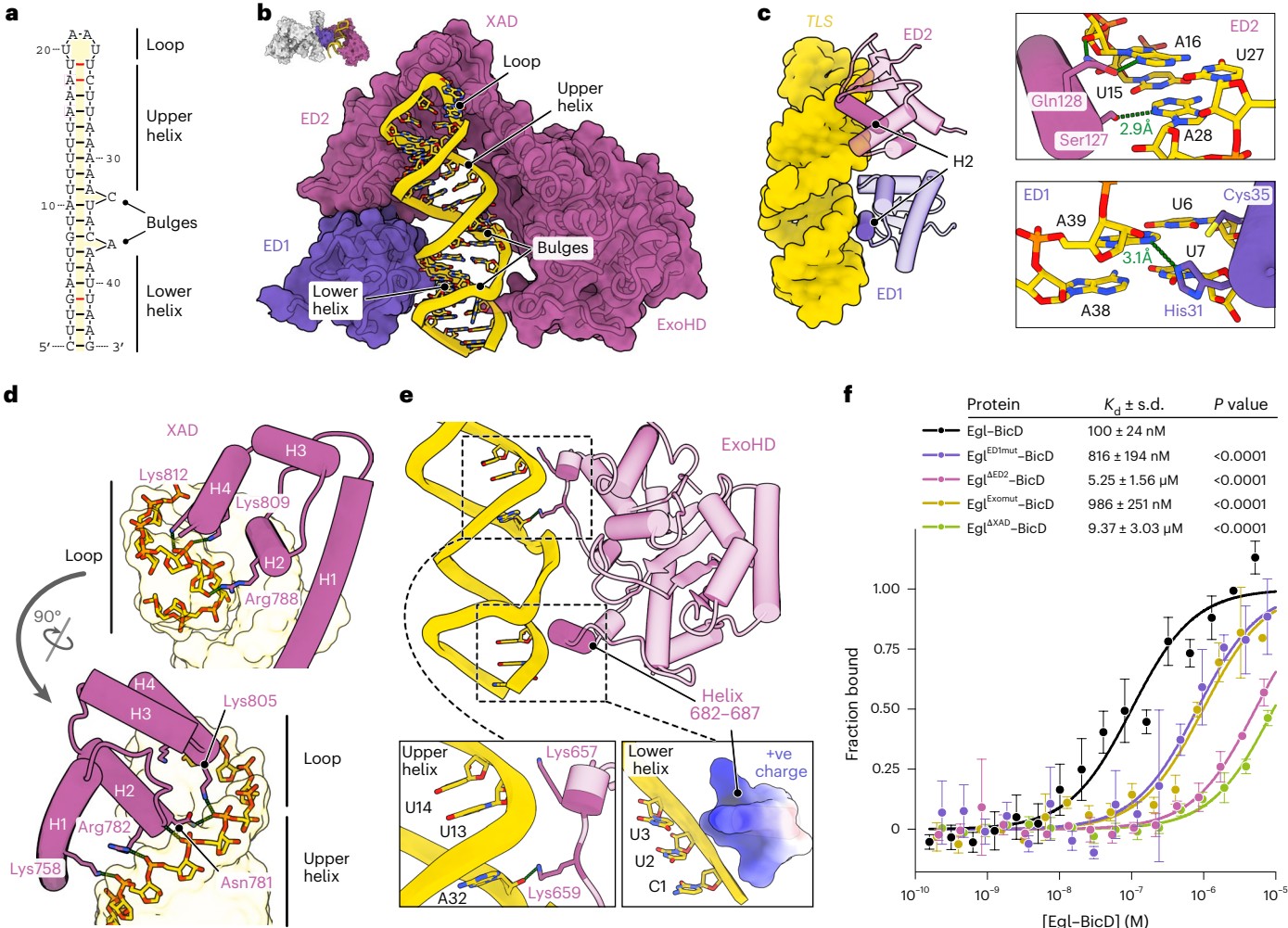

**Fig. 2 | Domains from Egl dimers form composite RNA-binding pockets.**
**a**, Empirically determined secondary structure of the *TLS* in complex with Egl–BicD. Noncanonical base pairs are indicated by red lines. **b**, Overview of the *TLS* in the composite RNA-binding pocket formed from two Egl polypeptides. **c**, Interaction of ED1 and ED2 with the *TLS* minor groove (left) and side-chain contacts made by these domains with RNA bases (right). Dotted green lines denote hydrogen bonds. Additional interactions with the sugar-phosphate backbone are shown in Extended Data Fig. 3c,d. **d,e**, Interactions of XAD (**d**) and ExoHD (**e**) with the *TLS* RNA backbone. Because of limited local resolution of ExoHD at the lower helix of the *TLS*, a surface charge representation of residues 682–687 is shown in **e** to indicate likely electrostatic interactions. Additional interactions with the

sugar-phosphate backbone are shown in Extended Data Fig. 3f,g. **f**, MST curves for the *TLS* bound to Egl–BicD variants. The Egl^ED1mut–BicD variant has a Cys35Tyr substitution, which is present in the classical *egl⁴ᵉ* mutation[47]. This substitution impairs mRNA association with Egl in vivo[72] and is predicted by our structure to clash with nucleotides in the *TLS* minor groove. The Egl^Exomut–BicD variant has Lys657, Lys659, Arg686 and Arg687 substituted to alanine. Data points show the mean ± s.d. for three (Egl–BicD variants) or six (wild-type Egl–BicD) independent experiments per condition, from which best-fit values for $K_d$ ± s.d. were derived. *P* values were determined using pairwise one-tailed extra sum-of-squares *F*-tests to compare each variant to wild-type Egl–BicD.

Whereas the contacts of ED1, ED2 and the ExoHD–XAD module with the *TLS* are extensive, the interfaces between these protein domains are small. This suggests these interdomain contacts are insufficient to stabilize the assembly in the absence of RNA. Such RNA-dependent stabilization is consistent with our inability and that of others[35] to determine the structure of the Egl–BicD complex in isolation. Together, our observations support a cooperative mechanism for assembly of the RNA-binding pocket in which the localization signal reinforces otherwise weak interdomain interactions.

**Localization signals share structural and sequence features**
To address how Egl recognizes different localization signals, we additionally determined cryo-EM structures of Egl–BicD bound to the *hSL1*, *ILS*, *bcdSLV* or *GLS* RNA stem loops, which are required for dynein-based transport of the *hairy*, *I-factor*, *bicoid* and *gurken* mRNAs, respectively[17,18,24,37] (Supplementary Figs. 4–7 and Tables 1 and 2). These elements were chosen as they are predicted to differ from each other,

as well as the *TLS*, in length, loop size and the placement of bulged nucleotides. Contrary to the earlier hypothesis that Egl has distinct binding modes for different RNA targets[48], our structures show that the protein binds each localization signal in a similar manner. In all cases, ED1 and ED2 contact adjacent minor grooves with ExoHD and XAD enclosing the rest of the stem loop (Fig. 3a,b and Supplementary Fig. 8a). Additionally, the interactions among ED1, ED2 and XAD ensure that their RNA binding sites are positioned equivalently across all stem loops (Fig. 3b).

Our structures explain how considerable variability among the different stem loops is tolerated. Variation in loop size is permissible because the XAD interacts only with the portion of the loop adjacent to the upper helix. Bulges on the 5′ strand, which are present in the *ILS*, *bcdSLV* and *GLS*, are positioned away from Egl-binding sites and, therefore, do not interfere with recognition. Deviations in positioning of bulges on the 3′ strand are accommodated by pivoting of the ExoHD–XAD module around the XAD–ED2 interface, allowing

**Table 2 | Cryo-EM data collection and refinement statistics for structures of Egl–BicD bound to *ILS*, *bcdSLV*, *GLS* and *hSL1-hSL2***

| Sample | *ILS*-A | *ILS*-B | *bcdSLV*-A | *bcdSLV*-B | *bcdSLV*-C | *GLS* | *hSL1-hSL2* |
|---|---|---|---|---|---|---|---|
| **EM Data Bank** | EMD-54300 | EMD-54301 | EMD-54302 | EMD-54303 | EMD-54304 | EMD-54305 | EMD-54306 |
| **Protein Data Bank** | PDB 9RW6 | - | PDB 9RW7 | PDB 9RW8 | PDB 9RW9 | PDB 9RWA | PDB 9RWB |
| **Data collection and processing** | | | | | | | |
| Micrographs collected | 8,009 | | 22,750 | | | 12,592 | 44,979 |
| Magnification | ×81,000 | | x81,000 | | | ×81,000 | ×81,000 |
| Voltage | 300 | | 300 | | | 300 | 300 |
| Pixel size (Å) | 1.059 | | 1.059 | | | 1.059 | 1.059 |
| Electron exposure (e⁻ per Å²) | 47 | | 50 | | | 53 | 50 |
| Frames collected | 40 | | 50 | | | 56 | 50 |
| Defocus range (µm) | 0.5–3.5 | | 0.5–3.5 | 0.5–3.5 | 0.5–3.5 | 0.5–3.5 | 0.5–3.5 |
| Final particle images | 360,597 | 154,880b | 369,516 | 52,770 | 53,475 | 97,960 | 202,684 |
| Symmetry imposed | $C_1$ | $C_1$ | $C_1$ | $C_1$ | $C_1$ | $C_1$ | $C_1$ |
| Map resolution (Å) | 3.4 | 3.7 | 3.4 | 4.4 | 4.2 | 3.9 | 3.4 |
| FSC threshold | 0.143 | 0.143 | 0.143 | 0.143 | 0.143 | 0.143 | 0.143 |
| Sharpening *B* factor (Å²) | −60 | −100 | −142 | −145 | −100 | −112 | −100 |
| **Refinement** | | | | | | | |
| Model resolution (Å) | 3.5 | - | 3.6 | 7.4 | 7.2 | 4.2 | 3.5 |
| FSC threshold | 0.5 | - | 0.5 | 0.5 | 0.5 | 0.5 | 0.5 |
| Model composition | | | | | | | |
| Nonhydrogen atoms | 7,764 | - | 7,822 | 7,192 | 7,192 | 7,385 | 7,738 |
| Protein residues | 702 | - | 720 | 1,028 | 1,028 | 722 | 711 |
| Nucleotides | 116 | - | 100 | 100 | 100 | 94 | 98 |
| Mean *B* factors (Å²) | | | | | | | |
| Protein | 100.63 | - | 110.41 | 238.13 | 256.98 | 69.69 | 87.86 |
| Nucleotides | 120.81 | - | 101.99 | 236.51 | 156.07 | 45.98 | 122.39 |
| R.m.s.d. | | | | | | | |
| Bond lengths (Å) | 0.005 | - | 0.005 | 0.004 | 0.004 | 0.004 | 0.005 |
| Bond angles (°) | 0.591 | - | 0.657 | 0.943 | 0.955 | 0.728 | 1.034 |
| **Validation** | | | | | | | |
| Clashscore | 7.15 | - | 5.42 | 11.3 | 9.4 | 6.66 | 6.24 |
| MolProbity score | 1.73 | - | 1.29 | 1.86 | 1.79 | 1.55 | 1.52 |
| Rotamer outliers (%) | 0.19 | - | 0 | 0 | 0 | 0 | 0.17 |
| Ramachandran plot | | | | | | | |
| Favored (%) | 95.2 | - | 98.44 | 95.73 | 95.83 | 96.88 | 96.99 |
| Allowed (%) | 4.36 | - | 1.56 | 4.17 | 4.07 | 2.97 | 2.44 |
| Outliers (%) | 0.44 | - | 0 | 0.1 | 0.1 | 0.15 | 0.57 |

For each complex, the best-resolved conformation (structure A) was used for analysis (Supplementary Figs. 5–7 and 10).

ExoHD to shift by up to 15 Å while maintaining contacts with the RNA (Extended Data Fig. 5a).

Despite the structural variability between localization signals, there are common features that were not apparent from previous secondary-structure predictions (Supplementary Fig. 8b). Like the *TLS*, each RNA contains upper and lower A-form helices of at least 6 bp separated by a short stem segment with bulged nucleotides on the 3′ strand (Fig. 3b and Supplementary Tables 1 and 2). Additionally, all stem loops exhibit a bend of ~19°–26° between the two helical segments, resulting in a remarkably similar overall RNA backbone conformation (Fig. 3c and Extended Data Fig. 5b,c). The RNA bend appears to be important for Egl engagement, as modeling indicates that a straight helix

(a partially synthetic T7 phage RNaseIII substrate from PDB 2NUE)[49] can neither simultaneously contact ED1 and ED2 without disrupting their interdomain interface nor engage both ends of the rigid ExoHD–XAD module (Extended Data Fig. 5d). Consistent with these observations, MST measurements showed that the straight PDB 2NUE helix did not detectably associate with Egl–BicD (Extended Data Fig. 5e). Our structures suggest that the bend in localization signals is instigated by the bulged nucleotides on the 3′ strand (Fig. 3c). This provides a mechanistic explanation for previous data showing that the bulged nucleotides in the *TLS* are required for *K10* localization[50,51]. Supporting the general importance of a bent helix, we found that deleting bulges from each of the five localization signals strongly impaired Egl–BicD

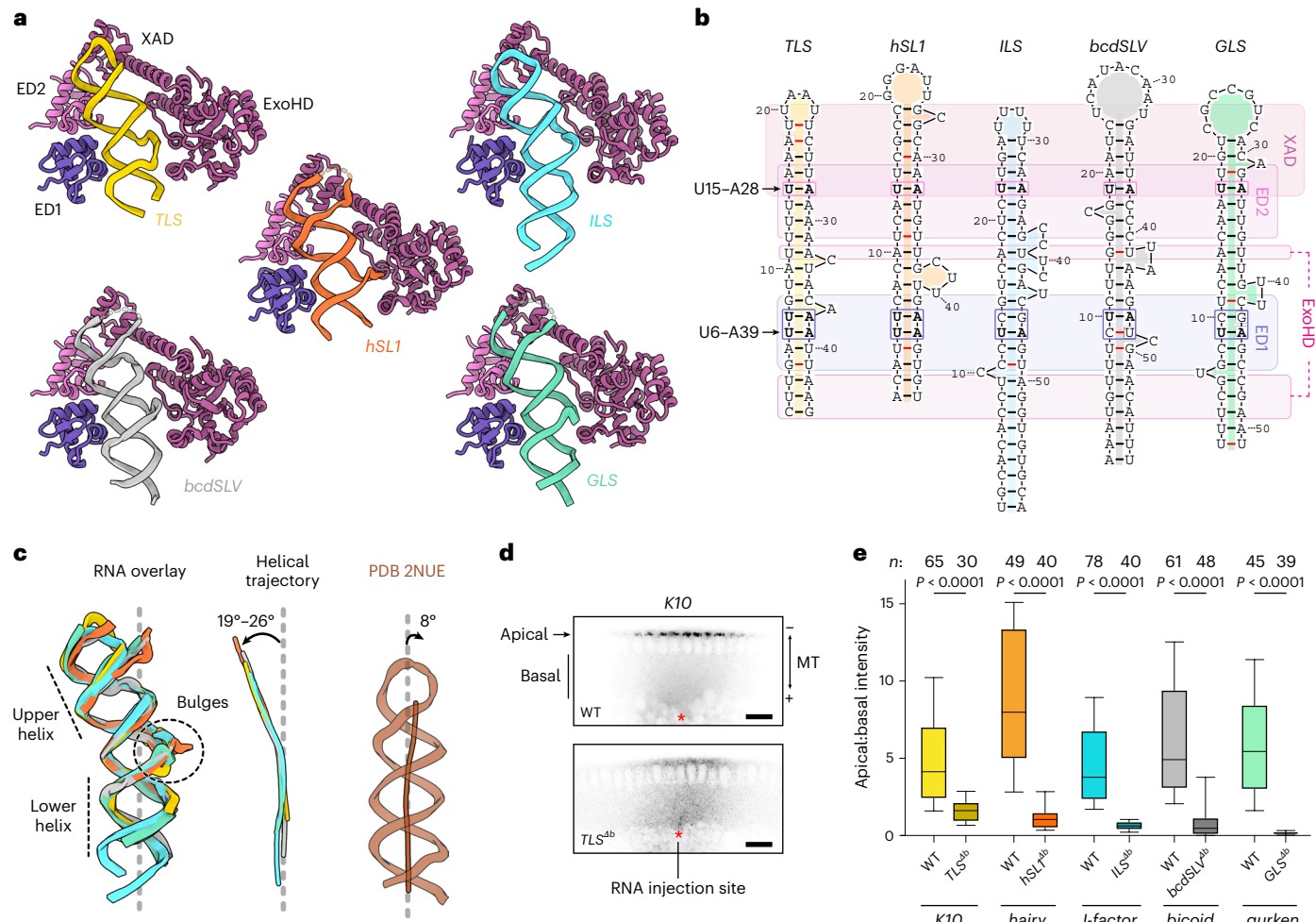

**Fig. 3 | Egl–BicD recognizes shared structural features of localization signals. a**, Cartoon models of Egl's composite RNA-binding pocket in complex with the indicated stem loops, as determined by cryo-EM. **b**, Empirically determined secondary structure of indicated localization signals in complex with Egl–BicD. Stem loops are aligned on the basis of the regions that interact with Egl's RNA-binding modules (rounded rectangles; purple and magenta differentiate the two Egl polypeptides in the complex). Noncanonical base pairs are indicated by red lines. Base pairs adjacent to ED1 residues His31 and Cys35 and ED2 residue Ser127 are boxed, with the U–A base pairs at these positions shown in bold. **c**, Superpositions of the stem loops (color-coded as in **b**) (left) and their helical trajectories (middle) compared to a canonical stem loop lacking bulged nucleotides (PDB 2NUE (ref. 49); right). **d**, Confocal images of *Drosophila* embryos injected with fluorescently labeled *K10* RNA or a mutant in which the bulges were deleted from the *TLS* (*TLS*$^{Δb}$). Microtubule (MT) polarity and apical and basal regions are indicated. The red asterisk represents the RNA injection site. Scale bar, 20 μm. **e**, Localization efficiency (apical:basal intensity) following injection into *Drosophila* embryos of indicated wild-type fluorescent RNAs and variants lacking bulges in localization signals. Boxes show the interquartile range, with the horizontal line denoting the median, and whiskers show the 10th–90th percentile range. A total of 30–78 embryos were analyzed from two to six independent injections for each transcript. *P* values were determined using pairwise two-tailed Mann–Whitney tests.

binding in vitro (Extended Data Fig. 6a–e), as well as dynein-driven apical localization of mRNAs injected into *Drosophila* embryos (Fig. 3d,e and Extended Data Fig. 6f–j).

In addition to a common RNA backbone shape, all five localization signals exhibit similarities in how their minor grooves bind ED1 and ED2. While there is considerable variation in the neighboring RNA sequences, the base pair next to Ser127 of ED2 (U15–A28 in the *TLS*; Fig. 2c) is a U–A in all structures (Fig. 3b). The base pairs in proximity to residues His31 and Cys35 of ED1 (equivalent to U6–A39 and U7–A38 in the *TLS*; Fig. 2c) are more variable but are also frequently U–As, especially in the position equivalent to U6–A39 (Fig. 3b). These observations raised the possibility that sequence identity at these two sites contributes to selective RNA recognition. To test this notion, we introduced mutations of U–A to G–C in the *TLS* at the ED1 (2×GC$^{ED1}$) or ED2 (GC$^{ED2}$) interaction sites (Fig. 4a). We found that each mutation diminished Egl–BicD binding in vitro (Fig. 4b) and disrupted *K10* localization in the embryo (Fig. 4c,d), while combining them in the same stem loop had even stronger deleterious effects (Fig. 4b–d).

To determine which base-pair combinations can be distinguished by Egl, we performed further mutagenesis on the most invariant base pairs within the ED1 and ED2 interaction sites of the *TLS*: namely U6–A39 and U15–A28 (Fig. 4e). Substitution of these U–A pairs to G–C (GC$^{ED1/ED2}$), C–G (CG$^{ED1/ED2}$) or A–U (AU$^{ED1/ED2}$) resulted in a significant decrease in affinity for Egl–BicD (Fig. 4e,f). Among these variants, C–G substitutions had the smallest effect, followed by A–U and then G–C, which strongly perturbed binding (Fig. 4f). These data reveal that U–A base pairs at the ED1 and ED2 sites of the *TLS* promote high-affinity association with Egl. Consistent with this observation, U–As in these positions are evolutionarily conserved (Supplementary Fig. 9).

To elucidate how Egl differentiates U–As from other base pairs, we returned to the Egl–BicD–*TLS* structure. At the ED1 site, the sulfur of Cys35 sits in a shallow cleft between the O2 atom of U6 and the C2 atom of A39, whereas His31 is positioned to make a hydrogen bond with the N3 atom of A39 (Fig. 4g and Extended Data Fig. 7a). The ED2 site contains electron density that likely corresponds to an ion or water molecule coordinated by Ser127 and the backbone

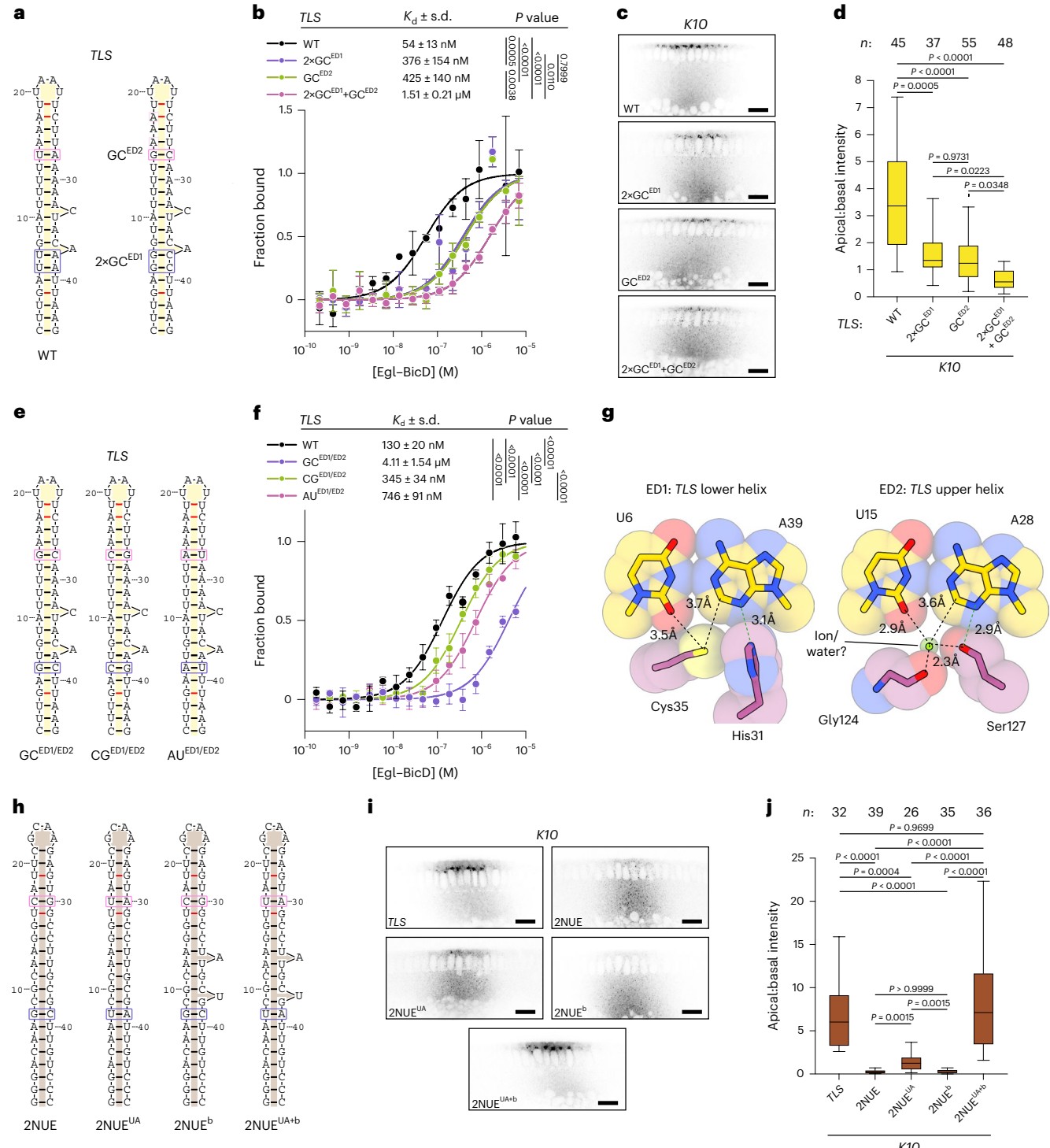

**Fig. 4 | Egl–BicD discriminates sequences at the ED1 and ED2 sites. a**, Empirical secondary structure of wild-type *TLS* (left) and positions of indicated base-pair substitutions in regions adjacent to ED1 and ED2 (boxed). **b**, MST binding curves for Egl–BicD with the *TLS* and the indicated mutations. **c,d**, Confocal images (**c**) and localization efficiency (**d**) of fluorescently labeled *K10* 3′ UTR with the indicated *TLS* variants injected into *Drosophila* embryos. Scale bar, 20 μm. **e**, Secondary structures of the *TLS* with indicated substitutions of the base pairs that interact with ED1 and ED2 (boxed). **f**, MST binding curves for Egl–BicD with the *TLS* and the indicated base-pair mutations. **g**, Atomic models for the U6–A39 (left) and U15–A28 (right) *TLS* base pairs and the adjacent Egl side chains at the ED1 and ED2 minor groove interaction sites, respectively. Base pairs are displayed in stick representation overlaid with space-filling spheres scaled according to van der Waals radii. Dotted green lines indicate hydrogen bonds. **h**, Secondary structure of the straight RNA stem loop from PDB 2NUE with indicated mutations (boxed). **i,j**, Confocal images (**i**) and localization efficiency (**j**) of fluorescently labeled *K10* 3′ UTR with the indicated PDB 2NUE variants replacing the *TLS* in injected *Drosophila* embryos. Scale bar, 20 μm. In **b,f**, data points show the mean ± s.d. for three independent experiments per condition, from which best-fit values for $K_d$ ± s.d. were derived. In **d,j**, boxes show the interquartile range, with the horizontal line denoting the median, and whiskers show the 10th–90th percentile range. A total of 26–55 embryos were analyzed from two independent injections for each transcript. *P* values were determined using a one-tailed extra sum-of-squares *F*-test (**b,f**) or a one-way Brown–Forsythe and Welch analysis of variance (ANOVA) test with Dunnett's T3 test for multiple comparisons (**d,j**).

carbonyl of Gly124. This density occupies an equivalent position to the sulfur of Cys35 at ED1 within the shallow cleft between U15 and A28 (Fig. 4g and Extended Data Fig. 7b). The ion or water is also positioned such that it can interact with the O2 atom of U15, further stabilizing the ED2–RNA interface. Therefore, at both sites, there is complementarity between the recognition surface of the ED and the U–A base pair.

We next modeled replacements of the aforementioned U–As of the *TLS* with all other canonical base-pair combinations. At ED1, mutation to A–U is expected to result in a clash between the adenine C2 atom and the sulfur group of Cys35 (Extended Data Fig. 7a). In the case of a C–G or G–C substitution at this site, the guanine N2 amino group would clash with Cys35 (Extended Data Fig. 7a). At the ED2 site, an A–U substitution likely repositions the adenine C2 atom closer to the ion or water, either clashing with it or causing its displacement (Extended Data Fig. 7b). Lastly, C–G or G–C mutations at ED2 introduce a clash between the guanine N2 amino group and both the bound ion or water and Ser127 (Extended Data Fig. 7b). Thus, our *TLS* structure suggests that a steric discrimination mechanism has a role in the recognition of U–A base pairs by both ED1 and ED2.

The structures of the four other localization signals indicate that this mode of minor groove recognition applies more generally. Firstly, all of the structures have U–A base pairs adjacent to His31 and Cys35 in ED1, with the exception of *bcdSLV* (Fig. 3b and Extended Data Fig. 7c). In this case, a conserved, noncanonical C–U pair (Fig. 3b and Supplementary Fig. 9) is tolerated, presumably because there is no bulky purine to clash with Cys35 (Extended Data Fig. 7c). Secondly, in those structures resolved to better than 3.5 Å (all except *GLS*-bound Egl–BicD), there appears to be density for an ion or water molecule next to the U–A base pair at the ED2 site, which is in a position analogous to that observed in the *TLS* (Extended Data Fig. 7d).

Collectively, our findings indicate that localization signals require a kinked dsRNA stem loop induced by 3′ bulges, together with base-pair identities at two sites separated by 8 bp. To test whether these features are sufficient to define a localization signal, we injected *Drosophila* embryos with a *K10* 3′ untranslated region (UTR) in which the *TLS* was replaced with variants of the straight PDB 2NUE stem loop (Fig. 4h). As expected, replacing the *TLS* with PDB 2NUE in this construct completely abolished apical localization (Fig. 4i,j). In this context, inclusion of only the spaced U–A base pairs (PDB 2NUE$^{UA}$) or only the 3′ bulges (PDB 2NUE$^b$) resulted in marginal or no improvement in localization, respectively (Fig. 4i,j). However, the combination of both features (PDB 2NUE$^{UA+b}$) restored localization to wild-type levels (Fig. 4i,j). These findings reveal how a set of simple structural and sequence features can be integrated to define a localization signal.

## Two RNA stem loops are required for efficient transport

Previous studies showed that single copies of mRNA molecules associate with Egl–BicD, leading to a model in which one localization signal is sufficient for dynein activation[34,35]. However, in our structures, Egl–BicD always binds two stem loops. To reconcile these seemingly contradictory results, we reconstituted motile RNA–protein complexes by assembling dynein, dynactin, BicD and Egl with substoichiometric amounts of Cy3-labeled or Cy5-labeled *TLS* RNAs and visualized their movement along microtubules in vitro. If single *TLS* molecules were sufficient for motility, we would detect no colocalization of Cy3 and Cy5 signals in motile complexes (Fig. 5a). In contrast, if two stem loops were needed, half the complexes would contain both fluorophores (Fig. 5a). We found that 44% of motile dynein complexes contained both Cy3 and Cy5, revealing the presence of two *TLS* molecules in the vast majority of cases (Fig. 5b,c). This indicates that initiation of transport is associated with the assembly of Egl dimers around two separate stem-loop elements. Nonetheless, in agreement with the results of previous studies[35], performing the assay with the full-length *K10* revealed only 6% colocalization and, therefore, a single copy of the RNA in most complexes (Fig. 5b,c).

These results could be explained by the full-length *K10* mRNA containing a second, previously unidentified RNA element that binds Egl–BicD along with the *TLS*. To test whether *K10* contains a second element that is important for localization, we injected a series of *TLS*-bearing truncations of the RNA into *Drosophila* embryos (Fig. 5d and Extended Data Fig. 8a). These experiments identified a 54-nt region of the 3′ UTR that is required for efficient apical localization and contains a 40-nt stem loop (Fig. 5d,e and Extended Data Fig. 8a). We refer to this structure as the *K10* localization support element (*KSE*). While the *KSE* could not promote apical transport of a heterologous *LacZ* RNA on its own, it significantly enhanced *TLS*-mediated localization of this transcript (Fig. 5f and Extended Data Fig. 8b). MST revealed that the *KSE* binds Egl–BicD, albeit with lower affinity than the *TLS* ($K_d$ = 507 nM versus 80 nM; Extended Data Fig. 8c). We also found using the dual-color motility assay that combining the *TLS* and the *KSE* in a minimal 145-nt RNA resulted in activation of dynein by a single RNA (Fig. 5g). Therefore, despite the lower independent affinity of the *KSE*, Egl–BicD shows a preference for binding both elements in a single RNA over two *TLS* elements in separate RNAs. This suggests that avidity conferred by the presence of the *TLS* and the *KSE* on the same transcript enhances Egl–BicD association.

The requirement for two RNA elements to activate dynein could explain the reported multipartite nature of certain other localization elements[18,37,52–54]. To test this possibility, we turned our attention to the 126-nt *hairy* localization element[18], which consists of *hSL1* and a second stem loop, *hSL2*. The latter element is not active in isolation

**Fig. 5 | Cis-acting support elements promote RNA localization. a**, Expected occurrence of fluorescence signals for hypothetical binding of one or two copies of RNA to Egl–BicD in transport complexes reconstituted in the presence of a 50:50 mixture of Cy3-labeled or Cy5-labeled RNA. **b**, Example kymographs (time–distance plots) from transport assays with a tenfold molar deficit (relative to Egl) of mixtures of the indicated Cy3-labeled or Cy5-labeled RNAs. **c**, Observed occurrence of fluorescence signals in motile complexes assembled with the indicated RNAs. **d**, Truncations of the *K10* 3′ UTR and their localization efficiencies when injected into *Drosophila* embryos. For the intact *K10* 3′ UTR (1608–3061), data are reproduced from Fig. 3e. **e**, RNAfold-predicted secondary structure of the *KSE* stem loop within the region that supports localization of the *K10* 3′ UTR. **f**, Localization efficiency of *LacZ* transcripts bearing different combinations of the *TLS* and *KSE* following injection into *Drosophila* embryos. **g,h**, Observed occurrence of fluorescence signals in motile complexes assembled with *TLS-KSE* (**g**) and *hSL1* or *hSL1-hSL2* (**h**) RNAs. **i**, Empirically determined secondary structure of *hSL1* and *hSL2* of the *hairy* localization element in complex with Egl–BicD. Stem loops are aligned based on the regions that interact with ED1 and ED2 (rounded rectangles; purple and magenta differentiate the two Egl polypeptides in the complex). Noncanonical base pairs are indicated by red lines. Base pairs adjacent to ED1 residues His31 and Cys35 and ED2 residue Ser127 are boxed, with the U–A base pairs at these positions shown in bold. **j**, Top, cryo-EM structure of Egl–BicD bound to *hSL1* and *hSL2*. The unresolved linker between *hSL1* and *hSL2* is depicted as a dotted line. Bottom, His31 of ED1 and Ser127 of ED2 contact U84–A129 and U96–A118 base pairs, respectively. Dotted green lines denote hydrogen bonds. **k**, Top, superposition of *hSL1* and *hSL2* stem-loop structures. Bottom, ExoHD–XAD modeled at the *hSL2*-binding site showing clashes with the *hSL2* stem loop. The ExoHD–XAD from the *hSL1*-binding pocket was overlaid onto the *hSL2*-binding site by aligning the two binding pockets using their respective ED2s. In **c,g,h**, the mean cumulative percentage ± s.d. from three or four independent experiments (black dots) is shown. A total of 235–1,032 motile complexes were analyzed per transcript. In **d,f**, boxes show the interquartile range, with the horizontal line denoting the median, and whiskers show the 10th–90th percentile range. A total of 22–85 embryos were analyzed from two to six independent injections for each transcript. *P* values were determined using a one-way Brown–Forsythe and Welch ANOVA test with Dunnett's T3. In **d**, the localization efficiency of each truncation was compared to that of the full-length *K10* 3′ UTR.

but, through an unknown mechanism, stimulates apical localization driven by *hSL1* (ref. 18). As with the *KSE* and *TLS*, we found that Egl–BicD associated with *hSL2* but with a lower affinity than observed for *hSL1* ($K_d$ = 603 nM versus 7 nM; Extended Data Fig. 8d). Moreover, dual-color motility assays revealed that, while two copies of an isolated *hSL1* were typically needed to activate dynein, an RNA containing both *hSL1* and *hSL2* was much more likely to be transported as a single copy (Fig. 5h) and was more efficient at initiating motility (Extended Data Fig. 8e). These observations indicate that *hSL2* serves an equivalent function to the *KSE* during activation of the motor by cooperating with a primary

signal through occupation of two RNA-binding sites within Egl–BicD. Therefore, our data reveal that the presence of two Egl-binding stem loops on the same RNA is another criterion by which mRNAs are selected for localization.

## The structural basis of dual-stem-loop recognition

To understand how the combination of a primary localization signal and support element is recognized, we determined a cryo-EM structure of Egl–BicD bound to the *hairy* localization element (Fig. 5i,j, Supplementary Fig. 10 and Table 2). This revealed that *hSL1*

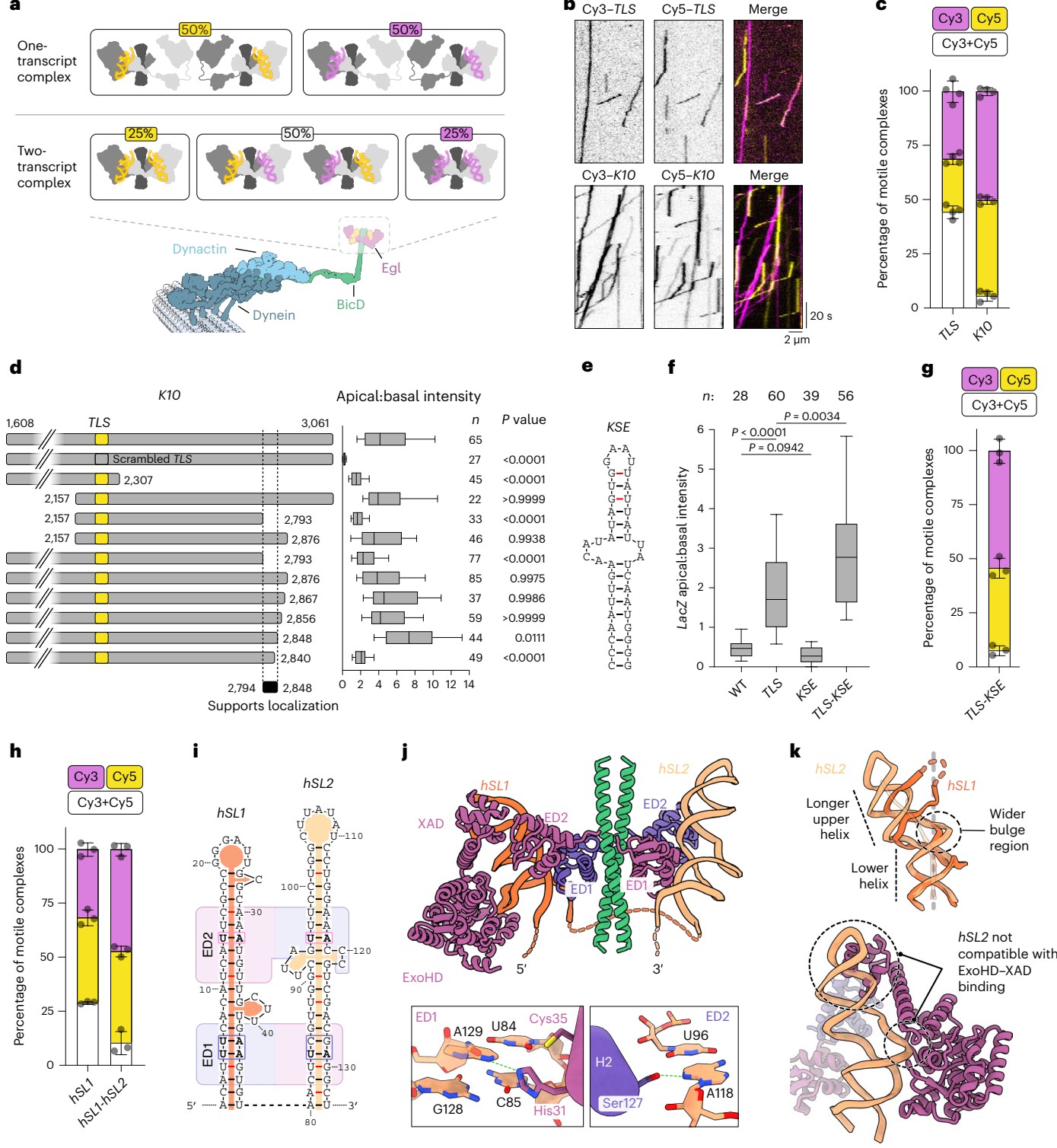

is bound by ED1, ED2, ExoHD and XAD (Fig. 5j) in the same manner as in the structure with this stem loop in isolation (Fig. 3a). In contrast, on the opposite side of BicD, *hSL2* only engages ED1 and ED2 (Fig. 5j). The interaction of *hSL2* with these domains is through a bent helical backbone and two sets of U–As separated by 8 bp (Fig. 5i–k, Extended Data Fig. 9a and Supplementary Fig. 11), as typically observed for primary localization signals. In contrast to the primary signals, however, *hSL2* contains additional unpaired nucleotides on the 5′ strand that expand the bulge region, resulting in a larger bending angle and a shift in the position of the upper helix relative to that of *hSL1* (Fig. 5k and Extended Data Fig. 9b,c). These structural deviations are accommodated by pivoting of ED2 around its interaction site with ED1, which strains the ED1–ED2 interface but still maintains interactions of the domains with the RNA (Extended Data Fig. 9d). Additionally, *hSL2* has a longer upper helix than *hSL1*, which, coupled to the wider bulge region, sterically prevents docking of ExoHD–XAD (Fig. 5k). The absence of ExoHD–XAD in the structure, as well as the suboptimal arrangement of ED1–ED2, likely explains the relatively weak binding of the isolated *hSL2* to the Egl–BicD complex (Extended Data Fig. 8d). Overall, this structure reveals how a single transcript engages Egl–BicD through simultaneous association of high-affinity and low-affinity RNA elements.

## Discussion

Our current understanding of RNA recognition largely derives from studies of proteins that target specific sequence motifs or conserved secondary structures[55,56]. However, the mechanisms by which RBPs identify RNA elements lacking such well-defined patterns remain poorly understood. Here, we determined structures of Egl bound to five different localization signals, as well as a bipartite RNA that contains a localization signal and a support element. Coupled to in vitro and in vivo functional analyses, this work reveals multiple layers of recognition that govern specific engagement in the absence of obvious conservation of sequence or secondary structure. Egl recognizes a bent dsRNA stem-loop structure that can be formed by diverse sequences containing bulges of varying number and size. In conjunction, Egl engages two distant base-pair positions within the RNA minor groove, selecting for U–A base pairs, while remaining compatible with noncanonical pyrimidine–pyrimidine pairs such as C–U. Specificity is further increased by the coincident recognition of two stem-loop elements within the same transcript by Egl dimers. Together, tertiary RNA structure recognition, minor groove base discrimination and dual-element detection provide a versatile yet robust strategy for selective recognition of diverse mRNA cargoes. The identification of these recognition principles paves the way for revealing novel mRNA targets for Egl and elucidating the functional consequences of their localization.

A previous NMR structure of the isolated *TLS* suggested that an A′-form dsRNA conformation contributes to mRNA localization[51]. Although we did not observe this feature in any of the localization signals in complex with Egl–BicD, the isolated *TLS* exhibits a helical bend that is indistinguishable from that in our structure and, therefore, requires only minor adjustments to adopt the Egl-bound state (root-mean-square deviation (r.m.s.d.) = 3.8 Å; Extended Data Fig. 9e,f). This contrasts with another structurally characterized RBP–localization signal complex[57], where the yeast *ASH1* mRNA element displays a large conformational switch between free and She2p–She3p-bound states (r.m.s.d. = 16.3 Å; Extended Data Fig. 9g). These comparisons suggest that localization signals need not be highly dynamic as previously proposed[57] but can instead preexist in a stable, recognition-competent conformation. Egl-based RNA recognition instead depends on the conformational flexibility of the protein itself. The RNA-binding domains of Egl can pivot about their interdomain interfaces to accommodate variations in loop structure, bulged nucleotides and even the increased stem-loop bend found in the *hSL2* support element, thereby enabling engagement of multiple RNA targets.

Comparison of our Egl–BicD–RNA structures revealed sequence determinants within the different localization signals that were not apparent previously. We found that U–A base pairs are favored at the sites where ED1 and ED2 bind. These sequence features were likely overlooked in earlier studies because they can be separated by a variable number of bases in the primary sequence due to Egl's ability to tolerate bulges of different sizes within a stem-loop structure. A key question is how this discrimination occurs through the RNA minor groove. Guanine-containing base pairs appear to be disfavored because of steric exclusion of the N2 amino group, as has been observed previously for dsRBD2 of the ADAR2 protein[58,59]. However, Egl achieves finer discrimination than ADAR2, as it can also distinguish U–A from A–U base pairs. This appears to be related to the position of the shallow cleft between the bases in the U–A pair, which is distinctly complementary to the ED1 and ED2 recognition surfaces. While such shape-based selectivity has been proposed to help polyamides discriminate between T–A and A–T base pairs in the DNA minor groove[60], to our knowledge, it has not previously been implicated in protein interactions with either DNA or RNA minor grooves.

In addition to these steric-based mechanisms, our data point to an additional layer of discrimination. Our binding assays with Egl–BicD and *TLS* variants showed that C–G is better tolerated than either A–U or G–C. This indicates a preference for a 5′ pyrimidine–3′ purine base-pair configuration. Such a bias could also contribute to U–A base pairs being favored at the ED sites. How can Egl read the configuration of pyrimidines and purines within the helix? One possibility is that it stems from features that are not resolved in our structures, such as subtle differences in hydrogen-bonding geometry that may favor interactions with 3′ purines over 3′ pyrimidines. Alternatively, the 5′ base may be more constrained by the surrounding interactions of Egl with the RNA backbone, meaning that a bulky purine in this position is less able to escape steric clashes. Consistent with this notion, the EDs and XAD make more backbone interactions with the 5′ strand than the 3′ strand of localization signals (Extended Data Fig. 10a,b and Supplementary Fig. 8). Taken together, our findings provide insights into how an RBP can read minor groove sequences through multiple, layered determinants.

Another key insight from our study is that the binding of Egl–BicD to two stem loops is a prerequisite for activation of RNA transport. We show that, for *K10* and *hairy*, this requirement is fulfilled by the combination of a high-affinity primary localization signal and a lower-affinity support element within the same transcript. The auxiliary role of the support elements is aligned with them being less strongly conserved in evolution than the primary signals (Supplementary Fig. 9). The use of two stem loops enhances specificity and localization fidelity by ensuring that motor activation is coupled to coincident detection of two separate elements by Egl. Within this context, the lower-affinity element would be easier to remove from the Egl dimer, thereby destabilizing the complex and enabling the efficient recycling of the protein components for transport of other mRNAs. We further found that different primary localization signals engage Egl–BicD with varying affinities (Extended Data Fig. 6a–e). This may allow mRNA species to modulate the onset and duration of their interaction with the transport machinery, balancing efficient delivery with timely release to ensure appropriate spatial and temporal distribution of their protein products.

Several other Egl target mRNAs, such as *wingless*[52], *fushi tarazu*[37], *oskar*[53,54] and *bicoid*[37], require regions beyond a single stem loop for proper localization. This observation suggests that occupancy of Egl–BicD by two elements within the same RNA is a widespread mechanism for cargo discrimination. Intriguingly, some of these elements are separated by several hundred nucleotides in the primary sequence. Such distant elements may be held in close proximity because of higher-order RNA folding. Alternatively, Egl binding to the two elements could lead to rearrangements of the RNA, potentially exposing target sites for other RBPs that coordinate transport with regulatory processes

such as mRNA decay or translation. It is also conceivable that some RNAs lack additional support elements and compensate by forming oligomeric structures[61–63] that present multiple primary localization signals for Egl–BicD engagement, thereby allowing selective transport of higher-order RNA assemblies.

Recognition of tens to hundreds of distinct mRNAs has been demonstrated for other RBPs[64]. These include factors that orchestrate mRNA localization, such as Staufen2 (refs. 20,65), FMRP[21,66], APC[25,67], She2p–She3p[15,57] and the recently characterized FERRY complex[26,68]. Similar to Egl, many of these RBPs contain multiple RNA-binding domains and/or form dimeric assemblies and, therefore, have the potential to recognize diverse RNA elements using multiple selection criteria. However, because identifying RNA recognition motifs has remained challenging, several of these proteins have been suggested to bind their targets through nonspecific binding strategies such as assembling into RNA granules or biomolecular condensates[9,10,69–71]. While these models remain plausible, our visualization of Egl–BicD bound to many targets has revealed that conserved recognition features can exist even among RNAs that differ substantially in sequence and secondary structure. This raises the possibility that mRNAs targeted by other RBPs also contain currently elusive patterns that support their selective recognition.

## Online content

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

## Methods

### Recombinant protein expression

The *Drosophila* Egl–BicD and human dynein complexes were expressed recombinantly from baculovirus in *Sf*9 insect cells (Thermo Fisher Scientific; derived from the ovarian tissue of *Spodoptera frugiperda* fall armyworm) using a polycistronic MultiBac system[73], as described previously[34]. A codon-optimized Egl coding sequence (isoform B; NM_166623) with a C-terminal TEV–ZZ affinity tag was synthesized (Epoch Life Sciences) and cloned into the pACEBac1 acceptor plasmid and Cre-recombined with a pIDC donor plasmid containing a codon-optimized coding sequence for BicD (NM_165220; Epoch Life Sciences). The recombined plasmid was incorporated into the baculovirus genome by transformation of DH10EMBacY cells. This genome was purified and used with FuGENE HD (Promega) to transfect adherent *Sf*9 cells. After ~96 h at 27 °C, transfection or infection of the majority of *Sf*9 cells was confirmed by YFP detection. The baculovirus was then amplified by using the supernatant from the transfection to infect a 50-ml suspension culture of *Sf*9 cells for ~96 h at 27 °C. The supernatant of this culture was used to infect 500-ml suspension cultures for final protein expression. Cells from these cultures were pelleted, frozen in liquid nitrogen and stored at −80 °C until processed for protein purification. All Egl variants were derived from the pACEBac1 construct described above by site-directed mutagenesis (Egl$^{ED1mut}$; Cys35Tyr) or by Gibson assembly (New England Biolabs) of PCR amplicons that excluded the targeted coding sequence (Egl$^{ΔED2}$: deletion of residues 94–167, Egl$^{ΔED3}$: deletion of residues 187–259, Egl$^{ΔXAD}$: deletion of residues 744–816) with the exception of Egl$^{Exomut}$ (Lys657Ala, Lys659Ala, Arg686Ala and Arg687Ala), which was commercially synthesized (Epoch Life Sciences) and cloned into pACEBac1. The instability of Egl–BicD lacking the ExoHD resulted in insufficient protein concentration for use. Deletion of ED1 was not pursued as it is expected to compromise Egl stability by eliminating BicD association[34]. The human dynein complex was expressed analogously to Egl–BicD, with sequences encoding dynein 1 heavy chain (NM_001376.4) bearing an N-terminal ZZ–TEV–SNAPf tag cloned into pACEBac1 and the remaining subunits (dynein intermediate chain 2 (DIC2: AF134477), dynein light intermediate chain 2 (DLIC2: NM_006141.2) and the light chains (Tctex: NM_006519.2, LC8: NM_003746.2, Robl: NM_014183.3)) cloned into pIDC[74].

### Protein purification

Egl–BicD complexes and dynein were produced as described previously[34], with purification steps at 4 °C. *Sf*9 cells were routinely confirmed as free of *Mycoplasma* using the MycoALERT kit (Lonza). Cells expressing recombinant complexes were suspended in lysis buffer (50 mM HEPES pH 7.3, 500 mM NaCl (100 mM NaCl for dynein), 10% glycerol, 1 mM DTT, 0.1 mM Mg-ATP, 2 mM PMSF and 1× cOmplete EDTA-free protease inhibitor cocktail (Roche)) and lysed by passage in a Dounce homogenizer with a tight pestle. The lysate was clarified by ultracentrifugation in a Type 70 Ti rotor (Beckman-Coulter) at ~500,000$g$, applied to prewashed (in lysis buffer) IgG–Sepharose FF resin (Cytiva) in a gravity-flow Econo column (Bio-Rad) and incubated with gentle rolling for 3 h. Flowthrough was collected by gravity and the protein-bound resin was washed twice with five column volumes of lysis buffer and twice with five column volumes of TEV buffer (50 mM Tris pH 7.4, 150 mM potassium acetate, 2 mM magnesium acetate, 1 mM EGTA–KOH pH 7.5 and 10% glycerol). The resin was then transferred to a 15-ml conical tube in a 15-ml final volume of TEV buffer and incubated overnight at 4 °C with TEV protease to cleave the protein complexes from the beads and ZZ affinity tag. Liberated protein complexes were collected by gravity flow from a fresh Econo column and concentrated to ~500 µl (Egl–BicD) or ~300 µl (dynein) using Amicon Ultra-4 (100-kDa molecular weight cutoff (MWCO)) concentrator units (Merck) before application to Superose 6 Increase 10/300 (Egl–BicD; Cytiva) or TSK-gel G4000SWxl with guard (dynein; TOSOH Bioscience) gel-filtration columns run in GF150 buffer (25 mM HEPES pH 7.3, 150 mM KCl, 1 mM

MgCl$_2$, 0.1 mM Mg-ATP, 5 mM DTT and 10% glycerol) on an AKTA Purifier fast protein liquid chromatography (FPLC) instrument (Cytiva). Fractions containing the protein complexes were pooled and concentrated as above to final concentrations of ~1.5 mg ml$^{-1}$ (Egl–BicD and dynein for single-molecule motility assays) or 4–5 mg ml$^{-1}$ (Egl–BicD for MST assays), as determined with Bradford reagent (Pierce).

Native dynactin was purified at 4 °C from pig brain extracts, as described previously[74]. Three pig brains were homogenized in lysis buffer (35 mM PIPES–KOH pH 7.2, 5 mM MgSO$_4$, 1 mM EGTA–KOH pH 7.5, 0.5 mM EDTA pH 7.4, 1 mM DTT, 2 mM PMSF and 1× cOmplete EDTA-free protease inhibitor cocktail (Roche)) using short bursts in a blender. The lysate was clarified at low speed at 38,400$g$ in a JLA 16.250 rotor (Beckman-Coulter) and then at higher speed by ultra-centrifugation at ~160,000$g$. The clarified lysate was filtered through a 0.45-µm-syringe-tip filter (Fisher) and loaded onto an SP-Sepharose (Cytiva) cation-exchange column (~250 ml bed volume) equilibrated in SP-buffer A (35 mM PIPES–KOH pH 7.2, 5 mM MgSO$_4$, 1 mM EGTA–KOH pH 7.5, 0.5 mM EDTA pH 7.4, 1 mM DTT and 0.1 mM Mg-ATP) and run on AKTA Pure FPLC instrumentation. Lysate was fractionated by washing with six column volumes of buffer A and eluting with a two-phase linear salt gradient with SP-buffer B (SP-buffer A with 1 M KCl), in which the first phase increased the KCl concentration to 250 mM over three column volumes and the second phase further increased the KCl concentration to 1 M over an additional one column volume. Dynactin-containing fractions were pooled and diluted twofold with Q-buffer A (35 mM PIPES–KOH pH 7.2, 5 mM MgSO$_4$, 1 mM EGTA–KOH pH 7.5, 0.5 mM EDTA pH 7.4 and 1 mM DTT) and filtered through a 0.22-µm-syringe-tip filter (Fisher) before loading onto a MonoQ 16/10 column (Cytiva) equilibrated in Q-buffer A and running on AKTA Pure FPLC instrumentation. After loading, the MonoQ 16/10 column was washed with five column volumes of Q-Buffer A and the bound protein eluted in a three-phase linear salt gradient with Q-buffer B (Q-buffer A with 1 M KCl), in which the first phase increased the KCl concentration to 150 mM over one column volume, the second phase increased the KCl concentration to 350 mM over ten column volumes and the third phase increased the KCl concentration to 1 M over one column volume. Fractions containing dynactin were pooled and concentrated in an Amicon Ultra-4 (100-kDa MWCO) concentrator unit (Merck) to a final volume of 300–500 µl before application to a TSKgel G4000SWxl with guard (TOSOH Bioscience) gel-filtration column running in GF150 buffer on AKTA Purifier FPLC instrumentation. Fractions containing dynactin were pooled and concentrated as above to a final concentration of 1–2 mg ml$^{-1}$, as determined with Bradford reagent. All purified proteins were dispensed into single-use aliquots, flash-frozen in liquid nitrogen and stored at −80 °C.

### RNA templates, synthesis and purification

RNA constructs for cryo-EM and MST were synthesized commercially (Horizon Discovery; see Supplementary Table 3) with the exception of the 151-nt RNA containing the *hairy* localization element, which was transcribed in vitro from a *hairy* 3′ UTR template[18] (GenBank X15905), as described below. The *TLS* and *hSL1* constructs (as well as their mutants) included 5′ and 3′ flanks of 8 nt (*TLS*) or 10 nt (*hSL1*) of the native sequence from *K10* or *hairy* 3′ UTRs. RNAs for MST included a 5′-Cy5 label attached to stem loops by a linker of two adenosine nucleotides (with the exception of the *TLS* and *hSL1* whose native 5′ flanking sequence served as the linker). For single-molecule motility assays with the *TLS* and *hSL1*, Cy3-labeled versions of these transcripts were used in addition to Cy5-labeled constructs.

*K10* 3′ UTR constructs for single-molecule assays and embryo injection were transcribed from and numbered according to the *K10* cDNA (GenBank AY060415). The template for the 1,490-nt 3′ UTR construct (positions 1608–3061 of the cDNA) contained the poly(A) signal from the cDNA and an additional 38 bp that included an SP6 promoter from the backbone that was not counted in the numbering.

*TLS* mutations within the *K10* 3′ UTR were introduced using a construct with engineered HindIII and NheI sites on either side of the *TLS* that was previously shown not to affect localization of mRNA in embryos[51]. Synthetic DNA oligos (Merck) encoding the desired *TLS* mutations and complementary to the HindIII and NheI overhangs were ligated to the digested plasmid containing the *K10* 3′ UTR, thereby replacing the encoded *TLS* with the mutant sequence. The scrambled *TLS* is a nonlocalizing control incapable of making a stem-loop structure[51]. The *hairy* 3′ UTR corresponds to positions 1185–1845 of GenBank X15905 and is part of a 730-nt transcript used for injection. Templates of a mutant version of the *hairy* 3′ UTR in which *hSL1* lacks bulges were synthesized commercially (GenScript). The 573-nt fragments of the *I-factor* RNA coding sequence containing the wild-type *ILS* or a version lacking bulges were transcribed from commercially synthesized template (Integrated DNA Technologies) and correspond to positions 2932–3498 of GenBank M14954.2. The 839-nt *bicoid* 3′ UTR RNA containing wild-type *bcdSLV* or a version lacking bulges was transcribed from a commercially synthesized template (GenScript) and corresponds to positions 1702–2536 of GenBank NM_169159.4. Full-length 1,718-nt *gurken* transcripts containing the wild-type *GLS* or a version lacking bulges were transcribed from commercially synthesized templates (GenScript) and correspond to positions 1–1718 of GenBank NM_057220.3.

*LacZ* RNAs with individual *TLS* and *KSE* elements or a combination of both were transcribed from a fragment of the *LacZ* gene that was comparable in length to the *K10* 3′ UTR and predicted to code for an RNA with minimal secondary structure, as determined by the greatest average ss-count in mfold predictions[75] (https://www.unafold.org/mfold/applications/rna-folding-form.php). At a position analogous to the location of the *TLS* in the *K10* 3′ UTR (673 bp), the *TLS* and *KSE* sequences were added to the design and commercially synthesized (GenScript). For the transcript containing both the *TLS* and *KSE*, the linker between the stem loops was the native sequence between *hSL1* and *hSL2* (ref. 18). This template was also used to transcribe the 145-nt *TLS-KSE* construct for single-molecule motility assays.

Linear DNA templates with a T7 or T3 polymerase promoter were prepared by PCR or by restriction digestion at the 3′ end of the desired transcript, followed by agarose gel purification. RNA was synthesized using MEGAscript T7 (full-length *K10*; Thermo Fisher) and MEGAshortscript T7 (*hSL1-hSL2* and *TLS-KSE*; Thermo Fisher) in vitro transcription kits or the mMESSAGE mMACHINE T7 or T3 in vitro transcription kit (Thermo Fisher) to make 5′-capped transcripts for injection into *Drosophila* embryos. Fluorescence labeling of in vitro transcribed RNA was achieved by stochastic incorporation of Cy3-labeled or Cy5-labeled UTP included in synthesis reactions at a ratio in the reaction mix of 1:4 labeled to unlabeled UTP for single-molecule motility assays or 1:9 labeled to unlabeled UTP for injection assays. Following in vitro synthesis for 2–4 h at 37 °C, template DNA was digested with DNaseI and RNA was purified by phenol–chloroform–isoamyl alcohol extraction (25:24:1; Ambion) followed by successive passage through two Microspin G50 columns (Cytiva). RNA was then precipitated with ammonium acetate and ethanol. RNA concentration was determined using a NanoDrop One spectrophotometer (Thermo Fisher) and RNA integrity was confirmed by agarose gel electrophoresis before the RNA was dispensed into aliquots and stored at −80 °C.

## Cryo-EM sample preparation

For assembling Egl–BicD complexes with the *TLS*, *hSL1*, *ILS* and *GLS*, 1.5 µM of the full-length Egl–BicD complex was incubated with 25 µM RNA for 45 min at 4 °C in GF150 buffer containing 0.00125% IGEPAL (MilliporeSigma). For the Egl–BicD–*bcdSLV* and Egl–BicD–*hSL1-hSL2* complexes, a truncation of BicD (residues 322–782) was used that excludes CC1. For Egl–BicD–*bcdSLV*, 0.75 µM of Egl–BicD(322–782) was incubated with 25 µM of *bcdSLV* RNA. For the Egl–BicD–*hSL1-hSL2* complex, 0.75 µM of Egl–BicD(322–782) was incubated with 2 µM of *hSL1-hSL2* RNA. The sample was then centrifuged at 12,000*g* for 2 min.

Then, 3.5 µl of the supernatant was applied to freshly glow-discharged Quantifoil R2/2 300-square-mesh gold grids (Quantifoil) in a Vitrobot IV (Thermo Fisher) at 95% humidity and 4 °C, incubated for 10 s and blotted for 2 s before being plunged into liquid ethane.

## Cryo-EM data collection and image processing

Cryo-EM samples were imaged using a FEI Titan Krios (300 kV) equipped with a K3 detector and an energy filter with a 20-eV slit size (Gatan). Automated data collection was performed using Thermo Fisher EPU. All data collection statistics can be found in Tables 1 and 2.

**Egl–BicD–*TLS* complex.** A total of 8,568 videos were acquired at a magnification of ×81,000 (1.059 Å per pixel) using a 100-µm objective aperture, with 40 frames per video and a total fluence of ~48 e⁻ per Å². Global motion correction and dose-weighting were performed in RELION 4.0 (ref. 76) using MotionCor2 (ref. 77) with a *B* factor of 150 and 5 × 5 patches. Patch-based contrast transfer function (CTF) estimation and initial processing steps were conducted in CryoSPARC[78]. Particles were initially picked using an ellipse (180 × 100 Å) as a reference. Subsequent two-dimensional (2D) classification was used to identify protein-like densities, which were used as references for an additional round of reference-based particle picking (Supplementary Fig. 3). Approximately 5 million particles were extracted with a box size of 200 pixels and a pixel size of 2.12 Å. Then, 2D classification was performed to select ~1.6 million particles belonging to classes with protein-like features. Ab initio reconstruction generated four three-dimensional (3D) references, which were subsequently used for heterogeneous refinement. Two of these classes (class 1 and class 2; Supplementary Fig. 3) displayed density corresponding to a coiled-coil region, RNA and distinct domains of Egl. Particles from these classes were selected for further processing in RELION 4.0 and 5.0 (ref. 79). A third class (class 3) was observed that lacked density for ExoHD and XAD. In this class, the RNA and its bound ED1 and ED2 exhibited notable flexibility, which precluded high-resolution 3D refinement.

Approximately 1 million particles from classes 1 and 2 of the heterogeneous refinement were reextracted in RELION 4.0 with a box size of 280 pixels and a pixel size of 1.059 Å. This was followed by 3D refinement using class 2 as the reference. This choice was informed by the observation that the RNA and associated Egl domains on either side of the BicD coiled coil exhibited flexibility relative to each other. To improve particle alignment, one of the ExoHD–XAD modules was omitted from the 3D reference and the mask. Local particle motion was subsequently corrected using particle polishing, during which the particles were reextracted with a 360-pixel box at a pixel size of 1.059 Å. Another round of 3D refinement was performed, followed by CTF refinement to refine per-particle defocus and per-micrograph astigmatism and to estimate beam tilt and trefoil parameters. A final 3D refinement was then conducted, resulting in a 3.1-Å-resolution consensus structure. To sort for conformational and compositional heterogeneity, 3D classification without alignment was performed, focusing on either side of the BicD coiled coil. Classification with a mask around the ExoHD–XAD module included in previous 3D refinements enabled sorting of conformations within the stable part of the complex. Two major conformations (structures A and B; Supplementary Fig. 3) were identified, differing slightly in the position of the *TLS* RNA and its bound ED2, ExoHD and XAD. Structure A, at 3-Å resolution, was used for analyzing interactions among the different components of the complex and for identifying sites for mutagenesis as it was the best-resolved structure.

In addition to focused 3D classification and refinement, we performed 3D classification with a mask around the full complex that also incorporated the ExoHD–XAD module excluded in prior 3D refinements. This facilitated sorting of Egl–BicD–*TLS* structures with either one or two ExoHD–XAD modules bound, as well as the conformations within these particle populations. Approximately 50% of particles

contained two ExoHD–XAD modules, while the other particles had only one. Among the structures with two ExoHD–XAD modules, distinct conformations (structures C, D and E; Supplementary Fig. 3) were observed. These conformations shared consistent RNA-binding interactions across the domains but differed in the relative orientation of the RNA and its associated domains with respect to the BicD coiled coil. Structure C, at 3.4-Å resolution, was the best-resolved structure with all domains engaged with the RNA and was, therefore, used to describe the architecture of the complex in Fig. 1.

**Egl–BicD–*hSL1* complex.** One dataset with 12,420 videos was acquired with 1.06 Å per pixel, 50 frames per video and a total fluence of ~50 e⁻ per Å² and another with 16,453 videos was acquired with 0.91 Å per pixel, 56 frames per video and a total fluence of ~52 e⁻ per Å². For each dataset, global motion correction and dose-weighting were performed in RELION 4.0 using MotionCor2 with a *B* factor of 150 and 5 × 5 patches. Patch-based CTF estimation and initial processing steps were conducted in cryoSPARC. Particles were picked using an ellipse (180 × 100 Å) as a reference and approximately 9.1 million particles were extracted in total at a box size of 90 pixels with a binning factor of 4. Then, 2D classification was used to select approximately 4.9 million particles that belonged to 2D classes displaying protein-like densities (Supplementary Fig. 4). Ab initio reconstruction was used to generate four 3D references, which were subsequently used for two rounds of heterogeneous refinement. The class displaying density corresponding to a coiled-coil region, RNA and distinct domains of Egl (class 1; Supplementary Fig. 4) was selected after each round of heterogeneous refinement. The selected particles were reextracted at a box size of 320 pixels (1.06 Å per pixel) or 360 pixels (0.91 Å per pixel), followed by another round of ab initio reconstruction. A subset of 560,115 particles, showing well-defined densities for Egl–BicD (classes 1 and 2; Supplementary Fig. 4), was selected for further processing in RELION 4.0 and 5.0.

Particles from both the datasets were merged at this stage and a round of 3D refinement was performed. Local particle motion was subsequently corrected using particle polishing. Another round of 3D refinement was performed, followed by CTF refinement to refine per-particle defocus and per-micrograph astigmatism and to estimate beam tilt and anisotropic magnification parameters. A final 3D refinement was then conducted, resulting in a 3.3-Å-resolution consensus structure. To sort for conformational and compositional heterogeneity, 3D classification without alignment was performed, focusing on either side of the BicD coiled coil. Classification with a mask around the ExoHD–XAD module included in previous 3D refinements enabled sorting of conformations within the stable part of the complex. One major conformation (structure A; Supplementary Fig. 4) was identified for this part of the complex and was resolved at an overall resolution of 3.2 Å. Additionally, classification with a mask around the ExoHD–XAD module excluded in prior 3D refinements facilitated sorting of Egl–BicD–*hSL1* structures with either one or two ExoHD–XAD modules bound, as well as the conformations within these particle populations. Approximately 50% of particles contained two ExoHD–XAD modules, while the other particles had only one. Among the structures with two ExoHD–XAD modules, distinct conformations (structures B and C; Supplementary Fig. 4) were resolved at an overall resolution of 3.9–4.1 Å. These conformations shared consistent RNA-binding interactions across the domains but differed in the relative orientation of the RNA and its associated domains with respect to the BicD coiled coil. Structure A was resolved at the highest resolution and was, therefore, used for analyzing interactions.

**Egl–BicD–*ILS* complex.** A total of 8,009 videos were acquired at a magnification of ×81,000 (1.059 Å per pixel) using a 100-μm objective aperture, with 40 frames per video and a total fluence of ~47 e⁻ per Å². Patch motion correction and patch-based CTF estimation were performed

in cryoSPARC. Particles were picked using an ellipse (180 × 100 Å) as a reference and approximately 4.4 million particles were extracted at a box size of 200 pixels and a pixel size of 4.24 Å. Then, 2D classification was used to select approximately 2 million particles that belonged to 2D classes displaying protein-like densities (Supplementary Fig. 5). Ab initio reconstruction was used to generate three 3D references, which were subsequently used for heterogeneous refinement. The class displaying density corresponding to a coiled-coil region, RNA and distinct domains of Egl (class 1; Supplementary Fig. 5) was selected for another round of ab initio reconstruction followed by heterogeneous refinement. A subset of 360,597 particles that had well-defined densities for Egl–BicD (class 1; Supplementary Fig. 5) was selected for further processing in RELION 5.0. Local particle motion was subsequently corrected using particle polishing, during which the particles were reextracted with a 360-pixel box at a pixel size of 1.059 Å. Then, 3D refinement was performed using a mask that excluded one of the ExoHD–XAD modules, focusing on the most stable regions of the complex. This yielded a 3.4-Å-resolution structure of the Egl–BicD–*ILS* complex (structure A; Supplementary Fig. 5).

To sort for compositional heterogeneity of the ExoHD–XAD module, heterogeneous refinement was performed, classifying particles into two groups: one with a single ExoHD–XAD module and another with two. Approximately 57% of particles contained one bound ExoHD–XAD module, while ~43% contained two. The 3D refinement of the class with two ExoHD–XAD modules resulted in a 3.7-Å-resolution structure (structure B; Supplementary Fig. 5). However, local resolution distribution indicated that the ExoHD–XAD modules exhibited flexibility relative to each other, leading to decreased local resolution in these regions when both modules were included in the mask for refinement. Because of its higher quality, the 3.4-Å-resolution structure was used for model building and to elucidate the binding mechanism of Egl to the *ILS*.

**Egl–BicD–*bcdSLV* complex.** A total of 22,750 videos were acquired at a magnification of ×81,000 (1.059 Å per pixel) using a 100-μm objective aperture, with 50 frames per video and a total fluence of ~50 e⁻ per Å². Patch motion correction and patch-based CTF estimation were performed in cryoSPARC. Approximately 13 million particles (200-pixel box; 2.12 Å per pixel) were initially picked using an ellipse (180 × 100 Å) as a reference (Supplementary Fig. 6). In parallel, 3.3 million particles were also picked using Topaz[80] with a model trained using a small subset of 12,000 particles selected from 2D classification. Subsequently, a round of heterogeneous refinement using 3D references similar to those used for the *TLS*-bound structure was performed. Classes showing defined features for Egl–BicD–RNA were selected for a round of 3D classification without alignment. The particles coming from template-based picking and Topaz picking were then merged. After removal of duplicate particles, approximately 880,000 particles remained. Local particle motion was subsequently corrected in RELION 5.0, during which the particles were reextracted with a 360-pixel box at a pixel size of 1.059 Å. Another round of 3D refinement was performed, resulting in a 3.3-Å-resolution consensus structure. To sort for conformational and compositional heterogeneity, 3D classification without alignment was performed, focusing on either side of the BicD coiled coil. Classification with a mask around the ExoHD–XAD module included in previous 3D refinements enabled sorting of conformations within the stable part of the complex. One major conformation (structure A; Supplementary Fig. 6) was identified for this part of the complex after 3D classification. The selected particles were then used for 3D refinement followed by CTF refinement (beam tilt and trefoil parameters) and finally another round of 3D refinement, which resulted in a 3.4-Å-resolution structure. Additionally, classification with a mask around the ExoHD–XAD module excluded in prior 3D refinements facilitated sorting of Egl–BicD–*bcdSLV* structures with either one or two ExoHD–XAD modules bound, as well as the conformations

within these particle populations. Approximately 37% of the particles contained two ExoHD–XAD modules, while ~63% had only one. Among the structures with two ExoHD–XAD modules, distinct conformations (structures B and C; Supplementary Fig. 6) were sorted after an additional round of 3D classification with the mask around the whole molecule and resolved at an overall resolution of 4.2–4.4 Å. However, local resolution distribution indicated that the ExoHD–XAD modules exhibited flexibility relative to each other, leading to decreased local resolution in these regions when both modules were included in the mask for refinement. Structure A was resolved at the highest resolution and was, therefore, used for analyzing interactions.

**Egl–BicD–*GLS* complex.** A total of 12,592 videos were acquired at a magnification of ×81,000 (0.91 Å per pixel) using a 100-μm objective aperture, with 56 frames per video and a total fluence of ~53 e⁻ per Å². Global motion correction and dose-weighting were performed in RELION 5.0 using MotionCor2 with a *B* factor of 150 and 5 × 5 patches. Patch-based CTF estimation and initial processing steps were conducted in cryoSPARC. Approximately 6 million particles were initially picked using an ellipse (180 × 100 Å) as a reference (Supplementary Fig. 7). Subsequently, a round of heterogeneous refinement using 3D references similar to those used for the *TLS*-bound structure (Supplementary Fig. 3) was performed. A class showing defined features for Egl–BicD–RNA was selected for a round of 3D classification without alignment. The best-defined 3D class was selected and two more iterations of heterogeneous refinement and 3D classification were performed while selecting the 3D class showing defined features for Egl–BicD–RNA at each step. This resulted in a total of 41,216 particles, which were then used to train a picking model in Topaz. The picked particles (~2.3 million) were then sorted using three rounds of heterogeneous refinement and 3D classification as for particles picked using an elliptical reference, resulting in a set of 66,615 particles. Both sets of selected particles were merged and duplicated particles were removed. For this complex, while there were classes from 2D classification that showed two ExoHD–XAD modules engaged with the complex, the low number of overall particles hindered the classification of complexes with one or two ExoHD–XAD modules in 3D. Therefore, after initial particle sorting, the processing was focused on resolving the complex while having only one ExoHD–XAD module within the mask. Then, 3D refinement in RELION 5.0 was performed followed by particle polishing and CTF refinement (per-particle defocus and per-micrograph astigmatism), resulting in a 3.9-Å-resolution structure, which was used to analyze interactions.

**Egl–BicD–*hSL1-hSL2* complex.** A total of 44,979 videos were acquired at a magnification of ×81,000 (1.059 Å per pixel) using a 100-μm objective aperture, with 50 frames per video and a total fluence of ~50 e⁻ per Å². Initial image processing steps including particle picking and sorting were performed as described for the Egl–BicD–*bcdSLV* structure (Supplementary Fig. 6). A total of 1.14 million particles were obtained after initial particle sorting (Supplementary Fig. 10). Local particle motion was subsequently corrected in RELION 5.0, during which the particles were reextracted with a 360-pixel box at a pixel size of 1.059 Å. Another round of 3D refinement was performed, resulting in a consensus structure. To sort for compositional heterogeneity, 3D classification without alignment was performed using a mask encompassing the full complex. We found that ~60% (675,190) of the particles had Egl–BicD bound to both *hSL1* and *hSL2*, representing the Egl–BicD–*hSL1-hSL2* complex. The remaining particles displayed both stem-loop-binding sites occupied by *hSL1* stem loops, resembling the Egl–BicD–*hSL1* complex.

To sort for conformational heterogeneity within the Egl–BicD–*hSL1-hSL2* complex, another round of 3D classification without alignment was performed using a mask encompassing the full complex. This identified a major conformation, which was used for further 3D

refinement, yielding a map at 3.4-Å resolution. This structure was used to analyze the interaction of *hSL2* with Egl–BicD.

Although the linker between *hSL1* and *hSL2* in the Egl–BicD–*hSL1-hSL2* structure appeared flexible, we detected additional low-resolution density connecting the two stem loops (Supplementary Fig. 10c) that was absent in the structures with only *hSL1* bound. This observation shows that both stem loops in the Egl–BicD–*hSL1-hSL2* structure originate from the same RNA molecule (Supplementary Fig. 10c).

We did not observe 3D classes where the ExoHD–XAD module was bound to the *hSL2* stem loop; in contrast, this module was consistently bound to only *hSL1*. In cases where both stem-loop-binding sites were occupied by *hSL1* either in this dataset or in the Egl–BicD–*hSL1* complex dataset, we observed structural classes with two ExoHD–XAD modules bound, in addition to those with only one (Supplementary Figs. 4 and 10). These observations, together with those presented in Fig. 5, indicate that *hSL2* is not compatible with ExoHD–XAD binding.

### Model building and refinement

For modeling Egl, the model of *D. melanogaster* Egl isoform B (UniProt Q9W1K4) generated using AlphaFold2 (ref. 38) (AF-Q9W1K4-F1) was obtained from the AlphaFold Protein Structure Database[81]. The individual domains (ED1, ED2, ExoHD and XAD) were fitted into the cryo-EM density using UCSF ChimeraX[82] guided by the side-chain densities and the shapes of the domains. Given the flexible linkers between domains in Egl, their assignment to specific Egl chains was guided by multiple structural criteria. ED1 and ED2 positioned across the BicD coiled coil were assigned to the same Egl molecule on the basis of cryo-EM density at low threshold connecting these domains and a linker length compatible with this arrangement. The ExoHD–XAD module was assigned to the same chain as it is adjacent to ED2. This was supported by an interaction between residues 386–394 and the ExoHD, which reduces the intervening sequence between ED2 and ExoHD to approximately 217 residues. Additional folded elements within this region likely shorten the effective linker length even further, making it more likely that the ExoHD adjacent to ED2 belongs to the same chain. Although less favorable, it is possible that the ExoHD–XAD module comes from a different Egl molecule to their neighboring ED2.

For modeling BicD, an AlphaFold2 prediction of *D. melanogaster* BicD (UniProt P16568) was used. The model was fitted into the cryo-EM map using UCSF ChimeraX guided by the side-chain densities and an AlphaFold2 prediction of ED1 (residues 1–82) bound to BicD residues 700–782.

For modeling the RNA stem loops, 3D structures predicted by AlphaFold3 server[83] or those generated by RNAComposer server[84] were used as starting models.

After placing individual protein domains or RNA stem loops into the cryo-EM map, the model went through iterative cycles of restrained flexible fitting using ISOLDE[85], followed by user-guided refinement in ISOLDE or Coot[86]. Final model refinement and model validation were performed in PHENIX[87]. All refinement statistics can be found in Tables 1 and 2.

### AlphaFold2 prediction

All structure predictions unless specified were performed using AlphaFold2 (ref. 38) (for single chains) or AlphaFold2-Multimer[88] (for multiple chains) through a local installation of ColabFold[89]. Both AlphaFold2 and AlphaFold3 accurately predicted the interaction between ED1 and BicD. However, we were unable to obtain reliable predictions for the Egl–BicD complex bound to different RNA targets.

### RNA secondary-structure prediction

RNA secondary structures were predicted using the RNAfold web server[90] (http://rna.tbi.univie.ac.at/cgi-bin/RNAWebSuite/RNAfold.cgi). Some nucleotides predicted to be part of bulges or loops were

found to be integrated into the dsRNA helix in the experimental structures, where they participated in stacking and noncanonical base-pair interactions (Supplementary Fig. 8b). Therefore, when generating the Δb constructs, nucleotides were removed on the basis of experimental structures.

## MST

The affinity of the Egl–BicD complex and its variants for RNA stem loops was measured using a NanoTemper Monolith instrument and MO.Control MST acquisition software (NanoTemper). Twofold serial dilutions of the Egl–BicD complex ranging from $1.59 \times 10^{-10}$ M to $7.70 \times 10^{-6}$ M (assuming 2:2 stoichiometry of Egl:BicD) were incubated with 1 nM Cy5-labeled RNA at room temperature for 15 min in GF150 buffer supplemented with 0.05% Tween-20. Higher concentrations of Egl–BicD could not be tested as they were prone to aggregation. Serial dilutions were loaded into standard capillaries (MO-K022) and irradiated with infrared light at room temperature for 10 s (medium MST power) at 20% excitation, with changes in fluorescence monitored by the pico-red detector. MST traces were analyzed in MO.Affinity Analysis with data fitting performed using 1.5 s on-time ($F_{hot}$ = 0.5–1.5 s; $F_{cold}$ = −0.5–0 s) and the $K_d$ model, which was able to fit binding curves in the absence of saturated RNA binding given sufficient signal-to-noise ratio (minimum value of 5.0 to be considered interacting). Data were further analyzed in GraphPad Prism (version 10.4.0) to determine the s.d. of the fit values of $K_d$.

## Injection of *Drosophila* embryos with fluorescent RNA

Wild-type embryos ($w^{1118}$ strain; Bloomington *Drosophila* Stock Center, BL5905) were collected and injected with Cy3-labeled RNA as described previously[18,46,51]. Typically, up to 60 dechorionated embryos were mounted in Voltalef oil 10S for injection of syncytial blastoderms with a 250 ng μl$^{-1}$ solution of RNA. The person performing the injections was blinded to the identity of the RNA being evaluated. Following the last injection, embryos were incubated at room temperature for 8 min (~13 min from injection of the first embryo) before fixation with formaldehyde-saturated heptane, removal of the vitelline membrane with fine syringe needles and mounting in Vectashield with DAPI (Vector Labs) for visualization of nuclei.

Imaging was conducted using a Zeiss LSM 710 or 780 confocal microscope (Supplementary Table 4) equipped with a ×40 (1.3 numerical aperture (NA)) oil-immersion objective. Laser intensity was adjusted to allow visualization of injected RNA without reaching saturation.

RNA localization efficiency was quantified by comparing in FIJI[91] the apical and basal RNA intensity within uniform regions of interest (ROIs), whose size (7.391 μm$^2$) was predefined as the mean area occupied by apically localized RNA at individual microtubule organizing centers (positioned just above nuclei) in ~100 embryos injected with wild-type *K10* 3′ UTR. For each image, four ROIs were placed in the apical region of the injection site, with each ROI centered on the brightest area of fluorescence intensity. Corresponding basal ROIs were translated vertically to a basal position just above the yolk. The mean fluorescence intensities of apical and basal ROIs were then averaged and background-corrected by subtracting the mean intensity of a basal region distant from the injection site before being expressed as a ratio of apical to basal intensity.

## Single-molecule-resolution RNA motility assays

Total internal reflection fluorescence (TIRF)-based motility assays of reconstituted dynein transport complexes assembled with the indicated RNAs were performed as previously described[34]. Assembly mixes of 100 nM dynein, 500 nM Egl–BicD, 200 nM dynactin and 50 nM RNA (25 nM each of Cy3-labeled and Cy5-labeled samples) were incubated in a total volume of 5 μl of GF150 on ice for 1 h before imaging. Complexes were assembled with a tenfold molar deficit of total RNA relative to Egl–BicD to assess the sufficiency of single RNAs to activate motility. Assembly mixes were diluted 10–80-fold in motility buffer (30 mM HEPES pH 7.3, 50 mM KCl, 5 mM MgSO$_4$, 1 mM EGTA pH 7.5, 1 mM DTT, 20 μM Taxol (Sigma), 0.5 mg ml$^{-1}$ BSA and 1 mg ml$^{-1}$ α-casein (Sigma)) with 1 mM Mg-ATP and an oxygen-scavenging system (1.25 μM glucose oxidase, 140 nM catalase, 71 mM 2-mercaptoethanol and 25 mM glucose) and applied to a ~10-μl flow chamber containing streptavidin-immobilized microtubules (labeled with HiLyte 488 and biotin porcine tubulin; Cytoskeleton) on a PEG-biotin-passivated cover slip. Motility of Cy3-labeled and Cy5-labeled RNA within these chambers was alternately recorded with an iXon$^{EM}$ + DU-897E electron-multiplying charge-coupled device camera (Andor) mounted on a Nikon TIRF system (Supplementary Table 4) with a Nikon APO TIRF ×100 (1.49 NA) oil objective using Micro-manager software[92] at the maximum possible frame rate (~2 frames per s for each channel) and 100 ms of exposure for each channel. Samples were illuminated with a 150-mW Coherent Sapphire 488-nm laser, a 150-mW Coherent Sapphire 561-nm laser and a 100-mW Coherent CUBE 641-nm laser. Assemblies of dynein–dynactin–BicD–Egl complexes with Cy3-labeled and Cy5-labeled *TLS-KSE* constructs were imaged using a Nikon Ring-TIRF system (Supplementary Table 4) controlled by Micro-manager and equipped with the iLas 2 platform (GATACA Systems) and the same objective as above. These samples were illuminated for 200 ms with 488-nm, 561-nm and 647-nm lasers within a Cairn Multiline Kompact laser box and the motility in Cy3 and Cy5 channels was recorded simultaneously on independent Photometrics Prime 95B complementary metal–oxide–semiconductor cameras at the maximal frame rate (~4 frames per s). Colocalization of Cy3 and Cy5 RNA was analyzed manually in FIJI using kymographs derived from acquisitions described above.

## RNA bend analysis

A custom Python program (https://github.com/carterlablmb/RNA_bend_analysis) was used to calculate the bend between the upper and lower helices of RNA localization signals. First, the Curves+ software[93] was used to obtain a helical trajectory for each RNA stem loop. From these trajectories, the coordinates representing the upper and lower helical regions were separately selected and fitted to best-fit lines in 3D space using a singular value decomposition algorithm. Each best-fit line was defined by a centroid and a principal axis vector and the bend angle (in degrees) was determined by computing the dot product of the two principal axis vectors and then applying the inverse cosine.

## Egl sequence conservation analysis and RNA alignments

Egl orthologs were identified by BLASTp[94] searches using the *D. melanogaster* Egl protein sequence (UniProt Q9W1K4) as a query against the nonredundant protein database. Hits displaying ≥55% sequence identity and ≥60% sequence coverage were retained. Orthologous sequences from 108 different species were used for multiple-sequence alignment (MSA). Sequences were aligned using Clustal Omega[95] with default parameters. The resulting MSA was imported into UCSF ChimeraX and per-residue conservation scores were calculated using the built-in sequence conservation analysis tools. Conservation values were mapped onto structural models for visualization.

Alignments of RNA localization signals and support elements were performed using the UCSC Genome Browser[96] and the following genome assemblies: *D. melanogaster*, August 2014 (BDGP Release 6 + ISO1MT/dm6); *Drosophila simulans*, September 2014 (ASM75419v2/droSim2); *Drosophila yakuba*, June 27, 2006 (dyak_caf1/droYak3); *Drosophila biarmipes*, March 4, 2013 (Dbia_2.0/droBia2); *Drosophila takahashii*, March 4, 2013 (Dtak_2.0/droTak2); *Drosophila kikkawai*, March 4, 2013 (Dkik_2.0/droKik2).

## Data visualization and statistical analysis

Cryo-EM maps and models were rendered using ChimeraX. Particle angular distribution was plotted using starparser[97]. Background-subtracted kymographs were generated using FIJI. Images

of *Drosophila* embryos injected with fluorescent RNA were analyzed using unsaturated raw pixel values; however, to aid presentation of apical versus basal RNA localization, lookup tables in representative images had maximum pixel values set to the mean intensity + 10 s.d. and the minimum pixel values set to the mode (typically 0 or 1). Plotting of data and statistical analyses were performed using GraphPad Prism (version 10.4.0). The normality or nonnormality of datasets was assumed but not explicitly tested.

**Reporting summary**

Further information on research design is available in the Nature Portfolio Reporting Summary linked to this article.

## Data availability

Atomic coordinates and cryo-EM maps were deposited to the Protein Data Bank and EM Data Bank, respectively, under accession codes PDB 9RVY and EMD-54292 (Egl–BicD–*TLS* structure A), PDB 9RVZ and EMD-54293 (Egl–BicD–*TLS* structure B), PDB 9RW0 and EMD-54294 (Egl–BicD–*TLS* structure C), PDB 9RW1 and EMD-54295 (Egl–BicD–*TLS* structure D), PDB 9RW2 and EMD-54296 (Egl–BicD–*TLS* structure E), PDB 9RW3 and EMD-54297 (Egl–BicD–*hSL1* structure A), PDB 9RW4 and EMD-54298 (Egl–BicD–*hSL1* structure B), PDB 9RW5 and EMD-54299 (Egl–BicD–*hSL1* structure C), PDB 9RW6 and EMD-54300 (Egl–BicD–*ILS* structure A), PDB 9RW7 and EMD-54302 (Egl–BicD–*bcdSLV* structure A), PDB 9RW8 and EMD-54303 (Egl–BicD–*bcdSLV* structure B), PDB 9RW9 and EMD-54304 (Egl–BicD–*bcdSLV* structure C), PDB 9RWA and EMD-54305 (Egl–BicD–*GLS*) and PDB 9RWB and EMD-54306 (Egl–BicD–*hSL1-hSL2*). For Egl–BicD–*ILS* structure B, only the map was deposited to the EM Data Bank under accession code EMD-54301. All other data, including large imaging datasets, are available upon reasonable request to the corresponding authors. Source data are provided with this paper.

## Code availability

The custom program for analyzing the bending angle of RNA helices is available from GitHub (https://github.com/carterlablmb/RNA_bend_analysis).

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

## Acknowledgements

We thank members of the A.P.C. and S.L.B. groups for their advice and support and the late D. Ish-Horowicz for long-term encouragement. We are grateful to the Laboratory of Molecular Biology (LMB) EM Facility for access to and support with EM sample preparation and data collection. We thank J. Grimmett, T. Darling and I. Clayson of LMB Scientific Computing for providing resources, the LMB media preparation team for supplying fly food, K. Turton and the LMB Baculovirus Facility for support with insect cell cultures and S. McLaughlin from the LMB Biophysics Facility for assistance with MST measurements. We also acknowledge the Diamond Light Source for access and support of the cryo-EM facilities at the UK's national Electron Bioimaging Center (proposal bi23268). Work in the A.P.C.

and S.L.B. groups is funded by the Medical Research Council as part of UK Research and Innovation (UKRI) (file reference numbers MC_UP_A025_1011 and MC_U105178790, respectively). The work was also supported by a UKRI Biotechnology and Biological Sciences Research Council (BBSRC) project grant to S.L.B. (BB/T00696X/1), a European Molecular Biology Organization postdoctoral fellowship to K.S. (ALTF 197-2021) and a UKRI BBSRC PhD studentship to S.C. (project reference 2273135 as part of BB/M011194/1). The funders had no role in study design, data collection and analysis, decision to publish or preparation of the manuscript.

## Author contributions

K.S., M.A.M., A.C. and S.L.B conceptualized the project. K.S., S.C., M.A.M. and S.L.B. generated the data. All authors analyzed the data and wrote the manuscript. A.P.C. and S.L.B. supervised the project.

## Competing interests

The authors declare no competing interests.

## Additional information

**Extended data** is available for this paper at https://doi.org/10.1038/s41594-026-01794-8.

**Correspondence and requests for materials** should be addressed to Mark A. McClintock, Andrew P. Carter or Simon L. Bullock.

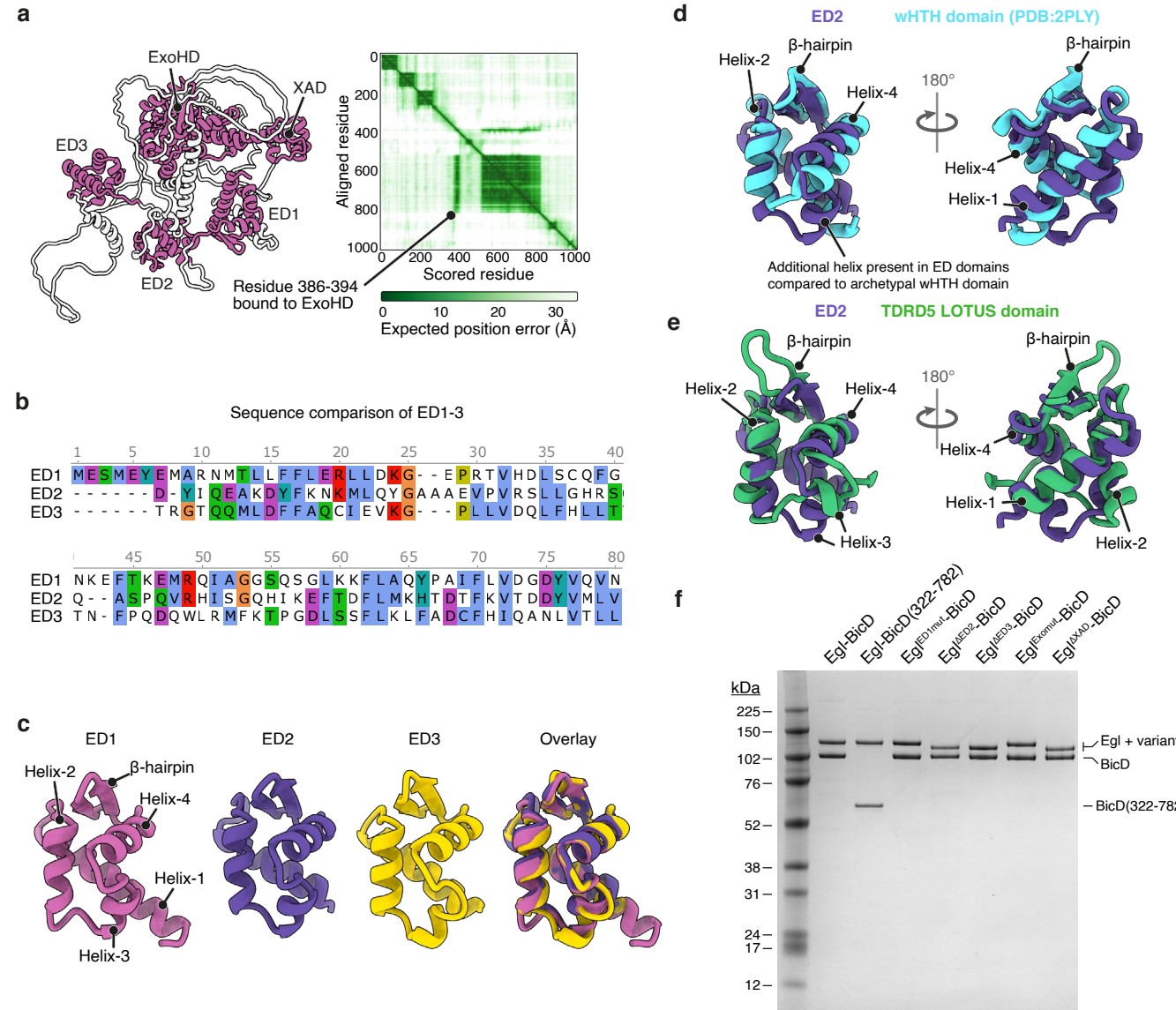

**Extended Data Fig. 1 | Predicted domains in Egl. a**, AlphaFold2 prediction of Egl with associated predicted aligned error (PAE) plot (dark green indicates high degree of confidence of local predictions). Structured domains within the RNA-binding region of Egl are shown in magenta. **b**, Sequence alignments of ED1–3. Alignments are colored according the Clustal X color scheme in which blue represents hydrophobic residues, magenta represents negatively charged residues, green represents polar residues, cyan represents aromatic residues, red represents positively charged residues, orange represents glycines, and yellow represents prolines. **c**, AlphaFold2 predictions of ED1–3, showing similar winged

helix-turn-helix (wHTH) folds. **d**, Comparison of predicted structures of EDs (using ED2 as representative) with a canonical wHTH fold in the selenocysteine-specific tRNA elongation factor SelB[98] showing the addition of ED helix-3 to the canonical wHTH archetype. **e**, Comparison of predicted structures of EDs (using ED2 as representative) with the AlphaFold2 prediction for the LOTUS domain of human TDRD5 (residues 294–396 of Uniprot ID Q8NAT2). **f**, SDS-PAGE image of all purified Egl–BicD constructs used in this study. The image is representative of the purity of Egl–BicD complexes from > 15 independent preparations.

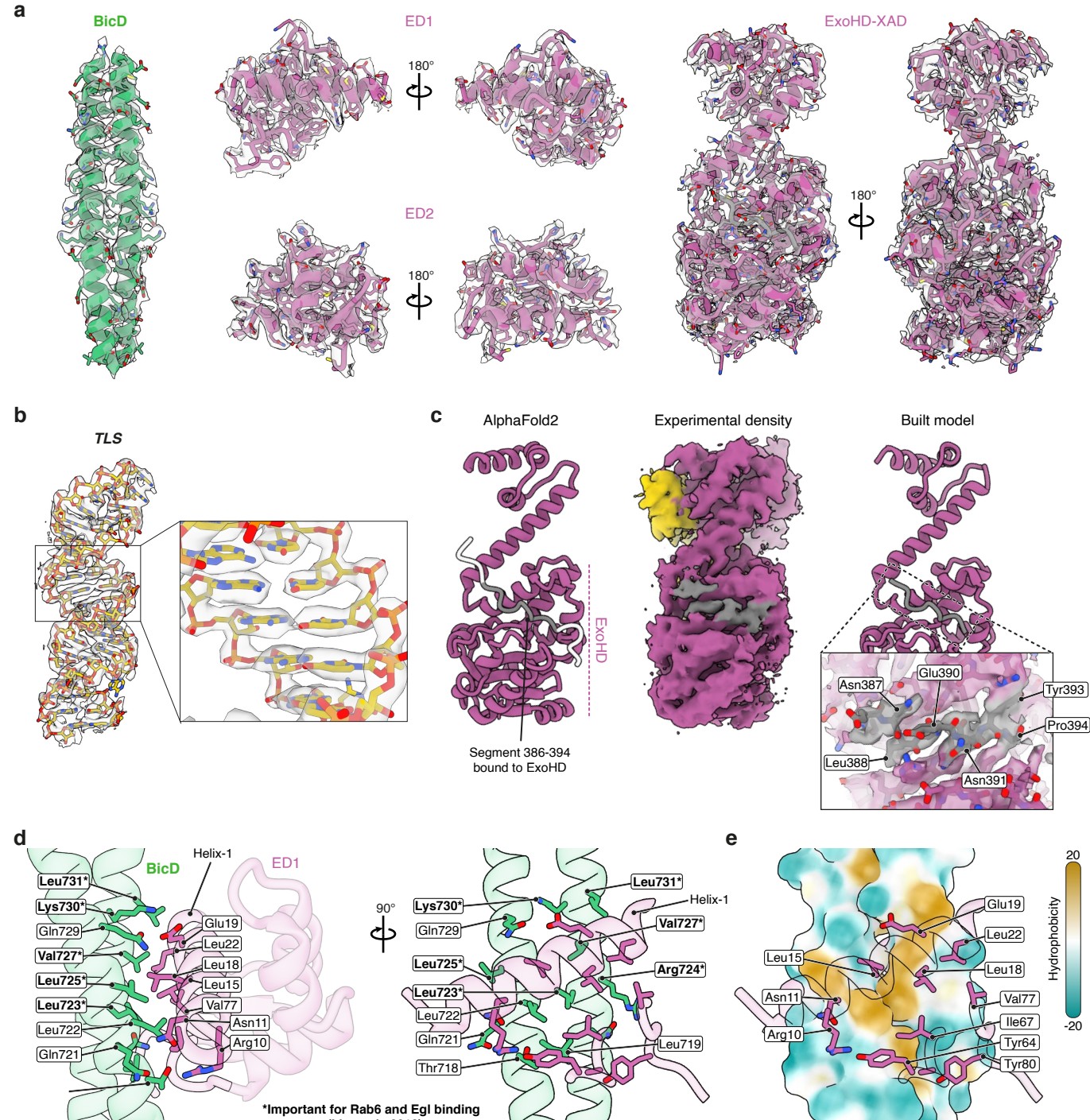

**Extended Data Fig. 2 | Cryo-EM density and modeled residues in the Egl–BicD–*TLS* complex. a** and **b**, Cryo-EM density and modeled residues for Egl–BicD (**a**) and the *TLS* stem loop (**b**) within the Egl–BicD–*TLS* structure. **c**, AlphaFold2 prediction of a nine-residue segment (residues 386–394, gray) bound to the ExoHD (left), compared to the experimental density of this region (center) and the model built into the density (right). **d**, Interaction interface between BicD and ED1, highlighting residues previously shown[99] to be important for BicD binding by Rab6 and Egl (bolded). **e**, Surface representation of the BicD coiled coil, colored by hydrophobicity (brown = hydrophobic, teal = hydrophilic), revealing a hydrophobic patch at the ED1 binding site. A cartoon representation of the interacting ED1 residues is overlaid.

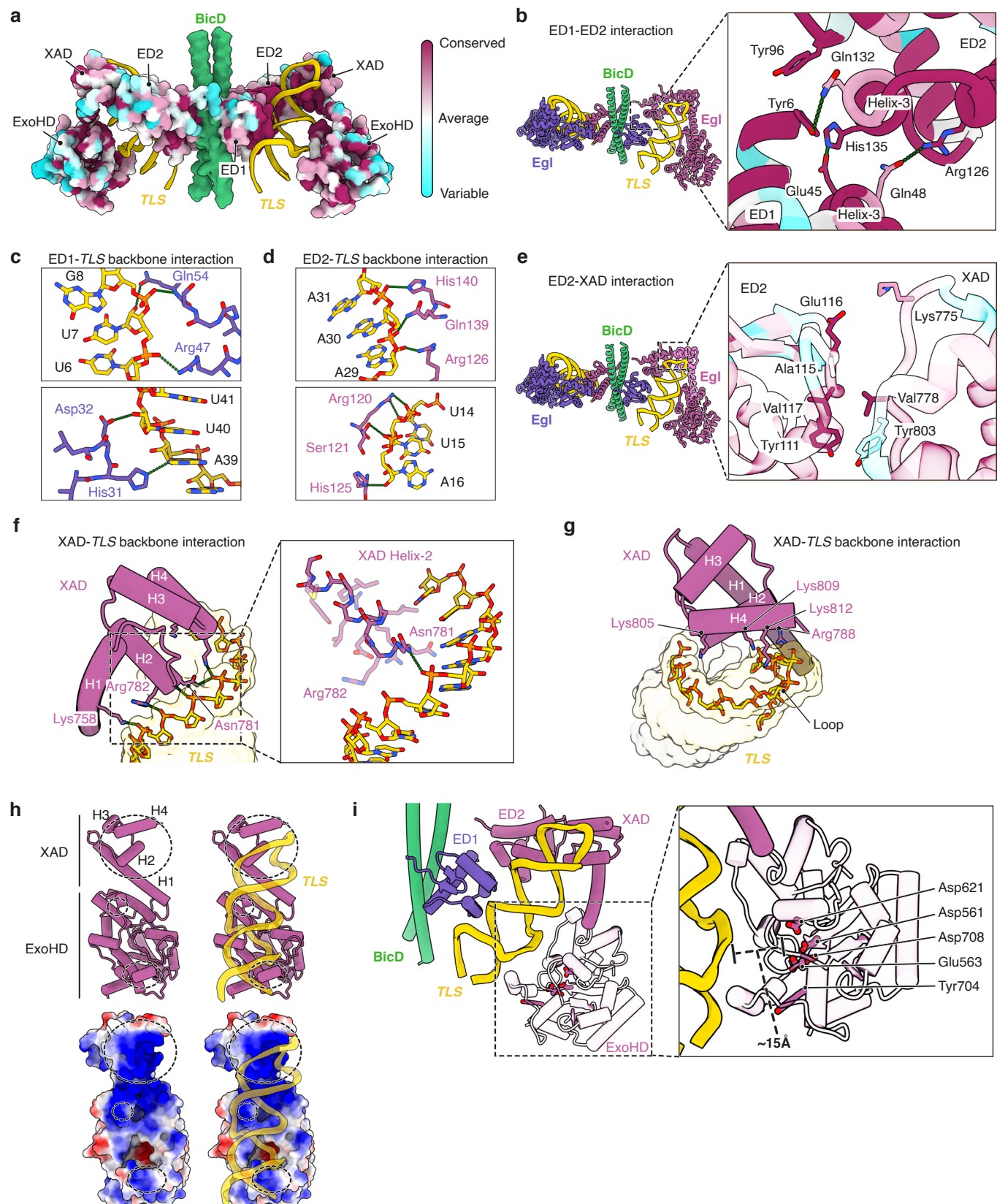

**Extended Data Fig. 3 | See next page for caption.**

**Extended Data Fig. 3 | Interactions within the *TLS* binding pocket of Egl.**
**a**, Egl–BicD–*TLS* complex with a surface representation of Egl colored according
to evolutionary sequence conservation. **b**, Binding interface between ED1
and ED2 when bound to the *TLS*, which is mediated by polar interactions that
are shown in green. Residues are colored based on their conservation using
the color scale shown in **a**. **c** and **d**, Interactions of ED1 (**c**) and ED2 (**d**) with the
ribose-phosphate backbone of the *TLS*. **e**, Binding interface between ED2 and
the XAD when bound to the *TLS*. The interactions are primarily hydrophobic
in nature driven by Val and Tyr residues. Residues are colored based on their
conservation using the color scale shown in **a**. **f** and **g**, Interactions between the
XAD and the ribose-phosphate backbone of the upper helix and loop of the *TLS*.
The N-terminal dipole of XAD helix-2 (H2) interacts with the phosphate backbone
of the upper helix (**f**), whereas helices-2 and -4 make contacts across the major
groove at the stem-loop junction (**g**). **h**, Cartoon representation (top) and surface
charge representation (bottom) of ExoHD–XAD overlaid with the bound *TLS*
stem loop (blue: positive charge, red: negative charge, white: neutral). Positively-
charged regions of ExoHD–XAD that bind the *TLS* are highlighted with a dotted
circle. **i**, Location of the putative catalytic residues of ExoHD and their average
distance from the bound *TLS* stem loop are shown.

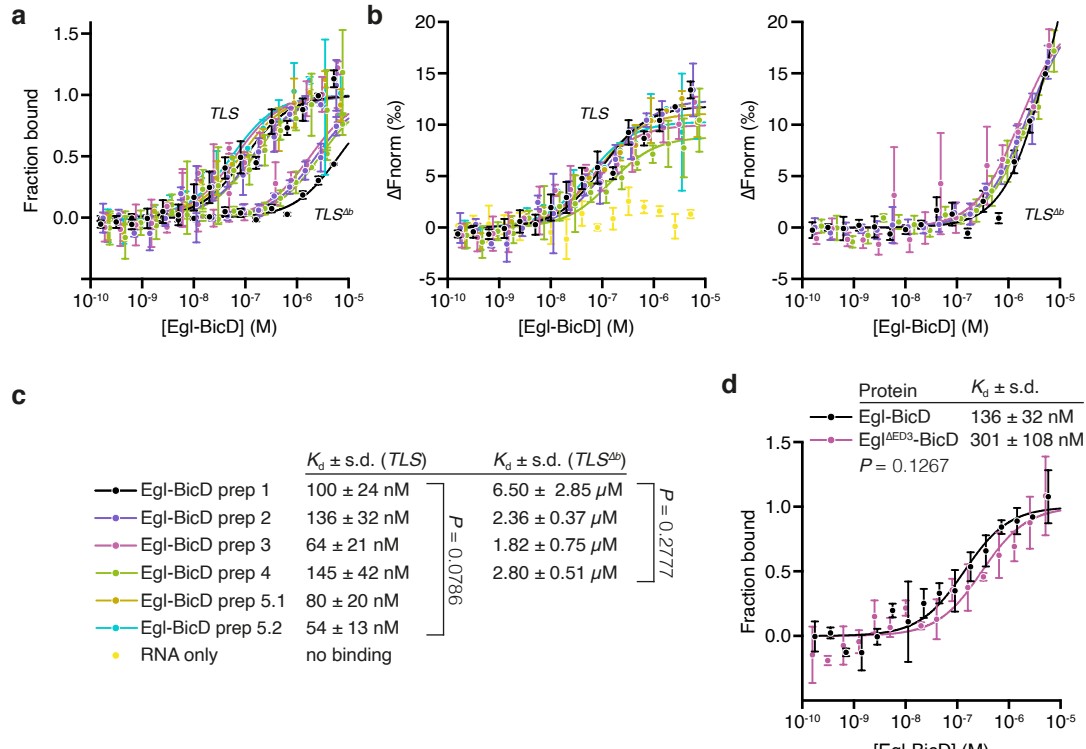

**Extended Data Fig. 4 | Demonstration of a robust MST assay and no significant effect of ED3 deletion on *TLS* binding. a–c**, MST data for the *TLS* or a non-localizing negative control mutant (*TLS^Δb*) bound to independent preparations of wild-type Egl–BicD used in this study. Each curve is normalized to show either the fraction of *TLS* that is bound by Egl–BicD (**a**), or the response amplitude (ΔFnorm (‰); **b**) relative to a *TLS*-only control (yellow dots) in which samples were scanned in the same order as those containing Egl–BicD dilutions. Data points show the mean ± s.d. for three (Prep 1: *TLS^Δb*, Prep 2: *TLS* and *TLS^Δb*, Prep 3: *TLS* and *TLS^Δb*, Prep 5.1: *TLS* and *TLS^Δb*, and Prep 5.2: *TLS* and *TLS^Δb*), four (Prep 4: *TLS^Δb*), five

(Prep 4: *TLS*) or six (Prep 1: *TLS*) independent experiments per condition, from which best-fit values for $K_d$ ± s.d. were derived (**c**). No significant deviation in affinity constants for the same RNA is seen between preparations, revealing that this is a robust assay. **d**, MST binding curve for the *TLS* bound to wild-type Egl–BicD or a variant lacking ED3. Data points show the mean ± s.d. for three independent experiments per condition, from which best-fit values for $K_d$ ± s.d. were derived. Statistical significance was determined using a one-tailed extra sum-of-squares F test.

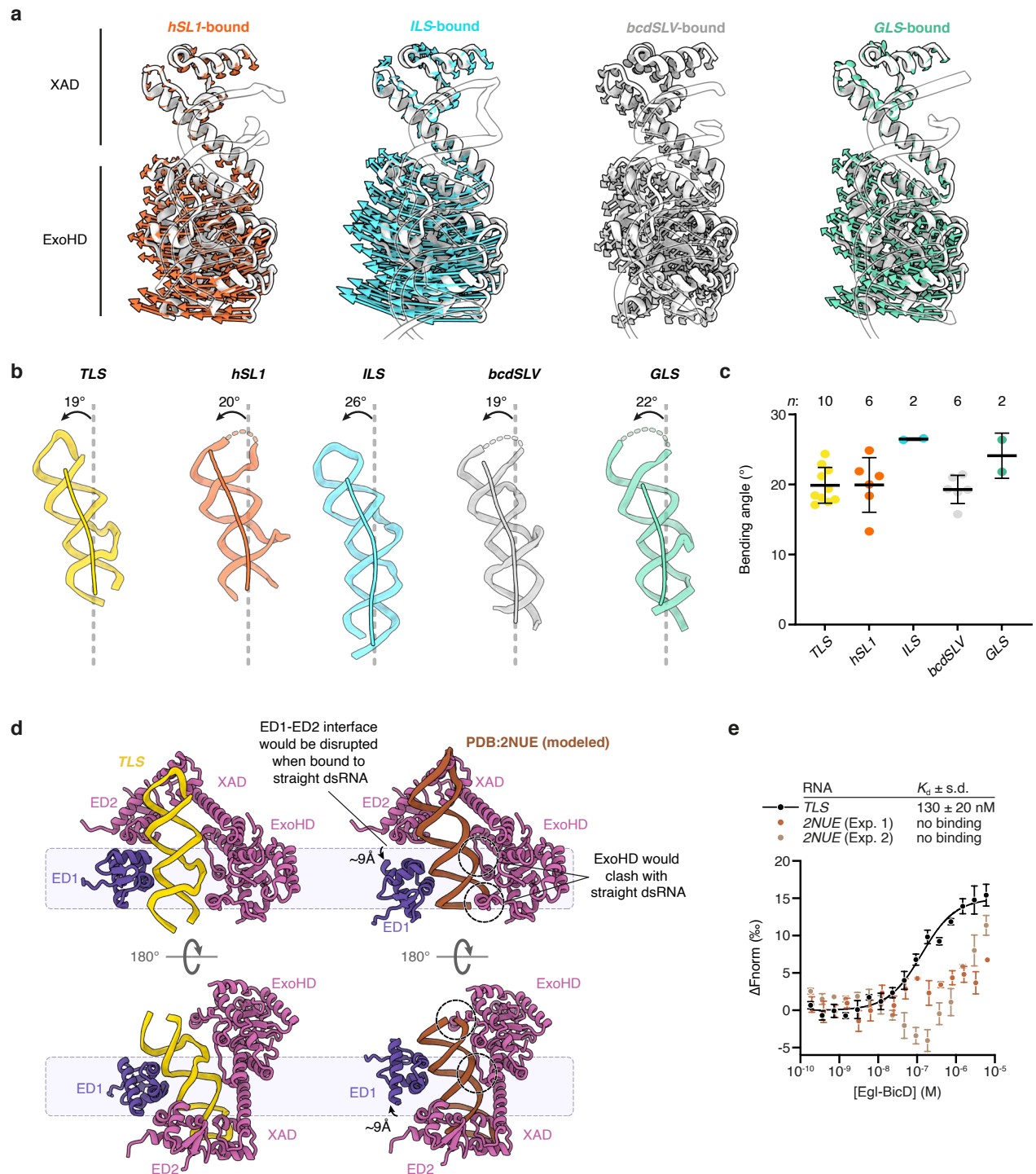

**Extended Data Fig. 5 | Structural comparison of localization signals and requirement for RNA bending in Egl–BicD engagement. a**, Models showing the change in the position of ExoHD–XAD bound to *hSL1, ILS, bcdSLV* or *GLS* compared to the *TLS*-bound structure (in white) using arrows. The corresponding RNAs are shown as translucent ribbons with gray outlines. **b**, Comparison of the bending angle between upper and lower helices of the five Egl-bound localization signals. The RNA stem loops are depicted as cartoons, with the helical trajectories (calculated using Curves +) overlaid as lines. **c**, Quantification of RNA bending angles measured from cryo-EM structures of different conformations of each Egl–BicD–RNA complex. Both bound copies of the localization signal from each 3D refined structure were analyzed (see Supplementary Figs. 3–7). Data points and mean ± s.d. are shown, with the total number of localization signals used for the analysis (*n*) indicated above each condition. **d**, Comparison of cryo-EM

structure of Egl–BicD bound to *TLS* (left) to a model of a straight A-form dsRNA stem loop (PDB: 2NUE, right) that has ED1 and ED2 bound to adjacent minor grooves and the XAD bound to the terminal loop. A straight dsRNA stem loop disrupts the ED1–ED2 interface and is not compatible with the simultaneous binding of both the XAD and ExoHD, illustrating the requirement for RNA bending to engage all four RNA-binding domains. **e**, MST binding assay for Egl–BicD with the *TLS* and a straight A-form dsRNA stem loop (PDB: 2NUE). Two independent experiments were performed for the *2NUE* RNA using different preparations of Egl–BicD. *2NUE* experiment 2 was performed in parallel with the data presented in Fig. 4f. In both experiments, the low signal-to-noise ratio indicates that Egl–BicD fails to bind a straight A-form dsRNA helix. Data points show the mean ± s.d. for three independent experiments per condition.

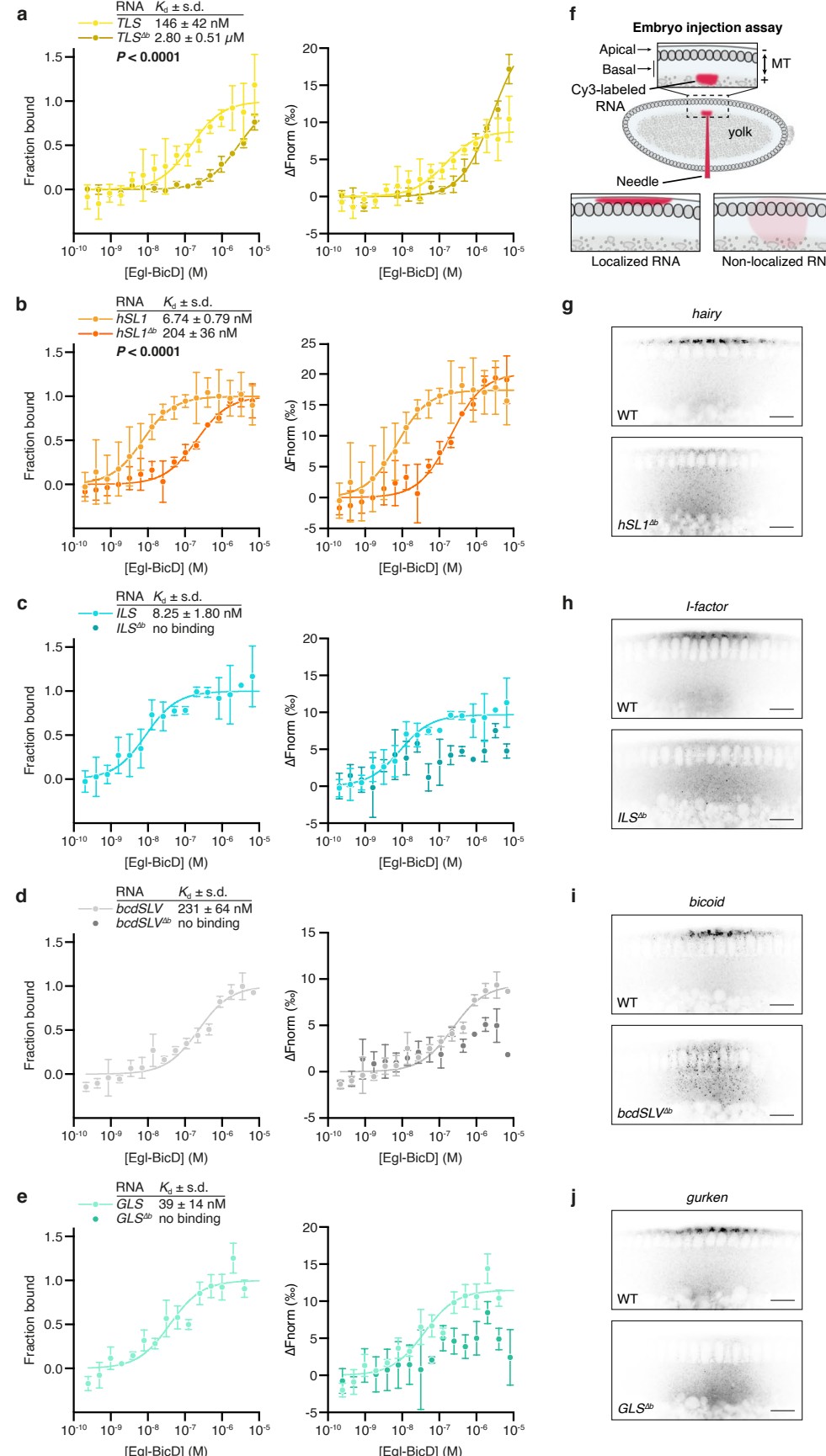

**Extended Data Fig. 6 | See next page for caption.**

**Extended Data Fig. 6 | Deletion of bulged nucleotides from localization signals impairs interaction with Egl–BicD and localization of cognate transcripts in *Drosophila* embryos. a**–**e**, MST curves for Egl–BicD with the *TLS* (**a**), *hSL1* (**b**), *ILS* (**c**), *bcdSLV* (**d**) and *GLS* (**e**) and respective bulge-deletion mutants (Δb). Each curve is normalized to show either the fraction of *TLS* that is bound by Egl–BicD or the response amplitude (ΔFnorm (‰)). Low signal-to-noise ratios for response amplitudes indicate a lack of binding at the concentrations tested (see Methods). Data points show the mean ± s.d. for three (all except *TLS*) or five (*TLS*) independent experiments per condition, from which best-fit values for $K_d$ ± s.d. were derived. Bold *P* values denote statistical significance, determined using a one-tailed extra sum-of-squares F test. **f**, Diagram of the *Drosophila* embryo injection assay that assesses dynein-mediated localization of injected transcripts. RNAs that are efficiently transported by dynein accumulate apically at microtubule (MT) minus ends, while most of the population of a non-localizing RNA remains in the basal cytoplasm[100]. **g**–**j**, Confocal images of *Drosophila* embryos injected with fluorescently-labeled *hairy* (**g**), *I-factor* (**h**), *bicoid* (**i**) and *gurken* (**j**) mRNAs and respective mutants in which bulged nucleotides were deleted from their localization signals. Images are representative of 39–78 injected embryos as specified in Fig. 3e. Scale bar, 20 μm.

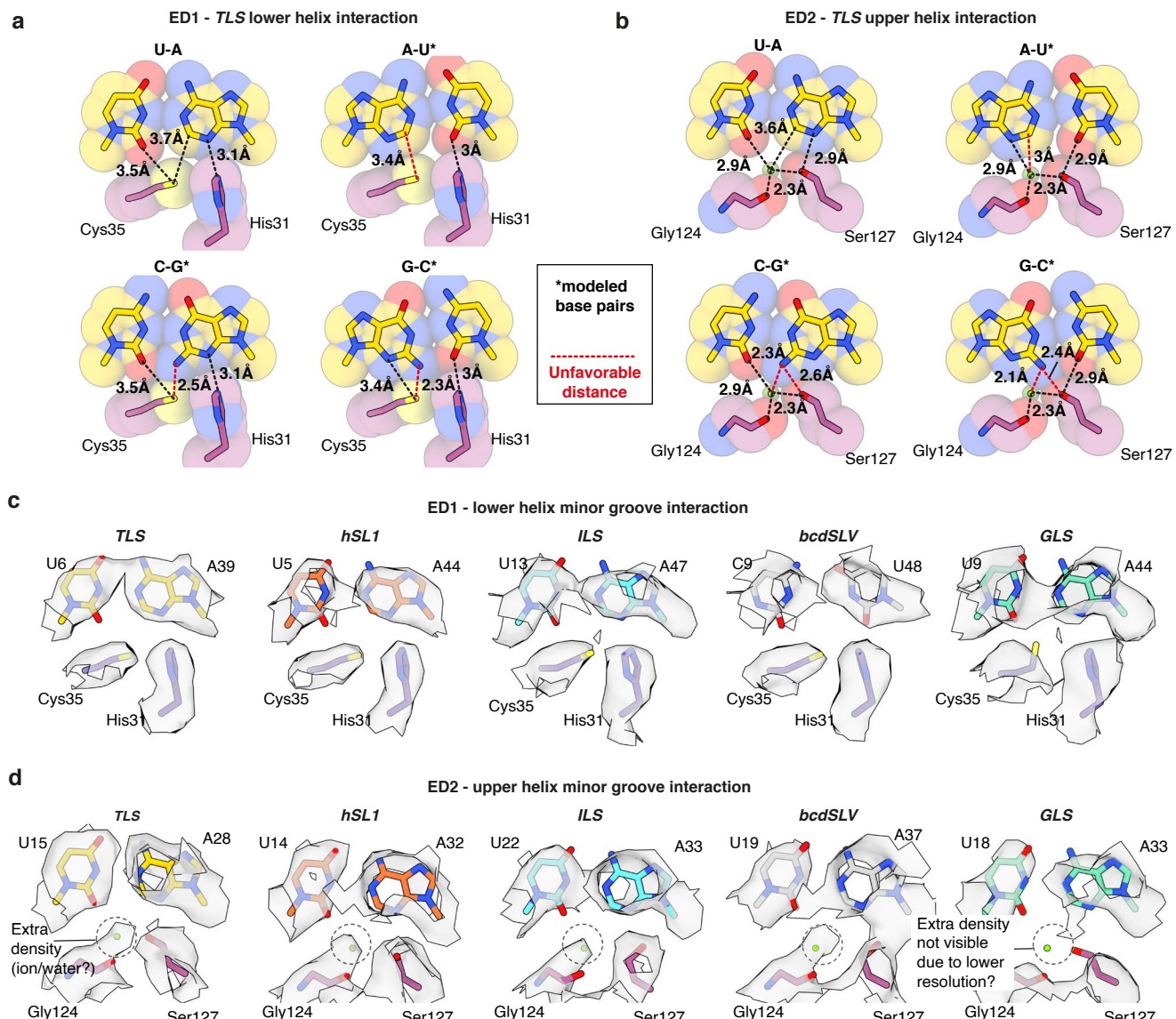

**Extended Data Fig. 7 | ED1 and ED2 enforce base selectivity at minor groove interaction sites. a**, ED1–*TLS* lower-helix interaction showing the wild-type U–A base pair and modeled alternative base pairs (A–U, C–G and G–C). Base pairs are displayed in stick representation overlaid with space-filling spheres scaled according to van der Waals radii. Distances between ED1 residues (Cys35 and His31) and atoms within the base pair are indicated. Unfavorable distances (red dashed lines) denote steric clashes defined as heavy-atom separations shorter than the sum of the corresponding van der Waals radii. Relevant contact limits are: C-S = 3.50 Å, O-S = 3.32 Å and N-S = 3.35 Å. In contrast, hydrogen bonds between compatible atoms typically occur at heavy-atom separations of ~2.7–3.2 Å (donor-acceptor distance) and were not considered unfavorable. **b**, ED2–*TLS* upper-helix interaction shown as in **a**. Base pair substitutions introduce steric incompatibilities according to the same van der Waals criteria. The additional density adjacent to Ser127 was evaluated assuming a magnesium ion. Relevant contact limits are: O-N = 3.07 Å, Mg-N = 3.28 Å and Mg-C = 3.43 Å. **c**, Cryo-EM density corresponding to the ED1 minor groove interaction across the five localization signal structures. Atomic models are shown as sticks within the corresponding density. **d**, Cryo-EM density corresponding to the ED2 minor groove interaction across the five localization signal structures. An additional density adjacent to Ser127, consistent with an ordered solvent molecule or ion, is indicated (dashed circle). In the *GLS* structure, this feature is not resolved, likely due to the lower overall resolution of the structure. In **a** and **b**, models of the wild-type U–A base pairs are reproduced from Fig. 4f to facilitate comparison.

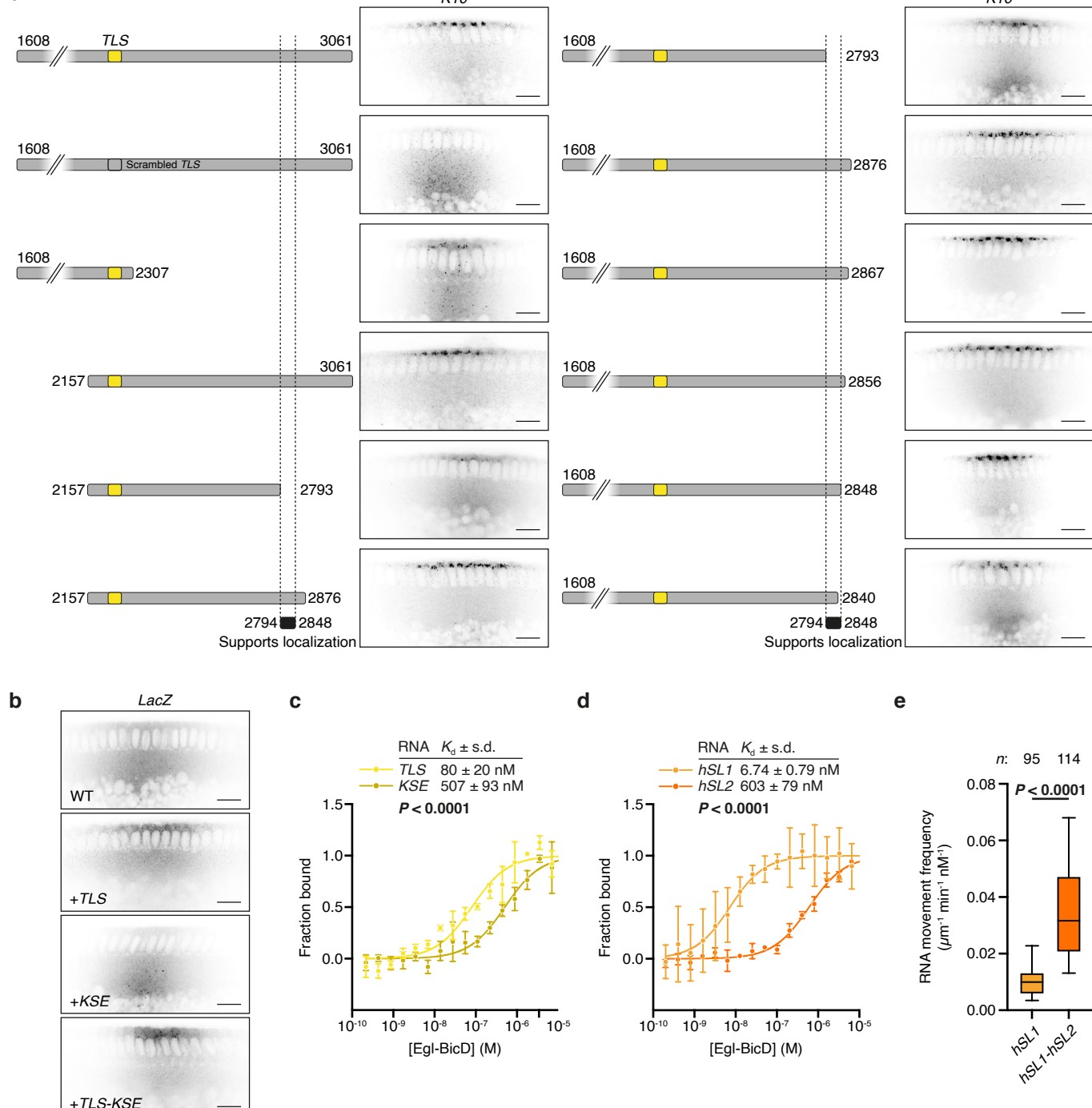

**Extended Data Fig. 8 | Cis-acting support elements potentiate RNA localization and associate with Egl–BicD. a** and **b**, Confocal images of *Drosophila* embryos injected with the indicated fluorescently-labeled fragments of the *K10* 3′ UTR (**a**) or *LacZ* transcripts bearing different combinations of the *TLS* and *KSE* (**b**). Images are representative of 22–85 injected embryos as specified in Fig. 5d and f. **c** and **d**, MST binding curves for Egl–BicD bound to the *TLS* and *KSE* (**c**) or *hSL1* and *hSL2* (**d**). Data for *hSL1* binding to Egl–BicD are reproduced from Extended Data Fig. 6b for comparison. Data points show the mean ± s.d. for three independent

experiments per condition, from which best-fit values for $K_d$ ± s.d. were derived. **e**, Frequency of dynein-driven RNA movements in reconstituted transport complexes with indicated transcripts. Boxes show interquartile range, with horizontal line denoting median, and whiskers show 10th–90th percentile range. 95–114 microtubules (*n*) were analyzed from three independent experiments representing 266–493 individual complexes per condition. Bold *P* values denote statistical significance, determined using a one-tailed extra sum-of-squares F test (**c** and **d**) or pairwise two-tailed Mann-Whitney test (**e**).

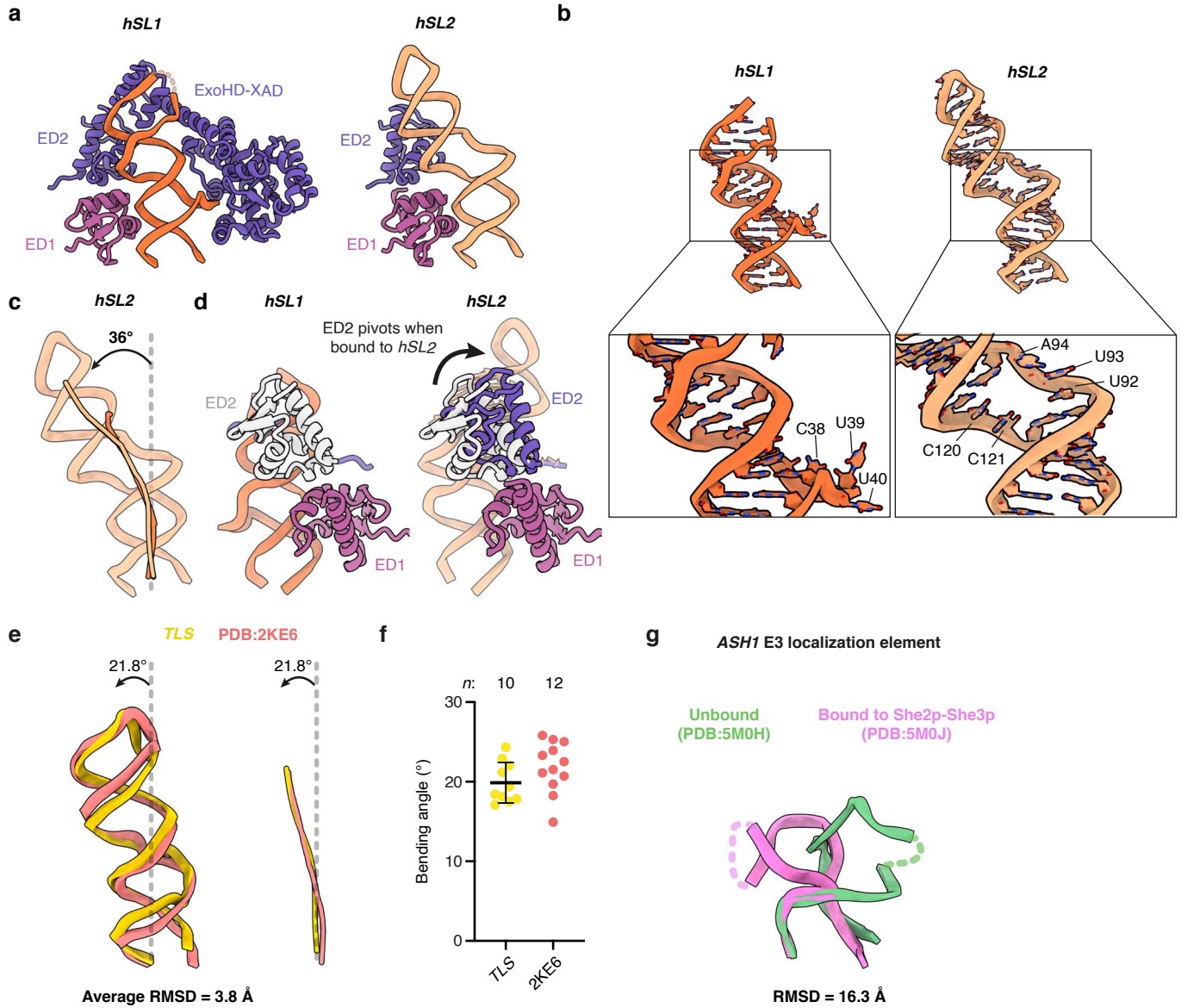

**Extended Data Fig. 9 | Structural comparison of *hSL1* and *hSL2* stem loops, and of isolated and RBP-bound localization signals. a**, Comparison of Egl's RNA-binding domains bound to *hSL1* or *hSL2* in the structure of Egl–BicD in complex with the *hairy* localization element. ExoHD–XAD is only bound to *hSL1*. **b**, Close-up views of the bulged nucleotides in *hSL1* and *hSL2*. The broader bulge in *hSL2* is formed by multiple consecutive unpaired residues and results in a wider major groove. **c**, *hSL2* RNA stem-loop structure with a measured bend of 36° between the upper and lower helices. The helical trajectory of *hSL1* (orange) is shown for comparison. **d**, ED2 pivots about its interface with ED1 to accommodate the altered RNA geometry of *hSL2* relative to *hSL1*. The magnitude of this pivot is shown by overlaying ED2 from the *hSL1*-bound structure (gray) onto ED2 in the *hSL2*-bound structure (purple). **e**, Comparison of the bending angle between the upper and lower helices of the *TLS* stem loop in its unbound state (solution NMR structure PDB: 2KE6[52]; coral) and when bound to the Egl–BicD complex (yellow). RNA stem loops are shown as cartoons (left), with the

helical trajectories (right), calculated using Curves + , are represented as lines. The average RMSD of the ensemble of NMR conformers relative to the Egl-bound structure and their average bending angle are indicated. **f**, Quantification of *TLS* bending angles measured from cryo-EM structures of different conformations of the Egl–BicD–*TLS* complex (yellow circles) and across individual conformers from the NMR ensemble of *TLS* (PDB: 2KE6) (red circles). For ease of comparison, *TLS* angles measured in cryo-EM structures are reproduced from Extended Data Fig. 5c and show mean ± s.d. As each conformer in the NMR ensemble of 2KE6 is derived from the same dataset, the mean ± s.d. is not applicable to this condition. The total number of localization signal models (EM) or conformers (NMR) used (*n*) is indicated above each condition. **g**, Structural comparison of the *ASH1* E3 localization element in its free state (PDB: 5M0H ref. 58; green) and when bound to the She2p–She3p complex (PDB: 5M0J[58]; pink). RNA stem loops are depicted as cartoons, and the RMSD between the free and bound conformations is indicated.

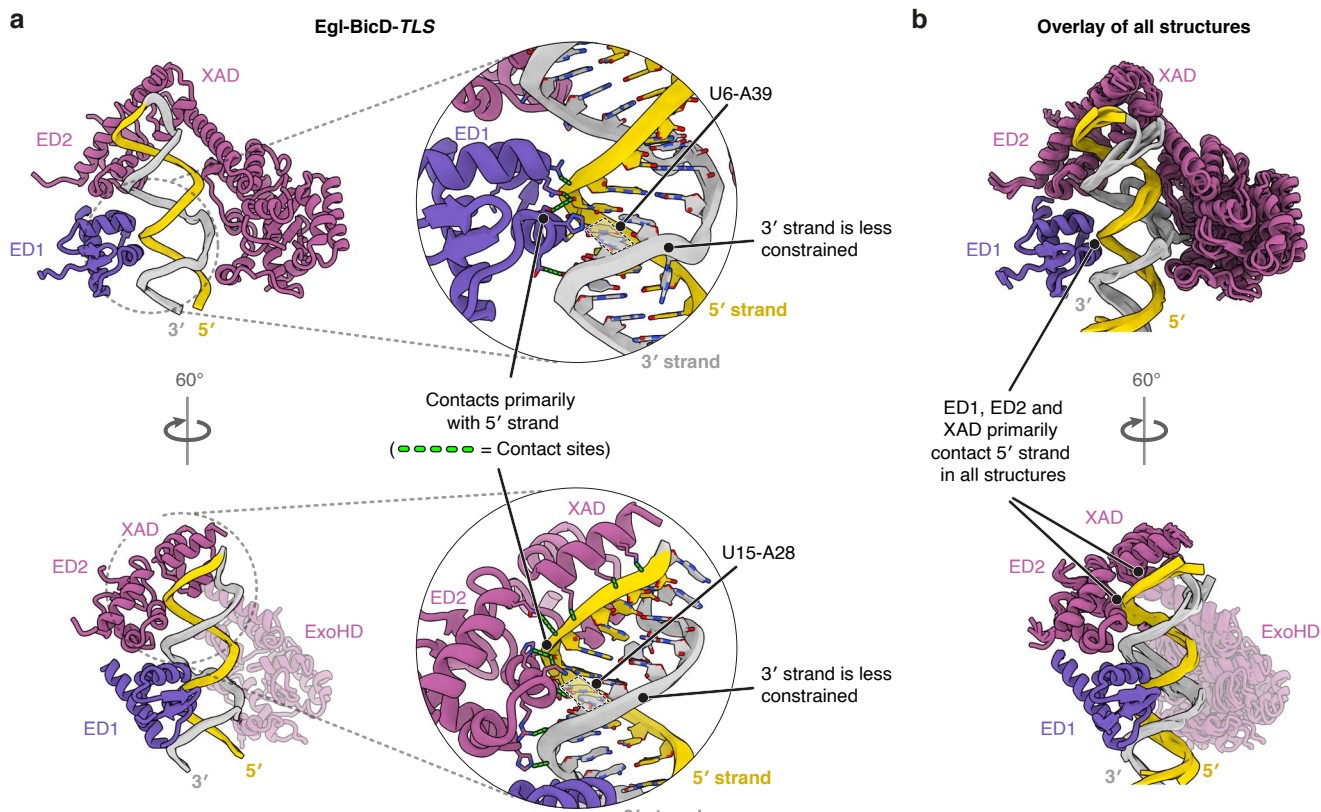

**a** Egl-BicD-*TLS*

**b** Overlay of all structures

**Extended Data Fig. 10 | ED1, ED2 and XAD preferentially engage the 5′ strand of localization signals. a**, Structure of the Egl–BicD–*TLS* complex showing interactions between distinct domains of Egl with the 5′ strand (yellow) and 3′ strand (gray) of the *TLS* RNA. Zoomed-in views highlight contacts between ED1 (top) and ED2 (bottom) with the RNA, showing that interactions are predominantly made with the 5′ strand, while the 3′ strand is comparatively less constrained. Contact sites are indicated by green dashed marks. Nucleotide positions U6–A39 and U15–A28 are labeled to indicate the base pairs whose sequences can be distinguished by Egl. Notably, the 3′ bases at these positions lie within the less constrained region of the *TLS*. **b**, Overlay of all determined structures illustrating that ED1 and ED2 consistently contact the 5′ strand across structures.

| | Corresponding author(s): | Mark McClintock, Andrew Carter and Simon Bullock |
| --- | --- | --- |
| | Last updated by author(s): | Mar 20, 2026 |

# Reporting Summary

## Statistics

For all statistical analyses, confirm that the following items are present in the figure legend, table legend, main text, or Methods section.

| n/a | Confirmed | |
| --- | --- | --- |
| ☐ | ☒ | The exact sample size (*n*) for each experimental group/condition, given as a discrete number and unit of measurement |
| ☐ | ☒ | A statement on whether measurements were taken from distinct samples or whether the same sample was measured repeatedly |
| ☐ | ☒ | The statistical test(s) used AND whether they are one- or two-sided *Only common tests should be described solely by name; describe more complex techniques in the Methods section.* |
| ☒ | ☐ | A description of all covariates tested |
| ☐ | ☒ | A description of any assumptions or corrections, such as tests of normality and adjustment for multiple comparisons |
| ☐ | ☒ | A full description of the statistical parameters including central tendency (e.g. means) or other basic estimates (e.g. regression coefficient) AND variation (e.g. standard deviation) or associated estimates of uncertainty (e.g. confidence intervals) |
| ☐ | ☒ | For null hypothesis testing, the test statistic (e.g. *F*, *t*, *r*) with confidence intervals, effect sizes, degrees of freedom and *P* value noted *Give P values as exact values whenever suitable.* |
| ☒ | ☐ | For Bayesian analysis, information on the choice of priors and Markov chain Monte Carlo settings |
| ☒ | ☐ | For hierarchical and complex designs, identification of the appropriate level for tests and full reporting of outcomes |
| ☒ | ☐ | Estimates of effect sizes (e.g. Cohen's *d*, Pearson's *r*), indicating how they were calculated |

*Our web collection on statistics for biologists contains articles on many of the points above.*

## Software and code

Policy information about availability of computer code

| Data collection | Zeiss ZEN 2.1 (confocal microscope image acquisition software-version 14.0.0.201); MicroManager (TIRF microscope image acquisition software version-1.4.22); MO.control (MST acquisition software-version 2.5.4); AlphaFold2; ThermoFisher EPU (CryoEM acquisition software-version 3.13) |
| --- | --- |
| Data analysis | FIJI v2.16.0/1.54p; RNAfold v2.6.3; mfold v3; Curves+ v2.6; ChimeraX v1.10; GraphPad Prism v10.4.0; ISOLDE v1.10.1; COOT v1.1.19; PHENIX v2.0; RELION (4.0 and 5.0); Topaz v0.2.5; CryoSPARC v5; MO.Affinity analysis v3.0.5. Custom code for analysis of RNA bending angles is available at https://github.com/carterlablmb/RNA_bend_analysis |

For manuscripts utilizing custom algorithms or software that are central to the research but not yet described in published literature, software must be made available to editors and reviewers. We strongly encourage code deposition in a community repository (e.g. GitHub). See the Nature Portfolio guidelines for submitting code & software for further information.

# Data

Policy information about availability of data

All manuscripts must include a data availability statement. This statement should provide the following information, where applicable:
- Accession codes, unique identifiers, or web links for publicly available datasets
- A description of any restrictions on data availability
- For clinical datasets or third party data, please ensure that the statement adheres to our policy

Data availability
Atomic coordinates and cryo-EM maps have been deposited in the Protein Data Bank (PDB) or Electron Microscopy Data Bank (EMDB), respectively, under accession codes 9RVY and 54292 (Egl–BicD–TLS structure A), 9RVZ and 54293 (Egl–BicD–TLS structure B), 9RW0 and 54294 (Egl–BicD–TLS structure C), 9RW1 and 54295 (Egl–BicD–TLS structure D), 9RW2 and 54296 (Egl–BicD–TLS structure E), 9RW3 and 54297 (Egl–BicD–hSL1 structure A), 9RW4 and 54298 (Egl–BicD–hSL1 structure B), 9RW5 and 54299 (Egl–BicD–hSL1 structure C), 9RW6 and 54300 (Egl–BicD–ILS structure A), 9RW7 and 54302 (Egl–BicD–bcdSLV structure A), 9RW8 and 54303 (Egl–BicD–bcdSLV structure B), 9RW9 and 54304 (Egl–BicD–bcdSLV structure C), 9RWA and 54305 (Egl–BicD–GLS), 9RWB and 54306 (Egl–BicD–hSL1–hSL2). For Egl–BicD–ILS structure B, only the map was deposited in EMDB under the accession code 54301. Numerical source data for all plots are included with this paper. All other data, including large imaging datasets, are available upon reasonable request to the corresponding author.

# Research involving human participants, their data, or biological material

Policy information about studies with human participants or human data. See also policy information about sex, gender (identity/presentation), and sexual orientation and race, ethnicity and racism.

| | |
|---|---|
| Reporting on sex and gender | Not applicable |
| Reporting on race, ethnicity, or other socially relevant groupings | Not applicable |
| Population characteristics | Not applicable |
| Recruitment | Not applicable |
| Ethics oversight | Not applicable |

Note that full information on the approval of the study protocol must also be provided in the manuscript.

# Field-specific reporting

Please select the one below that is the best fit for your research. If you are not sure, read the appropriate sections before making your selection.

☒ Life sciences  ☐ Behavioural & social sciences  ☐ Ecological, evolutionary & environmental sciences

For a reference copy of the document with all sections, see nature.com/documents/nr-reporting-summary-flat.pdf

# Life sciences study design

All studies must disclose on these points even when the disclosure is negative.

| | |
|---|---|
| Sample size | Sample sizes for Drosophila experiments were chosen based on the number of eggs that can be collected and processed in a cohort before aging beyond the blastoderm stage and precedents set in the literature, which capture reproducible differences in localization efficiency of RNAs. For single molecule data, the number of individual observations was determined by the number of microtubule-bound complexes per chamber, which were typically on the order of several hundred. These sample sizes are comparable to or larger than similar studies in the literature. |
| Data exclusions | Embryos that were clearly damaged at the injection site by the needle were not analyzed. |
| Replication | All results were reproducible in multiple replicates per experiment. RNA injections were replicated in 2 to 6 independent experiments. Pilot experiments showed that localization efficiency for a given RNA was consistent across different experiments. Single molecule observations were consistently made in 3 or 4 independent experiments. Affinity constants were determined by 3-6 independent MST measurements and no significant variability was observed in the affinities of different protein preparations for their RNA targets. |
| Randomization | Not applicable. All embryos from a given cohort were processed for injection with the same RNA. |
| Blinding | The experimenter performing injections was blind to the identity of the RNA being studied in order to eliminate unconscious bias |

# Reporting for specific materials, systems and methods

We require information from authors about some types of materials, experimental systems and methods used in many studies. Here, indicate whether each material, system or method listed is relevant to your study. If you are not sure if a list item applies to your research, read the appropriate section before selecting a response.

## Materials & experimental systems

| n/a | Involved in the study |
|-----|------------------------|
| ☒ | ☐ Antibodies |
| ☐ | ☒ Eukaryotic cell lines |
| ☒ | ☐ Palaeontology and archaeology |
| ☐ | ☒ Animals and other organisms |
| ☒ | ☐ Clinical data |
| ☒ | ☐ Dual use research of concern |
| ☒ | ☐ Plants |

## Methods

| n/a | Involved in the study |
|-----|------------------------|
| ☒ | ☐ ChIP-seq |
| ☒ | ☐ Flow cytometry |
| ☒ | ☐ MRI-based neuroimaging |

## Eukaryotic cell lines

Policy information about cell lines and Sex and Gender in Research

| | |
|---|---|
| Cell line source(s) | Sf9 (ECACC 89070101) derived from pupal ovarian tissue of the fall armyworm Spodoptera frugiperda. Commercially sourced from ThermoFisher Scientific |
| Authentication | Cell line was not authenticated by DNA analysis but had the stereotypical appearance of Sf9 cells |
| Mycoplasma contamination | Sf9 insect cells were routinely confirmed as Mycoplasma-free using the MycoALERT kit (Lonza). |
| Commonly misidentified lines (See ICLAC register) | None |

## Animals and other research organisms

Policy information about studies involving animals; ARRIVE guidelines recommended for reporting animal research, and Sex and Gender in Research

| | |
|---|---|
| Laboratory animals | Drosophila melanogaster (w1118 strain) |
| Wild animals | Not applicable |
| Reporting on sex | The sex of Drosophila embryos cannot be determined at the age we studied. However, approximately equal numbers of male and female Drosophila embryos would have been injected with RNA in each experiment |
| Field-collected samples | Not applicable |
| Ethics oversight | No ethical approval was required for this work |

Note that full information on the approval of the study protocol must also be provided in the manuscript.

## Plants

| | |
|---|---|
| Seed stocks | Not applicable - no plants used |
| Novel plant genotypes | Not applicable |
| Authentication | Not applicable |

