## [Peer Review File · Nature Structural & Molecular Biology]

Structural basis for recognition of diverse localizing mRNAs by Egl–BicD

Corresponding Author: Dr Simon Bullock

Version 0:

Decision Letter:

27th Aug 2025

Dear Dr. Bullock,

Thank you again for submitting your manuscript "Cryo-EM structures of Egl–BicD-RNA complexes reveal how diverse mRNAs are selected for subcellular localization". I apologise for the delay in sharing our decision. I am writing to let you know that we have decided to send your manuscript for peer review, but several points require your attention before we can proceed with peer review.

I am re-opening the manuscript submission system for you to resubmit your manuscript with all associated files needed for peer review directly, within 2-3 business days if possible. Please follow the link at the bottom of this email to upload the documents listed below. The following instructions pertain our general guidelines: in your case I think that providing maps, models, PDB reports and ensuring that the Reporting Summary is accurate are the guidelines that apply. If you have any issues, please reach out to us before completing the submission.

1- We require official wwPDB validation reports for newly described atomic structures, as per journal policy. We also request that authors provide cryo-EM maps, half-maps and models, as well as maps and models obtained from subtomogram averaging if it applies to the work, to help the reviewers in assessing the work. We recommend the use of figshare in our system, which allows for provision of anonymous access links for the referees (<https://www.springernature.com/gp/authors/research-data/figshare-integration>). Alternatively, please upload .zip folders directly with the submission. To ensure the ease of reviewer access to the data, please specify in the Data Availability section where the files can be found (i.e., provide a figshare link or direct the reader to the manuscript files).

2- We want to ensure that the methods and statistics reporting in our papers are of the highest quality. To that end, we ask authors to fill out a Reporting Summary that collects information on experimental design and reagents. If your paper includes ChIP-seq, flow cytometry data, we ask you take special care to complete those sections of the Reporting Summary as this data will aid greatly in the review of your manuscript. This document can be found by following the link below:

Reporting Summary:

3- In order for us to proceed with peer review, please provide accession numbers and reviewer tokens to access sequencing or proteomics datasets if any unpublished datasets are part of your study. Please add this information to your manuscript file.

4- Lastly, I would like to kindly request that you provide the code used to analyse the data to the reviewers, if newly developed (unpublished) code was used in the work. For the reviewers to evaluate the work adequately, they must be able to test the software/review the code themselves. If you have not yet provided the software, we therefore request that you provide a single compressed zip file containing the software with a readme.txt file or other user manual containing complete instructions for installing and running the software. If appropriate, please provide example data and expected output. Sufficient material should be provided for referees to directly test the performance of the software/algorithm. If the software and materials are small enough to fit in a single compressed zip file under 6MB in size, you may email this file directly to me. If the zip file is between 6 MB and 200 MB, you may upload it to our file transfer site. If necessary, a second zip file up to 200 MB in size can be used to supply the example data. Please let me know if you need to use this option and I'll send you further details. Alternatively, you can upload the code to GitHub and provide us with the link.

Please fill out and return to me the code and software submission checklist that will be made available to editors and

reviewers during manuscript assessment. Please note that this form is a dynamic 'smart pdf' and must therefore be downloaded and completed in Adobe Reader, instead of opening it in a web browser.

<https://www.nature.com/documents/nr-software-policy.pdf>

Please use the link below to submit the files. **Please also remember to move forward all other files associated with this version of the paper.**

Link Redacted

Sincerely,

Dimitris Typas
Senior Editor
Nature Structural & Molecular Biology
ORCID: 0000-0002-8737-1319

Version 1:

Decision Letter:

23rd Oct 2025

Dear Dr. Bullock,

Thank you again for submitting your manuscript "Cryo-EM structures of Egl-BicD-RNA complexes reveal how diverse mRNAs are selected for subcellular localization". I apologize for the delay in responding, which resulted from the difficulty in obtaining suitable referee reports. We now have comments (below) from the 4 reviewers who evaluated your paper. Please note that Reviewers #3 and #4 reviewed the manuscript together. In light of these reports, we remain interested in your study and would like to see your response to the comments of the referees in the form of a revised manuscript.

You will see that the reviewers appreciate the conceptual impact of the study, yet request important experimental improvements, as well as much need clarifications and better data presentation, in the form of a revised manuscript. In particular, co-reviewing Reviewers #3 and #4 assess that further mutagenesis data and bioinformatic analyses must be provided to substantiate the proposed guanine rejection mechanism; they also request the addition of relevant missing controls. Along similar conceptual lines, with respect to showing that the identified properties are necessary and sufficient for identifying specific RNA structures, Reviewer #2 proposes that in silico designed RNAs, based on the molecular rules obtained in the study, are shown to localise in fly embryos according to the underlying rules. We think that the combination of the mutagenesis and in silico design experiments would notably increase the impact of the study and strongly support the proposed model. On the other hand, Reviewer #1 requests that further insight and discussion in the affinity of the Egl-BicD complex for the full-length RNA and its specific elements is provided. This expert also asks for certain clarifications, potentially improved presentation of the data and expanding the discussion; co-reviewing reviewers R#3 and #4 echo this concern. Reviewer #2 notes the need for phylogenetic analyses to check the potential conservation of the primary localisation signals and additional discussion of both oskar mRNA trafficking and potential remodelling of the 3'UTR architecture.

Our standard revision period is 4 months. We appreciate that the requested revisions are extensive. We are committed to providing a fair and constructive peer-review process and encourage you to contact us if you would like to discuss the reviews or anticipate any delay with the resubmission.

We will be happy to consider your revision as long as nothing similar has been accepted for publication at NSMB or published elsewhere. Should your manuscript be substantially delayed without notifying us in advance and your article is eventually published, the received date would be that of the revised, not the original, version.

REPORTING AND DATA AVAILABILITY: Please provide updated files as follows with your revision. As you already know, we put great emphasis on ensuring that the methods and statistics reported in our papers are correct and accurate.

1. **REPORTING SUMMARY:** Please provide an updated file:

This form is a dynamic 'smart pdf' and must be downloaded and completed in Adobe Reader.

2. REPORTING OF STRUCTURAL DATA: If there are additional or modified structures in the revision, please submit the corresponding PDB validation reports with your other files, and use the figshare integration system to provide access to maps, half-maps, and models.

Manuscripts reporting new structures should contain a table summarizing structural and refinement statistics. Please include these tables for new cryo-EM or -ET (<https://www.nature.com/documents/nr-tables-cryo-em.doc> and modifying this file for ET), NMR (<https://www.nature.com/documents/nr-tables-nmr.doc>) and X-ray crystallography (<https://www.nature.com/documents/nr-tables-xray.doc>). To facilitate assessment of the quality of the structural data, a stereo image of a portion of the electron density map (for crystallography papers) or of the superimposed lowest energy structures (>10; for NMR papers) should be provided with the submitted manuscript. If the reported structure represents a novel overall fold, a stereo image of the entire structure (as a backbone trace) should also be provided. For cryo-EM structures, a representative micrograph showing individual particles should be provided in the figures alongside a processing workflow (please see our Editorial for more information: <https://www.nature.com/articles/s41594-025-01567-9>).

DATA DEPOSITION: We require deposition of coordinates (and, in the case of crystal structures, structure factors) into the Protein Data Bank with the designation of immediate release upon publication (HPUB). Electron microscopy-derived density maps and coordinate data must be deposited in EMDB and released upon publication. Deposition and immediate release of NMR chemical shift assignments are requested. To avoid delays in publication, dataset accession numbers must be supplied with the revised manuscript and appropriate release dates must be indicated at the galley proof stage.

3. REPORTING OF LIGHT MICROSCOPY DATA: For any revision that includes light microscopy data, we ask our authors to please include a completed light microscopy reporting table [https://www.nature.com/documents/Light_microscopy_reporting_table.xlsx] to ensure the methods are described thoroughly. The table will be available to reviewers and ultimately published, should the manuscript be accepted at the journal. When submitting the revised version of your manuscript, please pay close attention to our [Digital Image Integrity Guidelines](https://www.nature.com/nature-portfolio/editorial-policies/image-integrity).

4. CODE AND COMPUTATIONAL WORK: If newly developed, unpublished code was used in this work, it must be provided with the revision and access to it must be disclosed in the Code Availability Statement in the manuscript. Please also provide the completed software submission checklist. This form is a dynamic 'smart pdf' and must be downloaded and completed in Adobe Reader, instead of opening it in a web browser: <https://www.nature.com/documents/nr-software-policy.pdf>

If molecular dynamics (MD) simulations were performed, please refer to this Editorial and provide a completed MD simulations checklist with your revision: <https://www.nature.com/articles/s42003-023-04653-0>

Lastly, if it applies, please complete and upload the completed machine learning checklist with your submission for review: <https://www.nature.com/documents/machine-learning-checklist.pdf>

5. SOURCE GEL IMAGES: Unprocessed scans should be provided for all gels and western blots presented in figures and should be clearly labelled with the figure number in the file title. The source gel images should be uncropped, unmodified images, with molecular weight markers. Uncropped, unmodified gel images from existing experimental repeats, in addition to the data shown in the figures, can also be presented in source data. Please provide these files as PDF with your revision.

Please ensure that all control panels for gels and western blots are appropriately described as loading or IP controls. Lastly, please ensure that all images in the paper are checked for duplication of panels and for splicing of gel lanes.

6. SOURCE NUMERICAL DATA: we urge authors to provide, in tabular form in excel files, the data underlying the graphical representations used in figures. This is to further increase transparency in data reporting, as detailed in this editorial (<http://www.nature.com/nsmb/journal/v22/n10/full/nsmb.3110.html>). Please provide one excel file per figure max, with one panel per tab. When submitting files, the title field should indicate which figure the source data pertains to.

7. DATA AVAILABILITY: this journal strongly supports public availability of data. All data used in accepted papers should be available via a public data repository, or alternatively, as Supplementary Information. If data can only be shared on request, please explain why in your Data Availability Statement and in the correspondence with your editor.

For some data types, deposition in a public repository is mandatory as detailed below:

<https://www.nature.com/nature-research/editorial-policies/reporting-standards#availability-of-data>

EXTENDED DATA FIGURES

Link Redacted

Sincerely,

Dimitris Typas
Senior Editor
Nature Structural & Molecular Biology
ORCID: 0000-0002-8737-1319

Reviewers' Comments:

Reviewer #1 (Remarks to the Author):

In this manuscript, Singh and co-workers present the molecular basis for functional mRNA recognition by the *Drosophila* protein Egalitarian (Egl). Egalitarian is an essential component of the mRNA transport machinery, to which provides selectivity via its recognition of structured RNA elements within the target mRNAs. However, how Egalitarian would recognize RNA structures that have been predicted to be different, discriminating RNA targets from non-targets, is a key question we need to answer to establish the molecular bases of local mRNA translation during *Drosophila* development. A second important question is whether/how multiple recognition elements on the same RNA may be required. Here, the authors present a comparative analysis of the 2:1 Egalitarian-BicD protein-protein complex bound to five known RNA targets. They identify the main interactions responsible for recognition and link them to common features of the RNA targets, probing their conclusions by mutating protein and RNAs and testing the effect on i) the affinity of the interactions with the Egl-BicD complex and ii) mRNA localization in the *Drosophila* embryo. The results show that the protein makes contact with different elements in the RNA structure, including two minor grooves and a loop, as well as specifically recognizing individual bases which are precisely positioned in the complex. The recognition of multiple features within the RNA structure is mediated by different protein domains. Interestingly, to achieve this recognition two Egl chains are required, with a domain switch that lead one RNA element to be bound by three domains from one chain and one domain from the other. The complex web of inter-domain contacts imposes a structure selectivity that matches the RNA's conformation. Importantly, some flexibility exist: the domains can rotate and re-orient while maintain contact, which facilitate the recognition of non-identical RNA structures. Finally, the Egl-BicD complex can interact simultaneously with two RNA elements within the same RNA molecule and, using the structural and biophysical tools discussed above, the authors explore how simultaneous binding of a primary and a secondary recognition element favors recognition and transport.

The paper is clear and well organized. The figures are very well drafted and the analysis informative. In general, the conclusions are very well supported by the data, which are copious. study provides an initial answer to some of the key questions in the field of mRNA transport and localization. It is a high-quality paper that will have a substantial impact in the field. Below are a few minor points that should be addressed prior to publication.

1) The affinity of the protein complex for the different RNA elements (e.g. TLS vs hSL1) seems to be quite different. Could the

authors discuss the mechanistic implications of this?

2) Perhaps I have missed this, but have the authors measured the affinity of the Hsl1-Hsl2 two domain RNA element for the protein complex? They present a structure of the protein-RNA complex, and we have the affinity for the two separate elements, but I do not see the affinity for the full-length RNA. This would provide some mechanistic information on the extent to which cooperativity/coupling plays a direct role in the *in vivo* effect.

3) It would be very useful to see a sequence alignment of the different regions of the protein. Are the interfaces between the domains conserved? Are the linkers conserved? Such an alignment is important, can it please be added (and briefly discussed) if not there?

4) 17b and comparison with the free RNA structure: it is great to see the spread of the angle in the family of NMR structures, which is the best way of representing this. Could the authors please remove the error bars, as each of these structures is consistent with the same data ensemble? If the author wants to indicate the value of the angle in the average structure this could be done by a dot of a different colour, although having just the ensemble of structures is probably the best representation. Perhaps the author could report the angles of all the cryoEM and NMR structures side-by-side, in one panel? Regardless, it seems to me there is no meaningful difference between the angle in the NMR structures and the cryoEM one? If that is the case, please reflect this in the main text.

5) Lines 130-133. The fact that interface between domains are not predicted by AlphaFold it is possibly due to their uniqueness, I am not sure the authors need to use the AlphaFold result to support cooperativity. I would remove this comment.

6) In general, I find the discussion is very well written, but some points could be explored more in depth. For example, can the authors comment on other such elements that they have not looked at experimentally (they mention a few) and predict the structure? Can they predict more? I understand structure predictions have not always been accurate, but can the 'rules' they have uncovered help? I think some general comment on this would be interesting. Also, why do we need two separate RNA elements to recognize the RNA – the comment on recycling the transport system is a little brief. Could they explain more?

Reviewer #2 (Remarks to the Author):

Subcellular targeting of selected mRNAs was discovered decades ago and since then shown to underlie cell compartmentalization in various cell types. Dissection of the mechanisms driving the localization of few specific RNAs led to the identification of « zipcodes », ie cis-regulatory RNA elements that are both necessary and sufficient for localization. These elements further mediate the binding of RNA binding proteins and the subsequent recruitment of transport machineries. Although large-scale studies have identified populations of mRNAs that co-localize within specific compartments in various cell types, the number of examples where the molecular determinants of zipcode recognition have been resolved remains very limited. Both experimental and computational analyses have so far failed to reveal common RNA features that account for the localization of RNAs to shared cellular destinations. Thus, the field still lacks a unifying understanding of the rules governing the recognition of sets of co-localizing RNAs.

In this study, the authors present a systematic and high-quality characterization of the cryo-EM structure of the Egl-BicD complex bound to 5 different RNA elements, each known to mediate dynein-dependent localization of its corresponding RNAs in *Drosophila*. This system is a model of choice for a number of reasons: i- the RNA localization elements had been previously mapped, with established association with Egl-BicD, ii- these RNA elements lack obvious sequence or structural commonalities, and iii- structural insights can be validated through functional *in vitro* and *in vivo* assays.

Together, this study stands out in several respects: first, it provides high-quality cryo-EM structures of the Egl-BicD complex bound to 5 distinct RNA elements previously shown to mediate Egl-BicD-dependent localization. This substantial experimental effort, combined with a careful integrated analysis, led to the identification of key shared (but previously unrecognized) determinants required for localization. Second, the work combines high-quality cryo-EM and quantitative *in vitro* binding assays with both single-molecule live-imaging in reconstituted systems and RNA localization assays *in vivo* in early *Drosophila* embryos. This impressive array of advanced methods allowed the authors not only to dissect the molecular contacts governing RNA-protein interactions, but also to demonstrate their functional relevance in the assembly of transport-competent complexes. Third, the new findings shed light on previously reported results, enabling the integration of years of work from different groups into a coherent and elegant mechanistic framework. Last, the quality and clarity of the manuscript, including its text, figures and associated legends were outstanding, leaving little room for meaningful improvement.

Given the outstanding quality of this work, my comments are only minimal:

Major points:

- The authors appear to have uncovered the code governing the localization of mRNAs that depend on the Egl-BicD complex for their dynein-mediated transport. If this is true, an important final step would be to design *in silico* synthetic RNAs predicted to undergo efficient apical localization when injected in the *Drosophila* embryos. Including such a dataset would provide compelling evidence that the Egl-BicD recognition and localization can be fully predicted.

- Another well-characterized localization element that requires Egl-BicD for its dynein-dependent localization is the OES of oskar mRNA (Jambor et al., 2014; doi: 10.1261/na.041566.113.). Could the authors discuss how the OES fits with their

proposed unifying model of recognition ?

Minor points

- Abstract : "we present cryo-EM structures of Egl-BicD in complex with 6 different RNAs" : I counted 5, not 6

- Discussion: could the authors comment on how co-binding to stem loops that are hundreds of nucleotides apart within the same RNA molecule might modulate 3'UTR architecture ?

- Have the authors examined the phylogenetic conservation of primary localization signals versus support elements (through simple sequence alignments across divergent species)

Reviewer #3 (Remarks to the Author):

This manuscript by Singh, Chilaeva, and McClintock et al., reports high-resolution Cryo-EM structures of the *Drosophila* Egl-BicD adaptor complex bound to six functionally distinct, developmentally important localization signal-containing RNAs. The structural dissection of non-canonical dsRNA recognition by four Egl domains (ED1, ED2, ExoHD, and XAD) reveals a novel mechanism of cooperative binding and domain swap by flexibly tethered domains. These structures converge on the recognition of a ~20° bent stem loop consisting of an upper stem and lower stem separated by a bulge in the 3' strand. Using biophysical assays (FP, MST) and in vivo and in vitro RNA motility assays, the authors corroborate their structural findings with mutational analyses, provide support for a dual stem-loop recognition model that licenses transport, and identify a new secondary localization signal (KSE) in the K10 mRNA. The authors also test the sequence identity requirements and argue for a steric occlusion model where Cys35 or Gly124 reject guanines thereby imposing a U-A sequence preference. The manuscript is well-organized, clearly written and expertly illustrated. The work addresses an important question in cell and developmental biology — how specific mRNAs are selected for intracellular transport. It further provides substantial novel insights into RNA recognition by multi-domain RNA-binding proteins, and is thus a strong candidate for NSMB. However, there are several major concerns that should be addressed with additional experiments before publication, and some minor suggestions below intended for enhancing accuracy, clarity and readability of the manuscript.

Specific comments:

1. The most significant concern that I have is the incomplete mutagenesis data in support of the proposed guanine rejection mechanism by steric occlusion. The argument based on structural modeling, especially of the ion or water density close to Gly124/Ser127 clashing with the 2-amino of guanine, is strenuous (Fig. 3, ExFig. 13). All the examples in this study seem to have 5'U-3'A pairs instead of 5'A-3'U pairs. Why is it so? Is it due to the purine N3 interaction? The steric occlusion model does not address this sequence preference. Distances (or extents of overlap) are not provided in support of the proposed clashes. Some of the densities don't look great (e.g. ExFig. 13g), likely due to the inherent noise in the cryo-EM images as expected. If a guanine does bind in this location, there simply wouldn't be space to accommodate the loosely tethered ion/water, not that the ion/water here would necessarily prevent the guanine from binding due to a tightly bound structural role. As far as I can tell, the authors only tested the U-A to G-C pair, which is a fairly drastic transversion mutation. I suggest the authors also test A-U, C-G, and even U-G wobble pairs. The U-A to A-U swap will test if the N3-recognition by His31/Ser127 is functionally important, and specifies the purine preference in this strand. The U-A to C-G will maintain the N3 interaction to the purine, while not creating potentially disruptive transversions. One might expect this one to bind okay. A U-A to U-G substitution will not only maintain the purine N3 interaction, but also shift the U6/U15 up and away from the Cys35/Gly124 ion/water to preempt any steric clash. Notably, the G-U wobble pair would also alter the twist of the dsRNA duplex, which could also be important for binding.

2. Related to #1, is there bioinformatics information available or attainable for the sequence conservation or co-variation at these binding sites? Such information would help clarify whether there is indeed guanine avoidance here and if so whether the avoidance applies to both strands of the dsRNA stem. On the protein side, are Cys35, Gly124, Ser127 conserved? If Cys35 is substituted by a less bulky side chain, say in another species, would the authors expect a loss or reduction of sequence specificity of RNA transport?

3. The authors demonstrate strong binding and provide high-resolution structures for relevant cargo RNAs. To fully establish the principle of specific cargo selection, the manuscript should consistently include negative controls using one or more unrelated, non-localized RNA of similar size and/or secondary structure for their assays, such as the MST data using an unrelated straight dsRNA in ExFig. 11e.

4. It would be informative to mention whether attempts were made to determine cryo-EM structures of Egl/BicD proteins without RNA since their interactions don't require RNA, or low-affinity, non-cargo RNA complexes. If such data exist, the authors could comment on whether 3D classification revealed conformational changes, empty RNA-binding sites, or significant flexibility at the interface, which would shed light on the resting state of the proteins or structural consequences of non-recognition.

5. For the hairy RNA complex, if feasible please provide additional binding assays to verify that the ExoHD and XAD domains are not interacting with the low-affinity element hSL2, thereby strengthening the conclusion regarding the selective engagement of domains with each stem loop.

6. ExFig. 11a, the author should label the domains and potentially also show where the RNA would be in this uncommon viewing angle, which could be translucent so as not to obscure the view of the protein motions.

7. Line 225. The authors state "whereas His31 makes a base-agnostic hydrogen bond to help lock the domain in place". It is not clear that this interaction is truly sequence non-specific without direct testing. The structures only show His31 binding to N3 of purine when in a Watson-Crick pair and distorted O2 within a U48xC9 mismatch. It is not known if a cytosine here, or a uridine in a canonical base pair would bind His31 with similar strengths as a purine N3, which seems more favorable. Further, the O2 groups on cytosine and uridines are not chemically equivalent.

8. ExFig. 16e. The numerous arrows obscure the view of the new location/confirmation of ED2. It is likely better to show an overlay instead of using the strong arrows in this panel.

9. Fig. 2f and ExFig. 10, the authors use three shades of pink/magenta/purple as the sole distinguishing feature for the sets of data points. More contrasting colors are likely needed.

10. Several further points regarding manuscript presentation below could be addressed to enhance clarity and completeness: a) The authors could include representative gel images (e.g., SDS-PAGE) to demonstrate the purity of all protein constructs used in the binding and structural experiments. b) Extended Data Fig. 4e and 4f do not appear to be cited in the main text. c) The authors should consider including the Lotus domain in the protein sequence alignment shown in ExFig. 1b.

Reviewer #4 (Remarks to the Author):

I co-reviewed this manuscript with one of the reviewers who provided the listed reports. This is part of the Nature Structural & Molecular Biology initiative to facilitate training in peer review and to provide appropriate recognition for Early Career Researchers who co-review manuscripts.

Version 2:

Decision Letter:

Our ref: NSMB-A51556B

10th Mar 2026

Dear Dr. Bullock,

Thank you for submitting your revised manuscript "Cryo-EM structures of Egl-BicD-RNA complexes reveal how diverse mRNAs are selected for subcellular localization" (NSMB-A51556B). It has now been seen by the original referees and their comments are below. The reviewers find that the paper has improved in revision, and therefore we are happy to accept it in principle in Nature Structural & Molecular Biology, pending minor revisions to comply with our editorial and formatting guidelines.

Sincerely,

Dimitris Typas
Senior Editor
Nature Structural & Molecular Biology
ORCID: 0000-0002-8737-1319

Reviewer #1 (Remarks to the Author):

This is very high quality paper presenting structure which has long been discussed in the community. The authors have reviewed the paper thoroughly and again, to a high standard and have addressed all of my comments, they have done a great job. The paper is ready to go as far as I am concerned..

If I may add a comment on the minor groove recognition of specific nucleobases, the authors have done a good job of completing the picture, as far as it is possible. Regardless, in my opinion this is not the main point of interest of this work from a mechanistic point of view, and has been explored in sufficient depth already, but I leave to comment to the relevant

reviewer(s).

Reviewer #2 (Remarks to the Author):

The authors have performed excellent revision work, and addressed all my points. Their more systematic investigation of the impact of canonical base pairs at the ED1 and ED2 sites, together with their successful *in silico* design of a stem loop that sustain apical localization *in vivo*, now provides even stronger evidence that they have deciphered the code underlying Egl—BicD-mediated RNA recognition and dynein-dependent transport. Altogether, this manuscript represents an outstanding piece of work that should be published without further delay in *Nat struct mol biol*.

Reviewer #3 (Remarks to the Author):

We believe that the authors have done an excellent job in revising the manuscript, and have effectively addressed the reviewer comments and suggestions. The newly added experimental data substantially strengthened and expanded their conclusions. The visuals and illustrations have also further improved. We recommend this manuscript in its current form for publication with enthusiasm.

Reviewer #4 (Remarks to the Author):

I co-reviewed this manuscript with one of the reviewers who provided the listed reports. This is part of the Nature Structural & Molecular Biology initiative to facilitate training in peer review and to provide appropriate recognition for Early Career Researchers who co-review manuscripts.

Version 3:

Decision Letter:

23rd Mar 2026

Dear Dr. Bullock,

We are now happy to accept your revised paper "Structural basis for recognition of diverse localizing mRNAs by Egl—BicD" for publication as an Article in *Nature Structural & Molecular Biology*.

Acceptance is conditional on the manuscript's not being published elsewhere and on there being no announcement of this work to the newspapers, magazines, radio or television until the publication date in *Nature Structural & Molecular Biology*.

Over the next few weeks, your paper will be copyedited to ensure that it conforms to *Nature Structural & Molecular Biology* style. Once your paper is typeset, you will receive an email with a link to choose the appropriate publishing options for your paper and our Author Services team will be in touch regarding any additional information that may be required.

Your paper will be published online soon after we receive proof corrections and will appear in print in the next available issue. You can find out your date of online publication by contacting the production team shortly after sending your proof corrections.

Authors may need to take specific actions to achieve compliance with funder and institutional open access mandates. If your research is supported by a funder that requires immediate open access (e.g. according to [Plan S principles](https://www.springernature.com/gp/open-science/plan-s-compliance) or the [NIH public access policy](https://www.springernature.com/gp/open-science/us-federal-agency-compliance)) then you should select the gold OA route, and we will direct you to the compliant route where possible. Because authors warrant under our subscription licensing terms that they haven't committed to licensing any version of their article under a licence inconsistent with the terms of our agreement – including the applicable embargo period – publication under the subscription model isn't suitable for authors whose funders require no embargo.

Sincerely,

Dimitris Typas
Senior Editor
Nature Structural & Molecular Biology
ORCID: 0000-0002-8737-1319

We are grateful to all four reviewers for the thoughtful and constructive critiques of our manuscript. Their questions and suggestions have led us to perform several new experiments, analyses and textual revisions that have improved the work substantially. One major addition is that we have determined the effects of all canonical base pairs at the ED1 and ED2 sites, leading us to significantly refine and strengthen our model for how Egl recognizes localization signals. Moreover, we have leveraged this knowledge to engineer a heterologous, inactive stem loop to act as a localization signal. This provides a striking demonstration that we have uncovered a code that defines dynein-dependent mRNA localization signals.

Reviewer #1 (Remarks to the Author):

In this manuscript, Singh and co-workers present the molecular basis for functional mRNA recognition by the *Drosophila* protein Egalitarian (Egl). Egalitarian is an essential component of the mRNA transport machinery, to which provides selectivity via its recognition of structured RNA elements within the target mRNAs. However, how Egalitarian would recognize RNA structures that have been predicted to be different, discriminating RNA targets from non-targets, is a key question we need to answer to establish the molecular bases of local mRNA translation during *Drosophila* development. A second important question is whether/how multiple recognition elements on the same RNA may be required. Here, the authors present a comparative analysis of the 2:1 Egalitarian-BicD protein-protein complex bound to five known RNA targets. They identify the main interactions responsible for recognition and link them to common features of the RNA targets, probing their conclusions by mutating protein and RNAs and testing the effect on i) the affinity of the interactions with the Egl-BicD complex and ii) mRNA localization in the *Drosophila* embryo. The results show that the protein makes contact with different elements in the RNA structure, including two minor grooves and a loop, as well as specifically recognizing individual bases which are precisely positioned in the complex. The recognition of multiple features within the RNA structure is mediated by different protein domains. Interestingly, to achieve this recognition two Egl chains are required, with a domain switch that leads one RNA element to be bound by three domains from one chain and one domain from the other. The complex web of inter-domain contacts imposes a structure selectivity that matches the RNA's conformation. Importantly, some flexibility exists: the domains can rotate and re-orient while maintaining contact, which facilitates the recognition of non-identical RNA structures. Finally, the Egl-BicD complex can interact simultaneously with two RNA elements within the same RNA molecule and, using the structural and biophysical tools discussed above, the authors explore how simultaneous binding of a primary and a secondary recognition element favors recognition and transport. The paper is clear and well organized. The figures are very well drafted and the analysis is informative. In general, the conclusions are very well supported by the data, which are copious. The study provides an initial answer to some of the key questions in the field of mRNA transport and localization. It is a high-quality paper that will have a substantial impact in the field. Below are a few minor points that should be addressed prior to publication.

1) The affinity of the protein complex for the different RNA elements (e.g. TLS vs hSL1) seems to be quite different. Could the authors discuss the mechanistic implications of this?

This is an interesting observation. Mechanistically, different affinities of RNA elements for Egl-BicD likely affect the efficiency by which transport competent RNPs are assembled, as well as their lifetime. This could conceivably modulate the onset and duration of localization of their cognate transcripts *in vivo*, thereby ensuring appropriate spatial and temporal distribution of their protein products. We have added this point to the Discussion of the revised manuscript (page 18, end of paragraph 1).

2) Perhaps I have missed this, but have the authors measured the affinity of the Hsl1-Hsl2 two domain RNA element for the protein complex? They present a structure of the protein-RNA complex, and we have the affinity for the two separate elements, but I do not see the affinity

for the full-length RNA. This would provide some mechanistic information on the extent to which cooperativity/coupling plays a direct role in the *in vivo* effect.

We agree that measuring the affinity of Egl–BicD for an RNA containing both *hSL1* and *hSL2* could be informative. However, we have been unable to manage this because of technical difficulties. In our previous MST experiments, we used commercial synthesis to produce RNAs that had a fluorophore at one end, enabling monitoring of Egl–BicD binding without potentially interfering with the structure of the localization signal. At 126 nt long, the *hSL1-hSL2* sequence is too long for chemical synthesis. We therefore attempted to end-label an *in vitro* transcribed *hSL1-hSL2* RNA with a fluorophore. Despite using several methods and conditions, the labeling efficiency (~3–18%) was too low for meaningful analysis by MST.

We have therefore taken an alternative approach to address the reviewer's point about the extent to which coupling of *hSL1* and *hSL2* has a role in the transport process. We have quantified the frequency of RNA motility in our single-molecule reconstitutions of dynein activity stimulated by *hSL1* stem loop alone or *hSL1-hSL2*. This analysis revealed that, despite *hSL1* having a much higher independent affinity than *hSL2* for Egl–BicD (Extended Data Fig. 8d of the revised manuscript), *hSL1-hSL2* is transported approximately three-fold more frequently than the *hSL1* RNA. These data, which have been incorporated in the revised manuscript (Extended Data Fig. 8e; page 15, paragraph 1), provide further evidence that coupling of elements plays a direct role in stimulating transport *in vivo*.

3) It would be very useful to see a sequence alignment of the different regions of the protein. Are the interfaces between the domains conserved? Are the linkers conserved? Such an alignment is important, can it please be added (and briefly discussed) if not there?

We thank the reviewer for this suggestion. We have now included two different types of sequence alignment for Egl (Supplementary Figs. 1 and 2). The first shows overall patterns of conservation across 108 species, whereas the second allows visualization of complete Egl sequences from several representative species. We have referred to these figures when describing the conservation of domains and linkers in multiple points in the main text. Briefly, we observe that ED1, ED2, and the ExoHD–XAD module are well conserved (page 4, paragraph 1), whereas the linkers between them are not (page 4, paragraph 1). We also find that ED3 is less conserved than the other EDs (page 4, paragraph 1), consistent with its absence in our structures. Finally, we find that both RNA binding (page 5, paragraph 1) and interdomain interfaces (page 5 and paragraph 2) are conserved, supporting their importance for Egl function. Some of these points are also illustrated in new renderings of the structures that display sequence conservation of Egl (Extended Data Fig. 3).

4) 17b and comparison with the free RNA structure: it is great to see the spread of the angle in the family of NMR structures, which is the best way of representing this. Could the authors please remove the error bars, as each of these structures is consistent with the same data ensemble? If the author wants to indicate the value of the angle in the average structure this could be done by a dot of a different colour, although having just the ensemble of structures is probably the best representation. Perhaps the author could report the angles of all the cryoEM and NMR structures side-by-side, in one panel? Regardless, it seems to me there is no meaningful difference between the angle in the NMR structures and the cryoEM one? If that is the case, please reflect this in the main text.

While our intent was to show the variability of the helical bend amongst the structures that comprise the ensemble *TLS* in the NMR model, we now understand how the use of error bars in this plot was problematic. Moreover, we agree that the primary conclusion of this analysis – that the helical bend of the isolated *TLS* is indistinguishable from that observed in our cryo-EM structure – should be made explicit. We have therefore removed the error bars for the NMR data and reproduced the measured bending angles for the Egl-bound *TLS* in the same

panel (Extended Data Fig. 9f). Additionally, we have updated the text in page 16, paragraph 3 of the revised manuscript to clearly state the similarity between the TLS structures obtained in the two studies.

5) Lines 130-133. The fact that interface between domains are not predicted by AlphaFold it is possibly due to their uniqueness, I am not sure the authors need to use the AlphaFold result use to support cooperativity. I would remove this comment.

This point has been removed from the revised manuscript. Instead, we have added a point about our inability to determine a structure of Egl–BicD without RNA being consistent with RNA-induced assembly of the RNA-binding pocket (page 6, paragraph 3). This follows a request from Reviewers #3 and #4 to document attempts to resolve a structure of Egl–BicD in isolation.

6) In general, I find the discussion is very well written, but some points could be explored more in depth. For example, can the authors comment on other such elements that they have not looked at experimentally (they mention a few) and predict the structure? Can they predict more? I understand structure predictions have not always been accurate, but can the 'rules' they have uncovered help? I think some general comment on this would be interesting.

Our data show that Egl recognition of its targets requires precise positioning of bulges and specific base pairs within stem loop structures. Furthermore, an exciting series of experiments suggested by Reviewer #2 shows that these features are sufficient to convert a heterologous, inactive stem loop into a localization signal (Fig. 4h–j). Nonetheless, as the reviewer alludes to, whilst secondary structure prediction algorithms are effective in broadly predicting the presence of structured elements, they are less successful in defining the precise base pairing and bulge positioning within them (e.g., see Supplementary Fig. 8b). As such, a purely computational method for predicting Egl targets remains a challenge despite the extra constraints that our rules apply to searches. Efforts to refine this process are currently ongoing as part of a separate project in our labs, which we believe is well beyond the scope of this study. As requested, we now make a general comment in the Discussion about how our work will facilitate future efforts to identify novel Egl targets (page 16, end of paragraph 2).

Also, why do we need two separate RNA elements to recognize the RNA – the comment on recycling the transport system is a little brief. Could they explain more?

We have expanded the discussion of this point on page 18, paragraph 1 of the revised manuscript to elaborate how simultaneous detection of two stem loops improves specificity and how a low affinity interaction could facilitate disassembly of the transport complex and recycling of its components. We are grateful to the reviewer for the suggestion.

Reviewer #2 (Remarks to the Author):

Subcellular targeting of selected mRNAs was discovered decades ago and since then shown to underlie cell compartmentalization in various cell types. Dissection of the mechanisms driving the localization of few specific RNAs led to the identification of « zipcodes », ie cis-regulatory RNA elements that are both necessary and sufficient for localization. These elements further mediate the binding of RNA binding proteins and the subsequent recruitment of transport machineries. Although large-scale studies have identified populations of mRNAs that co-localize within specific compartments in various cell types, the number of examples where the molecular determinants of zipcode recognition have been resolved remains very limited. Both experimental and computational analyses have so far failed to reveal common RNA features that account for the localization of RNAs to shared cellular destinations. Thus, the field still lacks a unifying understanding of the rules governing the recognition of sets of co-localizing RNAs.

In this study, the authors present a systematic and high-quality characterization of the cryo-EM structure of the Egl-BicD complex bound to 5 different RNA elements, each known to mediate dynein-dependent localization of its corresponding RNAs in *Drosophila*. This system is a model of choice for a number of reasons: i- the RNA localization elements had been previously mapped, with established association with Egl-BicD, ii- these RNA elements lack obvious sequence or structural commonalities, and iii- structural insights can be validated through functional in vitro and in vivo assays.

Together, this study stands out in several respects: first, it provides high-quality cryo-EM structures of the Egl-BicD complex bound to 5 distinct RNA elements previously shown to mediate Egl-BicD-dependent localization. This substantial experimental effort, combined with a careful integrated analysis, led to the identification of key shared (but previously unrecognized) determinants required for localization. Second, the work combines high-quality cryo-EM and quantitative in vitro binding assays with both single-molecule live-imaging in reconstituted systems and RNA localization assays in vivo in early *Drosophila* embryos. This impressive array of advanced methods allowed the authors not only to dissect the molecular contacts governing RNA-protein interactions, but also to demonstrate their functional relevance in the assembly of transport-competent complexes. Third, the new findings shed light on previously reported results, enabling the integration of years of work from different groups into a coherent and elegant mechanistic framework. Last, the quality and clarity of the manuscript, including its text, figures and associated legends were outstanding, leaving little room for meaningful improvement.

Given the outstanding quality of this work, my comments are only minimal:

Major points:

- The authors appear to have uncovered the code governing the localization of mRNAs that depend on the Egl-BicD complex for their dynein-mediated transport. If this is true, an important final step would be to design in silico synthetic RNAs predicted to undergo efficient apical localization when injected in the *Drosophila* embryos. Including such a dataset would provide compelling evidence that the Egl-BicD recognition and localization can be fully predicted.

We agree that this would be a powerful demonstration of the importance of the features we have identified in enabling dynein-mediated RNA localization. To this end, we replaced the TLS of the *K10* 3' UTR with the partially synthetic stem loop from the '*2NUE*' RNA (a modified T7 phage sequence), which we contrast with Egl-BicD targets throughout the manuscript as a straight RNA helix that does not bind the adaptor complex. In line with this, we found that swapping the TLS for the *2NUE* stem loop completely abolishes localization of *K10* RNA when injected into *Drosophila* embryos. We then modified the *2NUE* stem loop in this RNA to include the defining features of localization signals derived from our structures and mutagenesis assays (which we have now refined in experiments requested by Reviewers #3 and #4). Remarkably, inclusion of two single U-A base pairs spaced 8 base pairs apart for ED1 and ED2 binding, together with 3' bulges between these positions to enable helical bending, completely restores localization of the engineered *K10* RNA to wild-type levels. Neither of these modifications rescue localization on their own, indicating that the presence of both features is needed for mRNA transport. We have included these results in Figure 4h-j and page 11, paragraph 3 of the revised manuscript. We are very grateful to the reviewer for the suggestion to take this highly informative approach.

- Another well-characterized localization element that requires Egl-BicD for its dynein-dependent localization is the OES of oskar mRNA (Jambor et al., 2014; doi: 10.1261/rna.041566.113.). Could the authors discuss how the OES fits with their proposed unifying model of recognition?

The OES is one of several additional localization signals that have been proposed to mediate transport by Egl–BicD–dynein–dynactin. It is not possible to determine with confidence how any of these signals fit our model of recognition for an individual stem-loop. This is because predicted secondary structure can differ from the real secondary structure, as we previously showed for the localization signals that are characterized in our study (Supplementary Fig. 8b). What is clear, however, is that *oskar* conforms to the criterion of requiring multiple stem-loops for robust localization by Egl (Mohr et al., 2021). We have therefore included *oskar* in the list of mRNAs that require regions beyond their primary localization signal for efficient transport (page 18, start of paragraph 2).

Minor points

- Abstract : “we present cryo-EM structures of Egl–BicD in complex with 6 different RNAs”: I counted 5, not 6.

In addition to the structures of Egl–BicD with five different primary localization signals, we also present a sixth structure of Egl–BicD bound to an RNA bearing one of these signals (*hSL1*) and its support element (*hSL2*). We have clarified this point on page 16, paragraph 2.

- Discussion: could the authors comment on how co-binding to stem loops that are hundreds of nucleotides apart within the same RNA molecule might modulate 3'UTR architecture?

The reviewer raises the fascinating possibility that binding of Egl dimers to two stem loops within the same RNA molecules leads to rearrangement of other features in the RNA that might affect their presentation to other RBPs. In response to the reviewer’s comment, we now speculate about this possibility in the Discussion of the revised manuscript (page 18, paragraph 2).

- Have the authors examined the phylogenetic conservation of primary localization signals versus support elements (through simple sequence alignments across divergent species)?

We have now performed RNA sequence alignments of these elements, as requested. These are included in the revised manuscript as Supplementary Figure 9 and described in page 10 (end of paragraph 2), page 11 (paragraph 2), and page 18 (paragraph 1). We observe that primary localization signals are extremely well-conserved between divergent *Drosophila* species. Sequence variation in these elements is generally restricted to the loop and bulged regions, consistent with our finding that these features are not bound directly by Egl–BicD. The finding that conservation extends beyond the ED1 and ED2 binding sites to include additional base-paired regions suggests these features either play a structural role in optimizing presentation of the RNA determinants to Egl–BicD or are bound by other RBPs involved in aspects of the mRNA’s biology beyond dynein-mediated transport (a point included in the legend to this figure). In contrast, the known localization support elements are less conserved across species. This likely reflects their lower-affinity mode of interaction with Egl–BicD imposing fewer constraints, as supported by our structural work on recognition of the *hSL1–hSL2* RNA.

Reviewer #3 and #4 (Remarks to the Author):

This manuscript by Singh, Chilaeva, and McClintock et al., reports high-resolution Cryo-EM structures of the *Drosophila* Egl–BicD adaptor complex bound to six functionally distinct, developmentally important localization signal-containing RNAs. The structural dissection of non-canonical dsRNA recognition by four Egl domains (ED1, ED2, ExoHD, and XAD) reveals a novel mechanism of cooperative binding and domain swap by flexibly tethered domains. These structures converge on the recognition of a ~20° bent stem loop consisting of an upper

stem and lower stem separated by a bulge in the 3' strand. Using biophysical assays (FP, MST) and in vivo and in vitro RNA motility assays, the authors corroborate their structural findings with mutational analyses, provide support for a dual stem-loop recognition model that licenses transport, and identify a new secondary localization signal (KSE) in the K10 mRNA. The authors also test the sequence identity requirements and argue for a steric occlusion model where Cys35 or Gly124 reject guanines thereby imposing an U-A sequence preference. The manuscript is well-organized, clearly written and expertly illustrated. The work addresses an important question in cell and developmental biology — how specific mRNAs are selected for intracellular transport. It further provides substantial novel insights into RNA recognition by multi-domain RNA-binding proteins, and is thus a strong candidate for NSMB. However, there are several major concerns that should be addressed with additional experiments before publication, and some minor suggestions below intended for enhancing accuracy, clarity and readability of the manuscript.

Specific comments:

1. The most significant concern that I have is the incomplete mutagenesis data in support of the proposed guanine rejection mechanism by steric occlusion. The argument based on structural modeling, especially of the ion or water density close to Gly124/Ser127 clashing with the 2-amino of guanine, is strenuous (Fig. 3, ExFig. 13). All the examples in this study seem to have 5'U-3'A pairs instead of 5'A-3'U pairs. Why is it so? Is it due to the purine N3 interaction? The steric occlusion model does not address this sequence preference. Distances (or extents of overlap) are not provided in support of the proposed clashes. Some of the densities don't look great (e.g. ExFig. 13g), likely due to the inherent noise in the cryo-EM images as expected. If a guanine does bind in this location, there simply wouldn't be space to accommodate the loosely tethered ion/water, not that the ion/water here would necessarily prevent the guanine from binding due to a tightly bound structural role. As far as I can tell, the authors only tested the U-A to G-C pair, which is a fairly drastic transversion mutation. I suggest the authors also test A-U, C-G, and even U-G wobble pairs. The U-A to A-U swap will test if the N3-recognition by His31/Ser127 is functionally important, and specifies the purine preference in this strand. The U-A to C-G will maintain the N3 interaction to the purine, while not creating potentially disruptive transversions. One might expect this one to bind okay. A U-A to U-G substitution will not only maintain the purine N3 interaction, but also shift the U6/U15 up and away from the Cys35/Gly124 ion/water to preempt any steric clash. Notably, the G-U wobble pair would also alter the twist of the dsRNA duplex, which could also be important for binding.

We are very grateful to the two reviewers for these suggestions. We have now assessed the binding of Egl-BicD to *TLS* mutants in which both base pairs that interact directly with ED1 and ED2 are mutated from U-A to C-G (CG^{ED1/ED2}), G-C (GC^{ED1/ED2}), or A-U (AU^{ED1/ED2}). We did not test the effects of U-G substitutions because, as the reviewers acknowledge, this mutation likely alters several aspects of the RNA structure, making any changes in the affinity for Egl-BicD difficult to interpret.

All of the variants we tested significantly reduce binding of the *TLS* to Egl-BicD, revealing a preference for U-As at the ED1 and ED2 sites. Consistent with the data for the 2xGC^{ED1}+GC^{ED2} mutant presented in the original manuscript, binding of the new GC^{ED1/ED2} mutant to Egl-BicD is dramatically impaired. Additionally, we found that Egl-BicD binding to the CG^{ED1/ED2} and AU^{ED1/ED2} mutants is reduced to a lesser extent, with the affinity for CG^{ED1/ED2} nearest that of the wild-type *TLS* (new Fig. 4e,f). In other words, 5' pyrimidine-3' purine base pairs are preferred to 5' purine-3' pyrimidine base pairs at the ED binding sites, with U-A favored over C-G and A-U favored over G-C within the respective configurations. Whilst these observations are consistent with a detrimental effect of guanine, as proposed previously, they identify an additional determinant of binding, which is a 5' pyrimidine-3' purine configuration at these sites. Collectively, this explains the enrichment of conserved U-A base

pairs across localization signals (please see next point). We now propose structural mechanisms that could account for our observations (page 10, paragraph 4/page 11, paragraph 1, and page 17, paragraphs 2 and 3), including those derived from new modeling efforts that suggest that U–A base pairs are distinctly complementary to the ED interaction interfaces (Extended Data Fig. 7a,b). We have included in these sections the reviewers' well taken points about displacement of the ion/water and potential hydrogen bonding preferences for purine vs pyrimidine bases.

We feel that the new experimental series suggested by the reviewers has led to significant improvements to this aspect of the study by providing a much more complete illustration of Egl's selectivity.

2. Related to #1, is there bioinformatics information available or attainable for the sequence conservation or co-variation at these binding sites? Such information would help clarify whether there is indeed guanine avoidance here and if so whether the avoidance applies to both strands of the dsRNA stem.

A sequence alignment of the localization signals and support elements was also requested by Reviewer #2 and is included in the revision as Supplementary Figure 9. This shows that the base pairs of primary localization signals that are contacted by ED1 and ED2 in our structures are well conserved (as well as being devoid of guanine in divergent species). This is consistent with our refined model for base pair selectivity at these positions.

On the protein side, are Cys35, Gly124, Ser127 conserved? If Cys35 is substituted by a less bulky side chain, say in another species, would the authors expect a loss or reduction of sequence specificity of RNA transport?

We have included sequence alignments of Egl in multiple species in response to a request by Reviewer #1 (Supplementary Figs. 1 and 2). This shows that Cys35, Gly124, Ser127 are conserved.

3. The authors demonstrate strong binding and provide high-resolution structures for relevant cargo RNAs. To fully establish the principle of specific cargo selection, the manuscript should consistently include negative controls using one or more unrelated, non-localized RNA of similar size and/or secondary structure for their assays, such as the MST data using an unrelated straight dsRNA in ExFig. 11e.

In the original manuscript, we ensured the consistency of MST affinity measurements by regularly assessing the binding of our Egl–BicD preparations to the wild-type *TLS* (Extended Data Fig. 5 in the original manuscript, now Extended Data Fig. 4). These tests typically also included a *TLS* mutant control (*TLS*^{Δb}) that exhibits significantly diminished binding to Egl–BicD (Dienstbier et al., 2009). This selectivity was demonstrated in Extended Data Fig. 12a of the original manuscript (now Extended Data Fig. 6a). However, for the purposes of clarity, the composite plot of all *TLS*^{Δb} measurements was excluded from the original Extended Data Fig. 5. These data have now been added to the equivalent figure in the revision (Extended Data Fig. 4a,b) to more comprehensively demonstrate the reproducible specificity of the assay.

Regarding the specificity of Egl–BicD for the other localization signals, this was also established in Extended Data Fig. 12 of the previously submitted manuscript. This figure showed compromised binding of Egl–BicD to bulge deletion mutants that disrupt mRNA localization in vivo. Nonetheless, we appreciate how the 'fraction bound' normalization of the binding curves (which precludes the fitting of data for non-binding RNAs) could give the impression that specificity had not been established in panels c–e of Extended Data Fig. 12. Therefore, for each signal tested in this figure (now Extended Data Fig. 6), we now also normalize the binding data using ΔF_{norm} , which enables plotting of data from the mutants that

did not detectably bind Egl–BicD. We believe that this more clearly demonstrates specific binding of wild-type localization signals in these experiments.

In addition to the points above, all new MST data presented in the revised manuscript were acquired with a side-by-side negative control of the straight *2NUE* stem loop, as requested by the reviewer. These results are presented in Extended Data Fig. 5e, the legend to which points out their acquisition alongside the data shown in Fig. 4f.

4. It would be informative to mention whether attempts were made to determine cryo-EM structures of Egl/BicD proteins without RNA since their interactions don't require RNA, or low-affinity, non-cargo RNA complexes. If such data exist, the authors could comment on whether 3D classification revealed conformational changes, empty RNA-binding sites, or significant flexibility at the interface, which would shed light on the resting state of the proteins or structural consequences of non-recognition.

We and others have long sought high-resolution EM models of Egl–BicD without RNA. However, this has not been successful. The RNA-bound structures in our manuscript offer an explanation for why this has been the case. In the absence of RNA, Egl's RNA-binding domains are expected to move freely about their flexible linkers and therefore not produce consistent class averages that are needed for high resolution structures. We propose that interaction interfaces between these domains are only stabilized in fixed positions in the presence of localization signals, indicative of cooperative interactions that form during recognition of RNA targets. Use of exclusively low-affinity or non-target RNAs will likely compromise some or all of these interactions, leading to the same flexibility issues that preclude structure determination in the absence of RNA. Nonetheless, our structure of Egl–BicD bound to the *hairy* localization element, which includes the primary *hSL1* element and the lower-affinity *hSL2* accessory stem loop, provides insight into how low affinity RNA elements can be recognized.

We recognize that other readers are likely to be curious about structural analysis of the RNA-free Egl–BicD complex. We have therefore included a brief summary of points raised above in page 6, paragraph 3.

5. For the hairy RNA complex, if feasible please provide additional binding assays to verify that the ExoHD and XAD domains are not interacting with the low-affinity element *hSL2*, thereby strengthening the conclusion regarding the selective engagement of domains with each stem loop.

We believe that the lack of any detectable ExoHD–XAD density in the *hSL2* pocket of the Egl–BicD–*hSL1*–*hSL2* structure provides compelling evidence that these domains do not engage this RNA element. Moreover, the structure rationalizes why this is the case (page 16, paragraph 1). Therefore, we elected to focus the time we had available for revisions on the other experiments suggested by the reviewers. As we cannot completely exclude a fleeting interaction of *hSL2* with ExoHD–XAD, we have adjusted the language on page 16, paragraph 1 of the revised manuscript to conclude that we do not detect association of these domains with *hSL2* in the structure.

6. ExFig. 11a, the author should label the domains and potentially also show where the RNA would be in this uncommon viewing angle, which could be translucent so as not to obscure the view of the protein motions.

This is another excellent suggestion. The figure (now Extended Data Fig. 5a) has been updated accordingly.

7. Line 225. The authors state “whereas His31 makes a base-agnostic hydrogen bond to help

lock the domain in place". It is not clear that this interaction is truly sequence non-specific without direct testing. The structures only show His31 binding to N3 of purine when in a Watson-Crick pair and distorted O2 within a U48xC9 mismatch. It is not known if a cytosine here, or a uridine in a canonical base pair would bind His31 with similar strengths as a purine N3, which seems more favorable. Further, the O2 groups on cytosine and uridines are not chemically equivalent.

We acknowledge that the language we chose implied complete equivalence of the hydrogen bonds made by His31/Ser127 with any of the four nucleobases in the minor groove. However, the absolute strengths of these interactions will depend on the local chemical environment, which will change with any mutation. Accordingly, we have removed the term 'agnostic' when discussing the interactions. Further, we now incorporate the possibility that differences in hydrogen-bonding propensities contribute to recognition (page 17, paragraph 3).

8. ExFig. 16e. The numerous arrows obscure the view of the new location/confirmation of ED2. It is likely better to show an overlay instead of using the strong arrows in this panel.

We thank the reviewer for this suggestion. The panel (Extended Data Fig. 9d) has been updated with an overlay, as requested.

9. Fig. 2f and ExFig. 10, the authors use three shades of pink/magenta/purple as the sole distinguishing feature for the sets of data points. More contrasting colors are likely needed.

Fig. 2f and Extended Data Fig. 10a (now Supplementary Fig. 8) have been updated with alternative colors or labeling styles to make the data easier to distinguish. For consistency, these schemes have been adopted elsewhere in the manuscript.

10. Several further points regarding manuscript presentation below could be addressed to enhance clarity and completeness: a) The authors could include representative gel images (e.g., SDS-PAGE) to demonstrate the purity of all protein constructs used in the binding and structural experiments. b) Extended Data Fig. 4e and 4f do not appear to be cited in the main text. c) The authors should consider including the Lotus domain in the protein sequence alignment shown in ExFig. 1b.

a) An SDS-PAGE gel image of all protein constructs used in the binding and structural experiments is now included as Extended Data Fig. 1f .

b) Callouts for Extended Data Fig. 4e and 4f were made twice in lines 109-110 and 145 of the original manuscript, appearing in both the main text and the legend for Figure 2. We have made sure this is also the case for the equivalent figure in the revised manuscript.

c) Comparison of the *Drosophila* EDs with the LOTUS domain of human TRDR5 was based on structural similarity. There is very little sequence homology between these domains, which precludes their comparison by sequence alignment.

References:

Dienstbier, M., Boehl, F., Li, X. & Bullock, S.L. (2009). Egalitarian is a selective RNA-binding protein linking mRNA localization signals to the dynein motor. *Genes Dev.*, 23, 1546–1558.

Mohr, S., Kenny, A., Lam, S. T. Y., Morgan, M. B., Smibert, C. A., Lipshitz, H. D. & Macdonald, P. M. (2021) Opposing roles for Egalitarian and Staufen in transport, anchoring and localization of *oskar* mRNA in the *Drosophila* oocyte. *PLoS Genet.* 17, e1009500.